# Determining the infrared radiative effects of Saharan dust: a radiative transfer modelling study based on vertically resolved measurements at Lampedusa

Daniela Meloni[1], Alcide di Sarra[1], Gérard Brogniez[2], Cyrielle Denjean[3,4], Lorenzo De Silvestri[1], Tatiana Di Iorio[1], Paola Formenti[3], José L. Gómez-Amo[5], Julian Gröbner[6], Natalia Kouremeti[6], Giuliano Liuzzi[7,8], Marc Mallet[9], Giandomenico Pace[1], Damiano M. Sferlazzo[10]

*Correspondence to:* Daniela Meloni (daniela.meloni@enea.it)

[1]Laboratory for Observations and Analyses of Earth and Climate, ENEA, Rome, Via Anguillarese 301, 00123, Italy
[2]Laboratoire d'Optique Atmosphérique, University of Lille 1, France
[3]Laboratoire Interuniversitaire des Systèmes Atmosphériques, UMR-CNRS 7583, Université Paris-Est-Créteil et Université Paris Diderot, Institut Pierre Simon Laplace, Créteil, France
[4]now at Centre National de Recherches Météorologiques, UMR 3589, CNRS, Météo-France, Toulouse, France
[5]Departament de Física de la Terra y Termodinàmica, Universitat de València, Spain
[6]Physikalisch-Meteorologisches Observatorium Davos/World Radiation Center, Davos Dorf, Dorfstrasse 33, 7260, Switzerland
[7]Solar System Exploration Division, NASA Goddard Space Flight Center, 8800 Greenbelt Rd., Greenbelt, MD 20771, USA
[8]Department of Physics, American University, 4400 Massachusetts Avenue NW, Washington, DC 20016, USA[9]CNRM UMR 3589, Météo-France/CNRS, Toulouse, France
[10]Laboratory for Observations and Analyses of Earth and Climate, ENEA, Lampedusa, Contrada Capo Grecale, 92010, Italy

**Abstract.** Detailed atmospheric and aerosol properties, and radiation measurements were carried out in summer 2013 during the Aerosol Direct Radiative Impact on the regional climate in the MEDiterranean region (ADRIMED) campaign in
the framework of the Chemistry-Aerosol Mediterranean Experiment (ChArMEx) experiment. This study focusses on the characterization of infrared (IR) optical properties and direct radiative effects of mineral dust, based on three vertical profiles of atmospheric and aerosol properties and IR broadband and narrowband radiation from airborne measurements, made in conjunction with radiosonde and ground-based observations at Lampedusa, in the central Mediterranean. Satellite IR spectra from the Infrared Atmospheric Sounder Interferometer (IASI) are also included in the analysis. The atmospheric
and aerosol properties are used as input to a radiative transfer model, and various IR radiation parameters (upward and downward irradiance, nadir and zenith brightness temperature at different altitudes) are calculated and compared with observations. The model calculations are made for different sets of dust size distribution (SD) and refractive index (RI), derived from observations and from the literature. The main results of the analysis are that the IR dust radiative forcing is non negligible, and strongly depends on SD and RI. When calculations are made using the *in situ* measured size
distribution, it is possible to identify the refractive index that produces the best match with observed IR irradiances and brightness temperatures (BTs). The most appropriate refractive indices correspond to those determined from independent measurements of mineral dust aerosols from the source regions (Tunisia, Algeria, Morocco) of dust transported over Lampedusa, suggesting that differences in the source properties should be taken into account. With the *in situ* size distribution and the most appropriate refractive index the estimated dust IR radiative forcing efficiency is +23.7 W m$^{-2}$ at
the surface, -7.9 W m$^{-2}$ within the atmosphere, and +15.8 W m$^{-2}$ at the top of the atmosphere. The use of column integrated

dust SD from AERONET may also produce a good agreement with measured irradiances and BTs, but with significantly different values of the RI. This implies large differences, up to a factor of 2.5 at surface, in the estimated dust radiative forcing, and in the IR heating rate. This study shows that spectrally resolved measurements of BTs are important to better constrain the dust IR optical properties, and to obtain a reliable estimate of its radiative effects. Efforts should be directed at obtaining an improved description of the dust size distribution, its vertical distribution, and at including regionally-resolved optical properties.

## 1 Introduction

Aerosol radiative effects in the infrared or longwave (LW) spectral range have been recognized to be non-negligible compared to that in the solar region for large particles like marine aerosols (e.g. Markowicz et al., 2003) and desert dust (e.g. Vogelmann et al., 2003; Otto et al., 2007; Osborne et al., 2011; di Sarra et al., 2011; Sicard et al., 2014; Meloni et al., 2015 and references therein). The major uncertainty affecting the LW aerosol direct radiative effect (ARF) is due to the poor knowledge of the IR aerosol optical properties, i.e. the aerosol optical depth, single scattering albedo and phase function.

Desert dust has the largest capability to perturb the IR radiative field, due to the particles' large size and the abundance at the global level, which peaks in the arid regions and their surroundings (Prospero et al., 2002; Ginoux et al., 2012).

Several field campaigns worldwide have been devoted to study the dust microphysical, chemical, and optical properties, with a focus in the infrared radiative effects: Saharan Dust Experiment, SHADE (Highwood et al., 2003), Radiative Atmospheric Divergence using Atmospheric Radiation Measurement (ARM) Mobile Facility, Geostationary Earth Radiation Budget (GERB) data, and African Monsoon Multidisciplinary Analysis (AMMA) stations, RADAGAST (Slingo et al., 2008), Saharan Mineral Dust Experiment, SAMUM (Ansmann et al., 2011), Geostationary Earth Radiation Budget Intercomparison of Longwave and Shortwave radiation, GERBILS (Haywood et al., 2011), NASA AMMA, NAMMA (Hansell et al., 2010), carried out in Africa, Asian Pacific Regional Aerosol Characterization Experiment, ACE-Asia (Seinfeld et al., 2004), and the Asian Monsoon Years field experiment (Hansell et al., 2012) in Asia. Most of these studies have been carried out close to the dust source regions and did not take into account the possible modifications in dust optical properties during long range transport. Other papers (e.g. Maring et al., 2003; Ryder et al., 2013; Weinzierl et al., 2017) examine the microphysical and/or chemical and/or optical properties of transported dust through the Atlantic Ocean.

A great improvement in the detection of dust in terms of spatial and temporal coverage has been reached after the development of specific algorithms applied to the thermal infrared (TIR) channels of satellite sensors, such as Meteosat (Legrand et al., 2001). More recently, the observations from multichannel sensors, such as MSG-SEVIRI (Schepanski et al., 2007; Brindley and Russell, 2009), high resolution spectrometers like Aqua-AIRS (De Souza-Machado et al., 2006), or interferometers like MetOp-IASI (Peyridieu et al., 2013) have been used to derive dust optical and microphysical properties, taking advantage of the spectral signature of dust in the IR (Clarisse et al., 2013; Capelle et al., 2014; Cuesta et al., 2015). However, the algorithms developed to infer dust properties from measured BT require that the spectral variation of aerosol extinction, which depends on the spectral complex RI and the particles' SD (Klüser et al., 2012; Vandenbussche et al., 2013), is known. Thus, understanding the impact of different aerosol optical properties (AOPs) on the infrared BT is crucial

for the satellite retrieval. Similarly, IR aerosol optical properties are crucial in determining the LW dust radiative effects in the atmosphere and at the surface (e.g., Meloni et al., 2015).

The Mediterranean Sea is very often affected by dust intrusions from the close Sahara desert (Meloni et al., 2008; Israelevich et al., 2012; Gkikas et al., 2013) producing significant perturbations to the solar and the IR radiation balance (di Sarra et al., 2011; Perrone et al., 2012; Meloni et al., 2015).

Lampedusa island, in the southern portion of the Central Mediterranean, is close to the Tunisian coasts, and hosts the ENEA Station for Climate Observations (35.52° N, 12.63° E, 45 m above sea level), operational since 1997. Due to its position, the small area (22 km$^2$) and flat surface surrounded by open sea, Lampedusa is an ideal site for the monitoring of transported dust physical and chemical properties. Measurements of aerosol optical depth (AOD) and vertical distribution started in 1999 (Di Iorio et al., 2009; di Sarra et al., 2015); in 2000 Lampedusa became an AERONET (Holben et al., 2001) site. Measurements of downward broadband SW and LW irradiances were started in 2004 (di Sarra et al., 2008; Di Biagio et al., 2010), together with the collection of filters to derive PM10 concentration and chemical composition (Becagli et al., 2012; Marconi et al., 2014).

The Ground based and Airborne Measurements of Aerosol Radiative forcing (GAMARF) field campaign was focussed on the determination of the LW radiative effects of Saharan dust and took place in Lampedusa in spring 2008 (Meloni et al., 2015). The campaign took great advantage from the measured vertical profiles of aerosol SD and LW irradiance from an ultralight aircraft, allowing to derive vertically resolved ARF and aerosol heating rate (AHR) for a case of desert dust intrusion. Moreover, the sensitivity of the SW and LW ARF and AHR to different AOPs was examined.

More recently, Lampedusa also hosted one of the ground-based sites of the Aerosol Direct Radiative Impact on the regional climate in the MEDiterranean region (ADRIMED) field campaign within the Chemistry-Aerosol Mediterranean Experiment (ChArMEx, http://charmex.lsce.ipsl.fr). The ChArMEx/ADRIMED experiment was held from 11 June to 5 July 2013 (Mallet et al., 2016). It aimed at characterizing the different aerosol particles and their radiative effect using airborne and ground-based measurements collected in the western and central parts of the basin.

In particular, ChArMEx/ADRIMED deployed two aircrafts based at the Cagliari's airport (Italy), the ATR-42 and Falcon 20, operated by the Service des Avions Français Instrumentés pour la Recherche en Environnement (SAFIRE).

During the campaign, different Saharan dust intrusions were observed at Lampedusa, mainly from 22 to 28 June and from 2 to 4 July 2013.

In this paper, we present three case studies, selected to represent desert dust of different load and properties (mainly the vertical profile of the size distribution), and for which airborne measurements are available: 22 and 28 June, and 3 July.

This study contributes at understanding how AOPs determine the LW radiative effects at the surface, in the atmosphere and at the top of the atmosphere, based on the combination of remote sensing and *in situ* observations from the ground, from airborne sensors, and from space, and radiative transfer modelling.

The novel aspect of this study is that the closure experiment is carried out not only on LW irradiance, but also on infrared spectral BT. In particular, the impact of different AOPs on the nadir spectral radiances measured by the airborne Conveyable Low-Noise Infrared Radiometer for Measurements of Atmosphere and Ground Surface Targets (CLIMAT) on-board the ATR-42 and IASI and by a ground-based zenith pyrometer is investigated. To our knowledge, this is the first closure experiment carried out simultaneously modelling the observations of instruments with different spectral intervals and on different platforms.

This paper is organized as follows: an overview of the ground-based, airborne and satellite observations used in the study is given in Sect. 2. Section 3 describes the radiative transfer model set-up and output, with a focus on the choice of the adopted aerosol optical properties, and the model sensitivity to the input parameters. The results of the model calculations and the comparison with measurements are presented in Sect. 4 in terms of surface irradiance and sky BT, airborne irradiance and BT profiles, and satellite spectral radiance at TOA, with associated uncertainties. Finally, the vertical profiles of ARF and AHR for the 22 June case are discussed.

## 2 Instruments and measurements

The complete list of the instruments deployed at Lampedusa during the campaign is provided by Mallet et al. (2016), while Denjean et al. (2016) describes the ATR- 42 airborne instruments. Here the instruments and measured parameters used for the radiative closure study are presented.

### 2.1 Surface observations

Measurements of AOD from the AERONET Cimel standard sunphotometer #172 are available after 17 June 2013, with some interruptions during the campaign. The Multi Filter Rotating Shadowband Radiometer (MFRSR) AOD measurements are also available for the whole time period. The Cimel sunphotometer was calibrated at the AERONET-EUROPE RIMA site in Valladolid just before the campaign, and again in January 2015. The MFRSR is regularly calibrated on site applying the Langley technique, and AOD measurements are corrected by means of empirical functions that take into account the forward scattering caused by desert dust particles, which were derived comparing about four years of simultaneous MFRSR and Cimel measurements at Lampedusa (di Sarra et al., 2015). Here the MFRSR AOD measurements are used because of their higher temporal resolution (about 1 minute) compared to that of the Cimel sunphotometer.

Figure 1 shows the time series of the MFRSR corrected AOD at 500 nm and of the Ångström exponent, α, calculated from the AOD at 500 and 870 nm. The first days of the campaign are characterized by low aerosol load (AOD at 500 nm below 0.15) and mainly small particles (most of the values of α between 0.7 and 1.75); since 20 June, the aerosol burden increases with a corresponding decrease in α, indicating the intrusion of desert dust (Denjean et al., 2016). The peak AOD is reached on 22 June. In the following days the alternation of low and moderate AOD conditions occur, but with α values always below 1.5, suggesting a mixing of particles with different dimensions, including desert dust.

The estimated uncertainty on AOD from the MFRSR is <0.02 for the typical conditions at Lampedusa (di Sarra et al., 2015).

The column volume SD is retrieved in 22 size bins between 0.05 and 15 µm (optically equivalent radius) from the Cimel sunphotometer radiance measurements by the inversion method of Dubovik and King (2000). The accuracy of the SD is estimated by Dubovik et al. (2002a) to be below 10% in the maxima and 35% in the minima for particle radii between 0.1 and 7 µm, increasing above 85% outside the interval. Level 2 quality assured inversion products are obtained only for AOD at 440 nm larger than 0.4: this is the case only for the Cimel SD retrieval on 3 July, while on 22 and 28 June Level 1.5 SDs are used.

Column-integrated water vapour (IWV) is obtained from the Humidity And Temperature PROfiler (HATPRO), a microwave radiometer developed by Radiometer Physics GmbH (Rose et al., 2005). The instrument measures the sky

BT at the high and low-frequency wings of the absorption water vapour and oxygen complex, centred at 22.235 and 60 GHz, respectively. The IWV is derived from the observations in the vertical mode, carried out every second, with a $\pm0.2$ mm uncertainty. Compared to Cimel sunphotometer, the HATPRO provides a much larger dataset with sensibly smaller uncertainties ($\pm1.5$ mm for the standard Cimel, according to Holben et al., 2001).

Direct, downward diffuse and global SW irradiance and downward LW irradiance are routinely monitored at Lampedusa. The downward LW irradiance is measured with a Kipp&Zonen CGR4 pyrgeometer, mounted on a solar tracker with shading balls.

As additional instrumentation deployed during the campaign, the Arctic Radiation Budget Experiment (ARBEX) system from the Physikalisch-Meteorologisches Observatorium Davos/World Radiation Center (PMOD/WRC) was placed on the

roof of the building. The ARBEX setup includes Kipp&Zonen CM22 and CM21 pyranometers for downward and upward SW irradiance and Eppley PIR and Kipp&Zonen CG4 pyrgeometers for downward and upward LW irradiance. A modified Kipp&Zonen CGR3 pyrgeometer only sensitive to the 8-14 µm band is also part of ARBEX. All radiometers were freshly calibrated: the modified pyrgeometer with the reference blackbody of the PMOD/WRC at Davos and the broadband ones by comparison with the respective World Standard Groups hosted by the PMOD/WRC in order to guarantee high quality

measurements. The measurement uncertainties (one sigma) of a standard and of the modified pyrgeometer calibrated at the WRC are $\pm2.3$ W m$^{-2}$ and $\pm1$ W m$^{-2}$, respectively (Gröbner et al., 2009). Considering the additional uncertainty associated to the data acquisition system (Meloni et al., 2012), an overall two sigma uncertainty on the broadband pyrgeometer measurements of $\pm5$ W m$^{-2}$ can be assumed. The CGR3 pyrgeometer was not shaded; laboratory tests have shown that solar leakage effects are negligible.

The mean bias (with one standard deviation) between the ENEA and the PMOD/WRC pyrgeometers during nighttime periods between June and September 2013 is $1.4\pm1.2$ W m$^{-2}$, i.e. within the threshold value of $\pm2$ W m$^{-2}$ defined by Philipona et al. (2001) for well-calibrated instruments.

A KT19.85 II Heitronics Infrared Radiation Pyrometer (IRP) measures the zenith sky BT in the 9.6-11.5 µm spectral interval with a narrow FOV (about 2.6°). The IRP accuracy is estimated as $\pm0.5$ K plus 0.7% of the temperature difference

between the instrument body and the observed target (IRP Operational Instructions, 2008). The total accuracy is estimated in $\pm1$ K for the operational conditions of the campaign. The pyrometer is housed in a ventilated box to reduce the measurement degradation due to dirt deposition on the IRP window: for the same reason, the instrument looks at the zenith through a gold mirror. During the campaign the mirror was cleaned every day. The IRP measurements were started on 22 June.

Figure 2 shows the time evolution of the broadband LW irradiance, of the irradiance in the 8-14 µm interval (from now on the WINDOW irradiance), and the infrared zenith BT during the campaign. All quantities are modulated by the atmospheric temperature and humidity; large increases in the irradiance and BT values are due to clouds, whose radiative effect is much larger than that associated with aerosols. Although the LW and WINDOW curves follow similar paths, some differences exist due to the different sensitivity of the two instruments to water vapour: the broadband pyrgeometer is sensitive to both

IWV and to the water vapour present in the lowest atmospheric layers, while the modified pyrgeometer is mostly sensitive to the IWV. The pyrometer has a minimum detection limit of 223 K, and lower BT values are cut. The BT behaviour resembles that of the irradiances, although some differences are present and are attributed to the different instrumental FOVs, with a larger variability for the IRP BT. Indeed, a cloud entering the narrow IRP causes a relevant increase in BT

(depending on the cloud properties), while its impact on the irradiance is minor if the rest of the hemisphere is free from clouds.

Aerosol backscattering, depolarization, and colour ratio profiles at 532 and 1064 nm wavelengths are obtained by a backscatter lidar developed by the University of Rome "La Sapienza" and ENEA, and operated at Lampedusa since 1998. The source is a Nd:YAG solid state laser, while the receiving system is composed of various telescopes detecting different altitude ranges in order to retrieve the backscattered signal from about 50 m to the tropopause. The lidar system can be operated in full daylight conditions thanks to the use of narrowband filters and analog signal detection. The details about the instrument set up and the inversion technique are provided by Di Iorio et al. (2009).

Vaisala RS92 radiosondes were launched in conjunction with the flights (not available on 22 June). Top altitudes reached by the sondes were between 25.5 and 33 km.

## 2.2 Airborne observations

The ATR-42 and Falcon 20 flight tracks during the ADRIMED campaign are summarized in Mallet et al. (2016).

In particular, the ADRIMED flights reached Lampedusa during dusty conditions, on 22 and 28 June, and on 2 and 3 July. 22 June and 2 July were the days with the largest AOD values throughout the ADRIMED campaign. On 22 June a trough located between France and Italy caused the transport of dust aerosols over the central Mediterranean basin and to Lampedusa with southwesterly winds at 3 km height. In the following days an upper-level low developed over central Europe, inducing a westerly flow from Tunisia at 700 hPa, with near-surface northwesterly winds. On 3 July the well-established Azores anticyclone forced northwesterly winds over Lampedusa, thus contributing to dust transport in the area. The meteorological fields and the maps of dust concentration for the days of the flights are shown and discussed in Denjean et al. (2016), as well as the aircraft measurement strategy.

The ATR-42 sounded the atmosphere during descents and ascents to infer the vertical structure and composition and to identify layers with different properties; on 22 and 28 June the ATR-42 landed at Lampedusa airport where it refuelled prior to taking off. The Falcon 20 flew above the ATR-42, following paths at nearly constant altitudes. In this study, only data from the descent portions of the ATR-42 flights (respectively, profile F35 on 22 June, F38 on 28 June, and F42 on 3 July) are used, since during the ascents the aircraft attitude is seldom horizontal. The 2 July flight is not analysed since the altitude interval sounded by the ATR-42 is too small to allow radiative closure experiments.

Table 1 summarizes the time, average AOD and IWV, and altitudes of the ATR-42 descents and Falcon 20 flights around Lampedusa. The AOD at 500 nm and IWV are mean and standard deviation of the MFRSR and HATPRO measurements, respectively, within the time duration of the ATR-42 flights. The flights were carried out under cloud-free conditions, as confirmed by the time series of surface solar and infrared radiation measurements and by the hemispheric pictures collected by the TSI-440 sky imager installed at the surface.

Aerosol microphysical and optical properties, thermodynamic state of the atmosphere, ozone concentration, downward and upward SW and LW irradiances are measured on-board the ATR-42. SW and LW radiation measurements are also available on-board the Falcon 20.

The aerosol SD at different altitudes in the diameter range 0.03-32 μm is obtained by combining measurements from the Ultra High Sensitivity Aerosol Spectrometer (UHSAS) in the 40-900 nm interval of the optical equivalent diameter, the Grimm 1.129 Optical Particle Counter for diameter sizes from 250 nm to 32 μm, and the Forward Scattering Spectrometer

Probe (FSSP) model 300 in the size range 0.28-20 μm (see Denjean et al., 2016 for details). The UHSAS and FSSP spectrometers measure SDs in ambient RH conditions, thus accounting for the possible particle growth due to water uptake. Denjean et al. (2016) used the measurements of the spectral scattering coefficient at 450, 550, and 770 nm and the derived scattering Ångström exponent calculated between 450 and 770 nm, together with the sub- and super-micron particles number concentrations, to show that dust particles were found in distinct layers, with variable concentrations of fine and coarse particles, mainly associated with airmass back-trajectories from Sahara (see discussion of Section 3.1.1).

The radiation instrumentation on-board the ATR-42 and the Falcon 20 includes Kipp&Zonen CMP22 pyranometers and CGR4 pyrgeometers. The upward- and downward-looking instruments are installed above and below the aircraft fuselage, respectively, and are aligned with the aircraft attitude, so that the instrument orientation is determined by the aircraft pitch, roll, and heading angle. The selection of the radiation measurements collected under horizontal position is described in Sect. 4.2. Instruments were calibrated in January 2013. On-board irradiance measurements have been corrected for the temperature dependence of the radiometer's sensitivity (Saunders et al., 1992). The estimated expanded uncertainties for both components of the instantaneous LW irradiances is ±6 W m$^{-2}$, obtained taking into account the accuracy of the instrument's calibration and of the acquisition system, and the consistency of airborne measurements.

The airborne version of the CLIMAT infrared radiometer (Legrand et al., 2000; Brogniez et al., 2003; 2005; Sourdeval et al., 2012) is installed below the fuselage to measure the nadir BT in three channels centred at 8.7, 10.6, and 12 μm with about 1 μm Full Width at Half Maximum (FWHM) bandwidth and about 3° FOV. While the contribution of the $O_3$ absorption band at 9.6 μm should not affect the CLIMAT measurements in the three bands, the 12 μm channel may be influenced by water vapour and $CO_2$. The absolute accuracy of BT measurements derived from CLIMAT is about 0.1 K. The instrument was calibrated at the Laboratoire d'Optique Atmosphérique in Febraury 2013.

### 2.3 Satellite observations

The Infrared Atmospheric Sounder Interferometer (IASI) radiance spectra simultaneous to ground-based measurements have been selected. IASI (Hilton et al., 2012) flies on the two EUMETSAT MetOp A and B polar platforms, with an equator crossing local solar time for the descending node around 9:30 and 8:45, respectively. The interferometer measures the spectral interval from 645 to 2760 cm$^{-1}$, with a sampling interval of 0.25 cm$^{-1}$ and an effective apodized resolution of 0.5 cm$^{-1}$, resulting in 8461 points per spectrum. The horizontal resolution at nadir is 12 km. The radiometric noise of IASI spectra is 0.123 K (in the band 780-980 cm$^{-1}$) and 0.137 K (in the band 1070-1200 cm$^{-1}$) (Serio et al., 2015).

Morning IASI spectra for the days of the campaign are selected in the region including Lampedusa, within the latitude/longitude range 35.0-36.0° N and 12.0-13.2° E. More details can be found in Liuzzi et al. (2017). Clear-sky FOVs are selected applying the Cumulative Discriminant Analysis approach (Amato et al., 2014); data contaminated by sun glint are removed.

### 3 Radiative transfer model

The MODTRAN release version 5.3 has been used to calculate spectral LW and WINDOW irradiances and BTs at the surface, at aircrafts altitudes, and up to the top of the atmosphere (TOA). The model band highest resolution is 0.2 cm$^{-1}$,

allowing to generate accurate spectral transmittances, radiances, and irradiances from 0 to 55000 cm$^{-1}$ (Berk et al., 2006; 2008; Anderson et al., 2009).

The DISORT (Stamnes et al., 1988) multiple scattering method with 8 streams with the fast and accurate correlated-k option is used for the simulations in the IR spectral region.

## 3.1 Model input parameters

Pressure, temperature, and relative humidity vertical profiles are obtained from the ATR-42 measurements. At altitudes above the aircraft flight, the radiosounding meteorological parameters and the AFGL standard midlatitude summer profile (Anderson et al., 1986) are used. The time corresponding to the radiosonde launch is reported in Table 1. AFGL profiles of the main absorbing gases ($O_3$, $CH_4$, $N_2O$, CO, $CO_2$,) are also adopted. The humidity profile is scaled according to the HATPRO IWV averaged over the flight portion, in order to take into account small differences in the radiosonde and the aircraft paths and time. Column ozone measured at the Station with a Brewer spectrophotometer is used to scale the vertical profile, while surface $CO_2$ mixing ratio measured with a Picarro G2401 analyser is input to translate the $CO_2$ AFGL profile. The model vertical resolution used is 0.1 km from the surface to the ATR-42 top altitude (5.8 km on 22 June, 5.4 km on 28 June, 4.8 km on 3 July), 0.25 km up to 10 km, 1 km up to the radiosounding maximum altitude, and larger than 1 km above. Radiosoundings were not launched on 22 June, so data from the European Centre for Medium-Range Weather Forecasts (ECMWF) operational analyses are used, with a spatial resolution of 0.125° x 0.125°. The ECMWF profile closest to the flight F35 (12 UT) is used on 22 June above the ATR-42 altitudes.

The vertical aerosol extinction profile is derived from the lidar backscatter profile and AOD measurements as described in Di Iorio et al. (2009) and Meloni et al. (2015). Figure 3 shows the lidar extinction profile at 532 nm for the three selected cases and the layers identified by the *in situ* aerosol measurements (Denjean et al., 2016). Mineral dust was not detected in the boundary layer, but was found to be stratified in different layers, each one characterized by uniform spectral scattering coefficients and number concentrations of sub- and super-micron particles, so that a homogeneous SD in each layer can be supposed. For example, during flight F35 the vertical profiles show very distinct dust layers above and below 3.5 km, the upper one containing dust from central Algeria and the lower one carrying dust from the southeastern Morocco-southwestern Algeria region. During flight F38 the scattering Ångström exponent above 0.5 and below 1.0 suggests that dust is mixed with pollution particles from the Mediterranean Sea.

The infrared emissivity of the sea water from Masuda et al. (1988) has been assumed in the model simulations. The sea emissivity in the 3.5-13 µm window region has been considered for a zenith view and for realistic wind speed. The value at 13 µm is applied also at longer wavelengths. The average surface wind speed is 6.7 m/s for F35, 7.3 m/s for F38, and 4 m/s for F42, and the sea emissivity corresponding to the wind speed value of 5 m/s has been adopted for all days. The sea emissivity has a maximum at 11 µm (0.993) and minima at 4.0 and 13.0 µm (0.977).

Daily MODIS sea surface temperature (SST) data at 1 km resolution (Feldman and McClain, 2014) produced and distributed by the NASA Goddard Space Flight Centre's Ocean Color Data Processing System (OCDPS) have been used.

### 3.1.1 Aerosol optical properties

The AOPs required by the model are spectral extinction, single scattering albedo, and phase function at each layer. They have been computed by applying the Mie theory for spherical particles to the SD obtained from AERONET and from *in situ*

observations (Denjean et al., 2016), with different values of the complex RIs. Figure 4 presents the comparison of AERONET and *in situ* SDs for the three case studies, normalized to 1 particle cm$^{-3}$. The AERONET SDs have been fitted with log-normal functions with three or four modes, depending on the case. The parameters of the log-normal SDs are reported in Table S1 in the Supplement. The *in situ* SDs within the dust layers (LAYER 3 for 22 June and LAYER2 for 3 July) are characterized by a larger number of particles in the largest mode (M4 in Table S1) compared with the AERONET SD. On 28 June the *in situ* SDs are characterized by lower radii compared to the AERONET SD. On all days the normalized number concentration corresponding to M4 mode increases with altitude. Dust particles from Algeria on 22 June (LAYER 3) and from Tunisia on 3 July (LAYER 2) present the largest median radius in M4 mode; however, the largest number concentration in found in the former day, thus having the dominant effect on the infrared radiation. On 28 June the median radii of the M4 mode on both layers are lower compared to those of the other two days, as expected when dust is mixed with pollution in the Mediterranean area. It must be emphasized that a direct comparison of the different SDs is problematic as the AERONET SD is column averaged and is an optically effective distribution essentially derived in the visible spectral range. Moreover, there are limitations in the capability of correctly reproducing the behaviour of the large particles mode, which is primarily important in the desert dust case (see e.g., the smaller accuracy for large particles, Dubovik et al., 2002a). On the contrary, the *in situ* SD is vertically resolved, and thus takes into account differences in aerosol types. On the other hand, small scale effects related to the aircraft position and flight pattern may affect the applicability of the retrieved SD to a somewhat larger scale. Notwithstanding this possible limitation, the vertically resolved SD derived from *in situ* observations is assumed to the one better representing the real atmospheric aerosol, and is used as a reference in the model calculations. Particles' sphericity is assumed in the analysis by Denjean et al. (2016), while the AERONET inversion algorithm that derives the aerosol SD includes spheroid-based parameterization of light scattering (Dubovik et al., 2002b).

No information about the infrared complex RI is available from direct measurements. The Optical Properties of Aerosol and Clouds (OPAC) database (Hess et al., 1998) is widely used as a reference of AOP in the solar and infrared spectral regions for the estimation of aerosol radiative effects (e.g. Gómez-Amo et al., 2014), as well as in the inversion of satellite observations (Klüser et al., 2012). The dust RI from Volz (1973) (Volz1973 from now on) is adopted in various climate models used to evaluate the global dust radiative effect (e.g. Balkanski et al., 2007; Miller et al., 2014).

Recent estimates of the complex refractive index of mineral dust in the infrared spectral region were obtained by Di Biagio et al. (2014a, 2014b; 2017) for different source regions of the world, including Sahara and Sahel deserts. In particular, Di Biagio et al. (2017) (from now on DB2017) retrieved the RI in the 2-16 μm interval for various samples of natural soil collected in Tunisia, Algeria, and Morocco, which are the most probable source regions of the dust particles detected during the ChArMex flights of this study. Indeed, by comparing air mass back trajectories and dust concentration maps, Denjean et al. (2016) have shown that dust particles were transported from these regions at different altitudes (for details see Table 1 of Denjean et al., 2016). More specifically, for flight F35 the dust layer above-3.5 km originated from southern Algeria, while the dust layer between 1.5 and 3.5 km was transported from southern Morocco. On 28 June (F38) dust came from Tunisia. On 3 July (F42) dust arriving at Lampedusa above 3 km altitude originated from Tunisia, while dust below 3 km altitude came from southern Morocco. For F35 the RI of Algerian dust from DB2017 is used in LAYER 3 and up to the top of the aerosol profile (see Figure 3), that of Moroccan dust in LAYER 2, and the OPAC water soluble RI for LAYER 1, i.e., below the dust layer, down to the surface. Similarly, the Tunisian dust RI from DB2017 is used for F38 flight in LAYER 2 and the OPAC water soluble one in LAYER 1. For F42 the Tunisian and the Moroccan dust RIs from DB2017 are used in

LAYER 2 (and to the top of the aerosol profile) and LAYER1, respectively, and the OPAC water soluble RI below LAYER 1.

The methodology applied in DB2017 allows to estimate the dust RI with an accuracy of about 20% and for conditions resembling the sandblasting process responsible for the generation of dust in the real environment. Moreover, DB2017 suggest that the LW RI does not change due to loss of coarse particles during the transport of dust from the source region, thus enabling its applicability also to transported dust.

The differences in the complex RI from OPAC, Volz1973, and the group of dust particles studied in DB2017 and used in this study can be found in Fig. S1 in the Supplement. Moreover, the spectral AOPs (extinction coefficients normalized to the value at 550 nm, single scattering albedoes, and asymmetry factors) computed using the combination of SDs and RIs for each layer of the three profiles have been shown in the Supplement (Figures S2, S3, and S4).

Table 2 summarizes the combination of SD (AERONET and *in situ*) and RI (OPAC, Volz1973, and DB2017) used to obtain the LW AOPs. The different combinations of SD and RI produce different values of the infrared AOD, reported also in Table 2 at 8.6 µm. The AOD at 8.6 µm is calculated from the MFRSR AOD at 500 nm and the ratio between the extinction coefficient at 8.6 µm and 500 nm obtained from the Mie calculations for each aerosol layer in Fig. 3. When the RIs from DB2017 and Volz1973 are used, which provide values only in the infrared spectral intervals, the aerosol RIs at 500 nm are obtained from OPAC: in particular the mineral dust RI is assumed for the dust layers and the water soluble one below. On F35 the AOD derived from the *in situ* SDs largely exceeds the AOD obtained with the AERONET SD (for example, by +151% for the OPAC RI). For a fixed SD, the AOD decreases when passing from OPAC to Volz1973 to DB2017 RI. The increase in AOD calculated with the *in situ* SD with respect to the AOD from AERONET SD is also evident for F38 and F42, corresponding to +126% and +81%, respectively.

### 3.2 Model outputs

The downward and upward broadband LW irradiances are calculated at the surface, at the different atmospheric levels, and at TOA for the different AOPs reported in Table 2. They are compared with the ATR-42, Falcon 20, and ground-based measurements.

The CGR3 downward WINDOW irradiance is simulated by integrating the spectral irradiance over the 8-14 µm interval.

The IRP and CLIMAT BTs are obtained integrating the modelled spectral BTs in the respective channel intervals, while IASI BT at the TOA are simulated in the 645-2760 cm$^{-1}$ (approximately 3.6-15.5 µm) spectral range. The CLIMAT BTs of each flight case have been computed at three altitudes, corresponding to a horizontal flying attitude of the ATR-42 and in layers characterized by different aerosol properties. The modelled BT profile shows a smooth change with height, so simulations at three altitudes only are sufficient to describe the vertical variations.

The aerosol direct radiative forcing and aerosol heating rates profiles are calculated after simulation of net (downward minus upward) irradiances and heating rates with and without aerosols for each model vertical layer, as in Meloni et al. (2015):

$$ARF(z) = NET_{AER}(z) - NET_{NOAER}(z) \tag{1}$$

$$AHR(z) = -\frac{g}{c_p}\left[\frac{\Delta NET(z)}{\Delta p(z)}_{AER} - \frac{\Delta NET(z)}{\Delta p(z)}_{NOAER}\right] \tag{2}$$

where $z$ is the altitude, *NET* is the net flux with and without aerosols, $p$ is the atmospheric pressure, $g$ the gravitational acceleration, $C_p$ is the specific heat of dry air at constant pressure, $\Delta NET$ and $\Delta p$ are vertical variations of NET and P, respectively.

The ARF of the atmosphere is defined as the difference of the ARF at TOA and at the surface.

ARF and AHR are estimated only for F35 case, which is characterized by the largest AOD at 500 nm, and thus by the largest LW radiative effects.

The aerosol dust forcing efficiency, ARFE, defined as the ARF per unit AOD at 500 nm, is also calculated at the surface, TOA, and in the atmosphere.

### 3.3 Uncertainty analysis

The sensitivity of the modelled infrared quantities to the main model input parameters has been investigated for aerosol-free and aerosol laden conditions with the aim of assessing the model uncertainty. The main model input parameters affecting infrared radiation in aerosol-free conditions have been considered (i.e., IWV, atmospheric temperature profile, SST, and surface emissivity). Each quantity has been perturbed one at a time by the amount of its uncertainty (+0.2 mm for IWV, +0.3 K for the temperature from the radiosonde, +0.4 K for the surface temperature). The total model uncertainty has been

calculated as the quadratic sum of uncertainties associated with the variation of each parameter. When considering the uncertainty associated with model simulations which include aerosol, the aerosol optical properties that have been varied are the AOD at 550 nm and the dust RI. Flight F35 has been selected as reference case, with INSU3 AOPs. The AOD at 550 nm has been increased by +0.02 while the dust imaginary part of the dust refractive index has been increased by 20%, whatever the dust origin (either Algeria or Morocco).

The model LW irradiance uncertainty decreases with increasing altitude for both the downward and upward components. The estimated model uncertainty on the downward and upward LW irradiances at the surface is 2.2 and 2.0 W m$^{-2}$ for simulations without and with aerosols, respectively. At the Falcon 20 altitude (about 10 km) the uncertainties are 0.6 and 1.5 W m$^{-2}$ for the downward and upward component, respectively, for both simulations with and without aerosol. The upward LW irradiance uncertainty is 1.4 W m$^{-2}$ at TOA, for simulations with and without aerosol. The estimated uncertainty on

ARF is obtained by the combination of the above values, and is 4.2 W m$^{-2}$ at the surface and 2.0 W m$^{-2}$ at the TOA. The uncertainty on AHR is largest at about 4.5 km altitude (0.030 K day$^{-1}$ with aerosol and 0.026 K day$^{-1}$ without aerosol), and close to the surface (0.050 K day$^{-1}$ with and without aerosol).

The uncertainty on the downward WINDOW irradiance is 0.9 and 0.6 W m$^{-2}$, with and without aerosol, respectively. The estimated uncertainty on the modelled zenith BT is 0.7 and 0.3 K, with and without aerosol, respectively.

The aerosol-free CLIMAT BT is much sensitive to SST and surface emission, with slightly larger values at 600 m (0.3 K) than at 5670 m (0.28 K). The overall uncertainty for the case with aerosol is 0.31 K at 600 m and 0.37 K at 5670 m.

The uncertainty on the spectral BT at TOA in the atmospheric window varies between 0.25 and 0.29 K in aerosol-free conditions, and between 0.32 and 0.50 K with aerosol.

## 4 Results and discussion

### 4.1 Surface irradiance and BT

Ground-based measurements of downward irradiance and BT have been averaged over a 10 minute interval around the time in which the ATR-42 reached the bottom atmospheric layers (see Table 1) or for the duration of the flight above Lampedusa on 3 July (14 minutes). The obtained LW and WINDOW irradiances and zenith sky BT in the 9.6-11.5 µm band are reported in Table 3. It is worth noting that the variability of the measured variables during the chosen time interval is much smaller than the measurement uncertainties. For example, on 22 June the standard deviation values are 0.2 W$m^{-2}$ for the LW irradiance, 0.3 W$m^{-2}$ for the WINDOW irradiance, and 0.1downward LW irradiances obtained in the aerosol-free simulations (NOAER) for the three days agree with measurements within their uncertainties: this is caused by the moderate AOD measured during the campaign and by the uncertainty associated with the measurement. The magnitude of the aerosol perturbation with respect to NOAER increases with the AOD. For the 22 June case (AOD at 500 nm of 0.36), the increase in LW irradiance at the surface due to dust is +4.8, 4.7, 3.3, 10.9, 10.6, and 8.1 W m$^{-2}$ with COL1, COL2, COL3, INSU1, INSU2, and INSU3, respectively. For the 28 June case (AOD at 500 nm of 0.21) the values are +3.2, 1.7, 6.3, 5.1 W m$^{-2}$ with COL1, COL3, INSU1, and INSU3.

The reasons for the largest (lower) perturbation of INSU1 (COL3) AOPs are twofold: the *in situ* SD of dust has generally a larger number or a larger median radius of the coarse fraction with respect to AERONET which implies a stronger infrared emission. On the other hand, the DB2017 RI is characterized by a smaller absorption than OPAC and Volz1973, which produces a smaller dust IR emission (see Table 2).

In all cases the AERONET SD is able to reproduce the downward LW irradiance with all RIs (COL1, COL2 and COL3). With the *in situ* SD all RIs produce larger irradiances than measurements on 22 June, while on 3 July the agreement with observations depends on the adopted dust RI: measurements are reproduced with DB2017 RI and overestimated with OPAC RI. On 28 June, the low AOD makes the model results coherent with measurements regardless of the aerosol properties, and the modelled quantities including aerosols are equivalent whatever the AOP.

The modelled WINDOW irradiance with aerosol is always overestimated, with the largest effect produced by the *in situ* SD. It is worth noting that the amount of the aerosol perturbation compared to the aerosol-free simulation is equal to that found for the LW irradiance for each AOP. As for the broadband LW, the WINDOW irradiance on 22 June can be reproduced with the AERONET SD: the RI yielding the largest (lowest) radiative effect is OPAC (DB2017). On 28 June the modelled WINDOW irradiance is within the measurement uncertainty for the NOAER and the COL3 AOPs, i.e. that providing the lowest radiative perturbation. On 3 July the modelled WINDOW irradiance is overestimated with all AOPs and measurements are reproduced only for the NOAER simulations. It should be pointed out that the irradiance in the 8-14 µm interval is very sensitive to the shape of the filter. A sensitive study has been performed reducing the spectral integration by 0.4 µm in different parts of the interval, i.e. considering the 8.2-13.8 µm, the 8.4-14 µm, and the 8-13.6 µm ranges. The INSU3 AOPs have been used for all cases. In all three cases the resulted WINDOW irradiances are considerably reduced (by 8.1, 7.1, and 9.3 W m$^{-2}$, respectively) with respect to the simulations in the 8-14 µm interval, and are close to the measured values within measurement uncertainty. These results show that for the atmospheric conditions met during the campaign the choice of the spectral interval for the CGR3 calibration may be critical. This aspect surely deserves a dedicated study, which is beyond the scope of the paper.

On 22 June the zenith BT estimated by the model with the AERONET SD is lower than observations, while is generally higher when calculations are made with the *in situ* SD. The combination of *in situ* SD and DB2017 refractive indices produces the best agreement with observations. A similar behaviour is found for 28 June.

The best agreement on 3 July for the zenith BT is obtained using the AERONET SD: the reason may be connected with the
absence of *in situ* information on the SD below 1600 m. The same SD as in the 1600-3500 m layer has been assumed in the lowest layers. This assumption may imply inaccurate model simulations using the *in situ* SD. However, as for the first two cases, the simulation with the *in situ* SD with DB2017 RI (INSU3) is sufficiently compliant with the measured LW irradiance and is close to the IRP BT.

Meloni et al. (2015) compared measured and modelled LW irradiances at the surface for a case with low (0.14 on 5 May
2008) and a case with high (0.59 on 3 May 2008) AOD at 500 nm during the GAMARF campaign. Similarly to this work, the simulations were made using SDs derived from AERONET and from *in situ* airborne measurements. For both cases the differences between LW irradiance measurement and model calculations are within measurement uncertainty either without aerosols, and accounting for the aerosol effect with the AERONET and the *in situ* SD. The results of the comparison shown in Table 3 are consistent with those by Meloni et al. (2015): in fact, congruity with measurements is obtained either
disregarding the aerosol presence, or including aerosols with the AERONET SD (whichever the RI) and with the *in situ* SD, but only on 28 June (all RI) and 3 July (DB2017 RI).

The final results of the analysis of the surface measurements show that irradiances, either broadband and in the 8-14 μm spectral interval, are not useful to reduce the uncertainty on the dust RI, since the impact of different RIs is below the measurement and model uncertainty. On the contrary, narrowband zenith BT seems to be suitable to constrain the dust RI
which better represents the dust optical properties either in moderate and in low dust loading conditions. These results apply also taking into account the model uncertainty discussed in Section 3.1.1. Under the assumption that the *in situ* SD is the most representative of the real aerosol dimensions, the DB2017 RI provides the best agreement between model and measurements, for LW and WINDOW irradiances and sky BT in the two cases where the atmospheric meteorological profiles and the *in situ* SD are measured down to the surface level (22 and 28 June).

## 4.2 Airborne irradiance and BT

### 4.2.1 Flight F35

F35 is the descending phase of the ATR-42 flight towards the Lampedusa airport on 22 June. From the flight top (5800 m)
down to 3700 m the flight path is above Lampedusa, then the ATR-42 moves East above the open sea while descending close to the surface, down to about 100 m above sea level (see Fig. 5a). Successively, it moves again towards the island before landing: this last part of the descent is not used in the analysis, and only data over the sea are considered.

The comparison of the measured and modelled irradiances has been restricted to the 100-5800 m altitude range. Figure 6 shows the measured LW irradiances, selected for pitch and roll angles lower than 1° and 1.5°, respectively, and the model
results without aerosol contribution and with all the AOPs. The average AOD during the descent is assumed as model input. The model-measurement comparison is provided in Table 4. The agreement is good for both downward and upward LW

irradiances with the AERONET SD, resulting in RMSDs of both components of 4.1 W m-2 with COL1 and COL2 and of 4.0 W m-2 with COL3. However the modelled upward component is not sensitive to different AOPs.

The NOAER downward LW irradiance profile agrees with observations throughout most of the altitude range, mainly due to the relatively low AOD value. Significant differences are found in the lowest atmospheric layer, where the model including aerosols better fits with observations. Similarly, the aerosol effect on the upward LW irradiance is apparent mainly in the upper altitude range. The aerosol contribution in the LW spectral region due to dust is an increase in the downward LW irradiance at lower altitudes and a decrease in the upward LW irradiances for increasing altitudes.

The upward and downward LW irradiances at the Falcon 20 altitude are also reproduced with all AOPs, as shown in Table 5. The downward LW irradiance at the Falcon 20 altitude is not affected by the aerosol AOPs, since the flight altitude is above the aerosol layer top (see Fig. 3). The upward irradiance calculated with the model is within the measurement uncertainties for both AERONET and *in situ* SD, and is sensitive to the different SDs, the model uncertainty at the Falcon 20 altitude being 1.5 W m$^{-2}$. The results confirm what found for the surface irradiance, i.e. that the broadband irradiance alone cannot help discriminating which SD and RI provide the best representation of the dust optical properties.

The CLIMAT BT profiles are plotted in Fig. 7. BT spikes are associated with ATR-42 passages over the island, where the surface temperature is larger than for the sea. The BT in the upper altitude range is less sensitive to SST, but more affected by the absorption and emission of the bottom layers, including aerosol. Moreover, while SST equally influences the BT at all CLIMAT channels, the spectral variation of the aerosol and atmospheric optical properties produces differences in BT above 4 km altitude. BTs calculated at selected altitudes (600, 3300, 5670 m), corresponding to a nearly horizontal attitude of the ATR-42, are also shown in Fig. 7. A good agreement at all altitudes is observed when aerosols are included, while an overestimation is observed at the highest altitude (5670 m) in the aerosol-free simulation, as also summarized by Table 6. Within the dust layer (LAYER 3 in Fig. 3) the occurrence of dust induces a significant decrease of BT with respect to the aerosol-free cases, with comparable effects for COL1, COL2 and INSU3 (Fig. 7b), and minor effects for COL3 (Fig. 7a).Conversely, the relevant BT reduction obtained with INSU1 and INSU2 cause a model underestimation compared to measurements (not shown)

Similarly to the upward LW irradiance, the largest CLIMAT BTs reduction with respect to aerosol-free conditions occurs above 4 km (see Fig. 7). The differences in BTs have been calculated for each channel with and without the aerosol contribution: the INSU1 AOPs produce the largest BT reduction at 5650 m (-1.7, -2.0, and -1.7 K at 8.7, 10.6, and 12 μm, respectively), while the lowest reduction is found for COL3 (-0.6, -0.7, and -0.2 K at 8.7, 10.6, and 12 μm, respectively). The perturbation caused by INSU3 AOPs are -1.1 K at 8.7 μm, -1.6 K at 10.6 μm, and -0.8 K at 12 μm: such differences are larger than the modelled BT uncertainty (0.37 K).

These results show the better sensitivity of BT to dust optical properties than broadband irradiance. When considering that *in situ* SD better represents the local aerosol distribution, the DB2017 RI from Algeria and Morocco provide the best model-measurement agreement.

The combination of SD and RI giving the best model-measurement match for the overall set of LW irradiances (downward at surface, upward and downward components on the ATR-42 and Falcon 20) have been evaluated by calculating the RMSD of all model-measurement absolute differences, and selecting only those AOPs for which the RMSD is below the ±5 W m$^{-2}$ threshold value. For the AERONET SD all the three RIs meet the requirement (RMSD between 3.2 and 3.3 W m$^{-2}$), while for the *in situ* SD only the DB2017 (RMSD 4.7 W m$^{-2}$).

For a given SD, the model results obtained with the OPAC and Volz1973 RIs are very similar. Thus, the model simulations for the other two profiles will be made only with the OPAC RI.

### 4.2.2 Flight F38

As for F35, the F38 profile ends at the Lampedusa airport on 28 June. The descent path to Lampedusa is rather complex (Fig. 5b), with a descending phase from 5420 m down to 3800 m, where the aircraft performed a large circle over the sea (with Lampedusa in the centre) at constant altitude, then continuing the descent. The ATR-42 path overpasses the island at different altitudes (between 600 and 900 m, and between 3500 and 5400 m), and the effect of land emissivity on the upward LW irradiance can be explored.

The AOD at 500 nm measured during the descent is the lowest of the group of three flights (Table 1). Model simulations coded as NOAER, COL1, COL3, INSU1, and INSU3 have been made for this profile.

The downward and upward LW irradiances are plotted in Fig. 8. Overall, the modelled downward LW irradiance follows very well the measured profile. Due to the low AOD, the modelled aerosol effect is small, similar to the irradiance measurement uncertainty. Conversely, a large upward LW irradiance variability is observed particularly below 2000 m: for example, the 10 W m$^{-2}$ increase occurring between 1500 and 1700 m cannot be reproduced by the model taking into account the measured atmospheric profiles, nor it can be attributed to the presence of thin clouds, since the humidity is very low (below 20%) in that layer. The ATR-42 flight path shows that at 1700 m the aircraft is close to the West coast of Lampedusa, and the downward-looking pyrgeometer is influenced by the island emission. Similarly, the upward LW irradiance profile shows altitude intervals with sensibly larger values (by about 16-18 W m$^{-2}$) than the local minima, as those around 900 m. The large irradiances correspond to ATR-42 passes over Lampedusa. Model simulations accounting for the island emissivity corroborate this hypothesis. Indeed, the red curve in Fig. 8 is obtained assuming that the surface is a mix of sea (50%) and land (50%), with their respective temperatures. The land temperature is derived from Level 2 MODIS Land Surface Temperature and Emissivity products with 1 km spatial resolution (MOD11_L2). The agreement with measurements at 900 m and at 1700 m confirms that the peak can be attributed to land contribution to the surface emission. A small peak is visible around 4100 m, suggesting that at this altitude the portion of land surface in the pyrgeometer's FOV is smaller than at lower altitudes.

In addition, differences between model and observations are present in the upward LW irradiance profile, in particular in the 2500-3200 m height range, where the model overestimates measurements by up to 14 W m$^{-2}$ around 2600 m. These differences cannot be explained with changes in the temperature and/or humidity profiles, that would have been captured by the airborne instrumentations. In addition, while the upward LW irradiance may be influenced by inhomogeneity in SST or surface emissivity, the SST shows very little variability within the region spanned by the ATR-42 (two sigma standard deviation less than 1 K in the entire area shown in Fig. 5b and less than 0.3 K in the surface area corresponding to the 2600-3200 m aircraft altitude range). The model-measurement disagreement in this height region may be related with the pyrgeometer response to rapid altitude (and consequently temperature) changes. The nominal CGR4 response time is about 6 seconds; however, a significantly longer time may be needed by an airborne instrument to attain the thermal equilibrium with ambient air, as reported by previous studies (e.g. Ehrlich and Wendisch, 2015 and references therein). Consequently, during fast descents the pyrgeometer's measurements may be negatively biased. The vertical velocities during the descent sections of the 22 and 28 June profiles were calculated and compared. The average descent rates in the altitude ranges where

model and measurements are in agreement are small: on 22 June between 3500 and 500 m is 2.8 m/s, and is 2.6 m/s on 28 June from 5400 to 4000 m. On the contrary, the average vertical velocity is 5.5 m/s from 3500 to 2000 m on 28 June. So the larger vertical velocities on 28 June might produce a possible underestimation of the upward LW irradiance.

The Falcon 20 path is described in Fig. 5b, while the model-measurement comparison is shown in Table 5. The best agreement between modelled and measured upward LW irradiance is obtained with the *in situ* SD. A good agreement is found also with the AERONET SD.

The CLIMAT BTs have been simulated at three different altitudes (700, 3000, and 5200 m) where the ATR-42 is horizontal (Fig. 9). The best agreement between model and measurement is found with the *in situ* SD with minor differences between OPAC and DB2017 RI (INSU1 and INSU3), and with COL1. This is confirmed in Table 6, where the RMSD is below the CLIMAT uncertainty for INSU1. However, the aerosol perturbation to the BT is not significant (even at the highest altitude) compared to the aerosol-free case, as shown in Fig. 9 (differences of -0.4, -0.6, -0.4 K at 8.7, 10.6, and 12 μm, respectively, with INSU3). A slight overestimation (0.17 K RMSD with INSU1) of the model is observed at 3000 m, in agreement with the results for the upward LW irradiance in Fig. 8. Assuming that the differences in BT between model and measurement could be attributed to the SST, their small amount is not sufficient to explain the disagreement in upward LW irradiance. It is important to underline that the CLIMAT response time is much lower (160 ms) than that of the pyrgeometer, so we do not expect that BT measurements are affected by the descent velocity of the ATR-42. BT simulations without aerosol show that the measured BTs spectral dependence, with nearly equidistant values, is well reproduced by the model. The spikes at 900 m and around 1700 m indicates that also CLIMAT captures the island emission.

### 4.2.3 Flight F42

During flight F42 on 3 July the ATR-42 arrived from north, performed a descent above Lampedusa, at altitudes between 4800 and 1600 m, and left southward (Fig. 5c). The short time period of the descent implies negligible AOD variations. As mentioned above, no *in situ* information on the aerosol properties below 1600 m is available.

Figure 10 shows the ATR-42 and the simulated LW irradiance profiles. The agreement between model and measurements for the downward component is good, except for the *in situ* SD and OPAC RI (INSU1), as also shown by the results in Table 4. The model overestimates the upward LW irradiance profile, except at 2000 m (see Fig. 5c), and around 4500 m, as also shown by Table 4. On the contrary, a good agreement with all AOPs is found when simulating the Falcon 20 irradiance. This may be a clue for possible inconsistencies in the description of the atmospheric thermodynamic state below the ATR-42 lower altitude, that affect more the simulations at lower altitudes and less those at higher altitudes. Nonetheless, the CLIMAT BTs (Fig. 11) are well reproduced by the model also at 3000 m, where the upward irradiance is overestimated. For these reasons the hypothesis of a bad representation of the thermodynamic profiles in the model is not acceptable. Additionally, non-homogeneities in SST are not evident in the area (two sigma standard deviation of 0.4 K on the whole area of Fig. 5c), which may explain the disagreement of the upward LW irradiance with the model simulations. As for the flight F38, the calculated average vertical velocity between 4800 and 1600 m is 5.3 m/s, thus the hypothesis of a LW irradiance underestimation due to the slow response time of the CGR4 pyrgeometer may be valid also for this case, and may affect both the upward and the downward components.

The results of Table 6 show that a good model-measurement agreement is found for the AERONET SD and the OPAC RI (COL1), and for the *in situ* SD and DB2017 RI (INSU3). This supports the conclusions of Section 4.2.1, that different

combinations of SD and RI may lead to the same values of BT. On the other hand, also in this case the DB2017 RIs appear to produce the best match with observations when the directly measured SD is used. However, according to the estimated uncertainty, the simulated BT including aerosol is not significantly different from that in aerosol-free conditions (differences of -0.3, -0.4, -0.3 K at 8.7, 10.6, and 12 μm, respectively, with INSU3).

As a conclusion of the simulation of airborne irradiance and BT for all the cases, it is worth remarking that irradiance is not sufficiently sensitive to the different AOPs, while the aerosol perturbation to the nadir infrared BT can be appreciated only at altitudes above the bulk aerosol emission, like for the flight F35.

### 4.3 IASI radiance simulations

The IASI spectra measured during the morning of 22 and 28 June and 3 July have been considered for the comparison with model simulations, with the aim of assessing the dust perturbation of the TOA infrared BT. IASI spectra are averaged in a region 1° latitude x 1.2° longitude centred at Lampedusa. The resulting standard deviation on the TOA spectral BT is around 0.6 K (0.2%) for 22 and 28 June, and 0.3 K (0.1%) for 3 July. A triangular slit function of 0.5 cm$^{-1}$ FWHM has been applied to the simulated spectra in order to reproduce the measured TOA radiances. The AOD at 500 nm and IWV model input have been sampled in a 30-minute interval centred at the time of the IASI observation (8:45 AM for 22 June, 9:06 AM for 28 June, and 9:03 AM for 3 July). Model-measurement differences are calculated in the 8-14 μm interval (714-1250 cm$^{-1}$), the most significant for evaluating the aerosol effect; the regions of the $O_3$ (980-1070 cm$^{-1}$) and $CO_2$ (below 780 cm$^{-1}$) absorption bands have been excluded and the comparison has been evaluated in the 780-980 cm$^{-1}$ and 1070-1200 cm$^{-1}$ intervals, as in Liuzzi et al. (2017), in terms of BT. The measured and simulated radiance spectra have been resampled at 15 cm$^{-1}$ to reduce the differences due to their respective spectral resolution.

The results of the comparison, presented as spectral differences in Fig. 12 and as RMSD in the two intervals in Table 7, show that differences between model and measurements are larger on 22 June than on 28 June and 3 July. The model shows that dust produces a decrease of the TOA BT with respect to aerosol-free conditions. On 22 June the *in situ* SD causes the larger effect compared to the AERONET SD: the maximum reduction of BT in the 780-980 cm$^{-1}$ interval is 1.3 K with COL2 and 2.6 K with INSU2, while in the 1070-1200 cm$^{-1}$ the aerosol-free TOA BT is reduced by 1.1 K with COL2 and by 2.3 K with INSU2. The reduction tends to cancel below 8.5 μm (i.e. above 1176 cm$^{-1}$) and above 12.5 μm (i.e. below 800 cm$^{-1}$) because water vapour and carbon dioxide absorption becomes dominant and hides the aerosol effects. The best match with the measured spectra is obtained with COL1 (INSU3) in the 780-980 cm$^{-1}$ (1070-1200 cm$^{-1}$) interval. On 28 June no significant perturbation is produced by the inclusion of the aerosol; the BT reduction is less than 0.8 K, the largest residual obtained with INSU1. The average residual in the 780-980 cm$^{-1}$ interval is lowest with COL3 AOPs, below the IASI radiometric noise (0.123 K). On 3 July the perturbation to the TOA aerosol-free BT reaches 1.1 K with INSU1 in both intervals, and the best match with the measured BTs between 780 and 980 cm$^{-1}$ is found with INSU1. The results in Table 7 suggest that, although the radiative closure at the TOA does not enable the univocal retrieval of the aerosol RI, the perturbation due to the aerosol effect is significant when the AOD is sufficiently large and that the inclusion of aerosol in the radiative transfer simulations produces a non-marginal improvement of the results.

Liuzzi et al. (2017) simulated the IASI TOA BT with the σ-IASI-as radiative transfer model using the *in situ* SD and two sets of dust RI, the one from Shettle and Fenn (1979) and those from DB2017. The AOPs of Liuzzi et al. (2017) using the *in*

*situ* SD and DB2017 RI are equivalent to INSU3 used here. Some differences in the choice of the model setup and input parameters exist with respect to the present study. First of all, the σ-IASI-as has a finer (0.01 cm$^{-1}$) resolution than MODTRAN5 (0.1 cm$^{-1}$). The limitation in the MODTRAN5 resolution does not allow to reproduce the high resolution IASI spectra: however, the scope of simulating the IASI measurements in this work is to show that TOA BTs are sensitive to the

dust occurrence and to its AOPs, and that they can be reproduced with the same input parameters that allow to simulate irradiance and BT at the surface and in the atmosphere. Liuzzi et al. (2017) results show that the best agreement is found with INSU3 for all cases in the 780-980 cm$^{-1}$ band, while Shettle and Fenn (1979) RI seems to perform better in the 1070-1200 cm$^{-1}$ spectral range on 28 June and 3 July.

The perturbation to the aerosol-free BT on 22 June with INSU3 AOP (2.3 K at 980 cm$^{-1}$ and 1.9 K at 1100 cm$^{-1}$) is

comparable to that found in Liuzzi et al. (2017). It must be highlighted that our results are obtained without tuning any parameter, by including in the simulations all the available (and significant) observed quantities, and by obtaining a good agreement also along the vertical profile, and at the surface. The good agreement between model and measurement is thus a quite robust result.

### 4.4 Radiative forcing and heating rate

The dust LW radiative forcing and heating rate profiles have been estimated for 22 June, which is the case with largest AOD, according to Eq. (1) and (2), with the AOPs that give the model-measurement match for the overall set of LW irradiances with RMSD below ±5 W m$^{-2}$, i.e. COL1, COL2, COL3, and INSU3. Moreover, the results obtained with INSU1 AOPs, i.e. those causing the largest aerosol radiative perturbation on LW irradiance, are presented to assess the radiative effect of different AOPs (Fig. 13).

ARF is positive at the surface and at TOA, which corresponds to a higher IR irradiance reaching the surface, and lower irradiance leaving the atmosphere with respect to aerosol-free conditions. A common feature of all profiles is the maximum radiative effect around 3800 m altitude, i.e. below the thickest dust layer (LAYER 3 in Fig. 3), as expected due to dust emission (e.g. Meloni et al., 2015). Very different ARF values are obtained depending on the AOPs, as shown in Table 8. The largest ARF is produced by INSU1, followed by INSU3 AOP. The lower DB2017 absorption is evident compared to

OPAC and Voltz RIs in the simulations with the AERONET SD. At the surface the ARF for INSU3 is 74% larger than for COL2, and 146% larger than COL3; the ARF for INSU1 is 33% larger than for INSU3. At TOA the ARF for INSU3 is about 71% larger than for COL2, and 172% larger than for COL3; the ARF for INSU1 is 47% larger than for INSU3. The large differences in ARF for COL1, COL2, COL3, and INSU3 show that different combination of SD and RI may all lead to reasonably simulated LW irradiances, but may also produce very different radiative effects. This behaviour underlines the

importance of determining accurate AOPs to correctly infer the aerosol radiative effects.

The aerosol dust forcing efficiency ARFE has been calculated at the surface, TOA, and in the atmosphere. This quantity is relevant because it gives an estimation of the aerosol radiative effect for large aerosol burden, i. e. for AOD at 500 nm equal to one. The estimated ARFE values are reported in Table 8. INSU1 AOPs produce the largest ARFE absolute values. Among the AOPs obtained with the measured SD and the RI giving agreement with measured irradiances, INSU3 produce

the largest ARFE absolute values at the surface, in the atmosphere and at TOA. The ARFE at the surface, +23.7 W m$^{-2}$, is comparable with the one determined at Lampedusa for the dust event occurring on 3 May 2008 during the GAMARF campaign (Meloni et al., 2015). The ARFE at TOA is nearly doubled with respect to that found during GAMARF, while the

atmospheric ARFE is lower. COL1, COL2, and COL3 AOPs provide similar ARFE values, although significantly smaller than for INSU3, at the surface, in the atmosphere and at TOA. The COL3 AOP produces the lowest ARFE at the surface and at TOA, due to the smaller absorption associated with the DB2017 RI compared to OPAC and Volz1973. If the ARF calculated with the moderate AOD of 22 June is not sensitive to the different AOPs according to the model uncertainties,

when the aerosol loading is large the ARF notably increases and the effect of the various AOPs becomes significant.

The AHR profile is negative within LAYER3, due to cooling caused by dust emission, and becomes positive just below the altitude of the maximum ARF. The layers below 200 m are locally heated by surface emission. Similarly to ARF, the local heating/cooling is very different when considering the various AOPs. For example, the maximum cooling reached at 4400 m is -0.28 K day$^{-1}$ with INSU1, -0.17 K day$^{-1}$ with INSU3, -0.13 K day$^{-1}$ with COL1, -0.12 K day$^{-1}$ with COL2, and -0.08 K

day$^{-1}$ with COL3. Larger differences are observed at the top of LAYER3, around 6200 m, where the AHR with INSU1 is - 0.20 K day$^{-1}$, with INSU3 is -0.14 K day$^{-1}$ and decreases to -0.06 K day$^{-1}$ with COL1 and COL2, and to -0.04 K day$^{-1}$ with COL3.

**5 Conclusions**

Three cases of Saharan dust occurring during the summer 2013 ChArMex/ADRIMED campaign have been selected to

perform a radiative closure study and to assess how different aerosol optical properties may affect the simulations of various radiation quantities in the infrared spectral interval. The average AOD at 500 nm for the selected cases were moderate: 0.36 on 22 June, 0.21 on 28 June, and 0.26 on 3 July.

Downward surface broadband LW irradiance, irradiance in the 8-14 µm range, and zenith sky BT in the 9.6-11.5 µm band were measured at the ENEA Station for Climate Observations in Lampedusa. Simultaneously, instruments on board the

SAFIRE ATR-42 and Falcon 20 aircrafts flying around the island collected vertical profiles of LW irradiance (upward and downward components) and nadir BT in three narrowband channels centred at 8.7, 10.6, and 12 µm. The IASI spectra closest in space and time to the flights have also been selected. In parallel, all parameters influencing the IR radiation were measured or, in some cases, inferred from satellite observations. These include in particular the atmospheric thermodynamic vertical profiles, and the aerosol size distribution.

The MODTRAN5.3 radiative transfer model has been initialized with all the available information. The sensitivity of the computed irradiances and BTs on different aerosol size distributions (columnar from AERONET and *in situ* from airborne observations) and infrared refractive indices (OPAC, Volz1973, and DB2017) has been investigated. Vertically resolved in situ measurements of the aerosol SD are used as reference because they provide the most detailed description of the atmospheric aerosol distribution. The computed IR AOD (at 8.6 µm) strongly depends on the size distribution: for example,

on 22 June the AOD with the *in situ* size distribution increases by a factor of 2.5-2.8 compared to the AOD with the AERONET size distribution, depending on the dust refractive index. The radiative closure has been carried out simultaneously using radiances and irradiances measured at the ground, airborne, at different atmospheric levels, and from space. This method constitutes a strong constraint on the aerosol properties and allows to identify key parameters also for low-to-moderate values of AOD, as is the case of this study.

The main results of the study can be summarized in the following points:

1. different combinations of SD and RI may produce similar LW and WINDOW irradiances; more specifically, column integrated SD may produce a good agreement between modelled and observed radiative fluxes with specific values of RI, which however do not necessarily correspond with the occurring value;

2. the integration of *in situ* radiation measurements at different levels in the atmosphere and in different IR bands helps in constraining the aerosol properties and the radiative effects; measurements of broadband LW irradiance are helpful for the determination of the dust properties and radiative effects, but alone do not permit reducing the uncertainty on the dust RI. In addition, the moderate AOD values measured during the campaign cause the measured irradiances to agree with simulations either in aerosol-free conditions and including aerosols with different AOPs, when uncertainties are taken into account;

3. knowledge of the dust SD, assumed to correspond with the *in situ* vertically resolved profiles, allows to constrain the RI, and consequently aerosol radiative forcing and heating rate. For the ADRIMED campaign, with dust particles originating from northern Algeria, Tunisia and Morocco, the dust RI from DB2017, derived for soil samples from the same regions, produces the best agreement with observations among the cases with the SD derived from *in situ* measurements ;

4. the use of inaccurate, although optically equivalent, size distributions and refractive indices, has a large impact on the ARF determination. The ARF may change by a factor as large as 2.5 at surface and 2.7 at TOA, depending on the SD and RI, for cases producing a good agreement with observed irradiances. Thus, the knowledge of the SD is crucial for a correct estimate of ARF;

5. the values of ARFE retrieved on 22 June are +23.7 W m$^{-2}$ at the surface, -7.9 W m$^{-2}$ in the atmosphere, and +15.8 W m$^{-2}$ at TOA. Significantly smaller ARFE are obtained when equivalent SDs and RIs are used;

6. similarly, the heating rate profile significantly depends on SD, RI, and vertical distribution;

7. knowledge of the whole vertical profile is important to obtain a good closure with respect to the radiative quantities; moreover, the dust vertical distribution appears to influence the derived ARF, primarily at TOA.

The dust radiative effects in the IR are thus non negligible, and their improper estimate may prevent a correct representation of processes acting on local and regional scales. Thus, it appears that specific efforts should be dedicated at determining reliable estimates of the dust vertical distribution and SD, and at implementing source-dependent RIs in regional and large scale models, with the aim of obtaining better determinations of the dust IR radiative effects at the regional scale.

## 6 Data availability

All the data used in this study are available upon request. Please contact the corresponding author.

The in situ distribution data from the ATR-42 aircraft are available on the ChArMEx database at http://mistrals.sedoo.fr/ChArMEx/. The user must register before having access to the data.

The authors declare that they have no conflict of interest.

**Acknowledgements**

This work has been supported by the Italian Ministry for University and Research project NextData. Measurements presented here are from the Chemistry-Aerosol Mediterranean Experiment project (ChArMEx, http://charmex.lsce.ipsl.fr), which is the atmospheric component of the French multidisciplinary program MISTRALS (Mediterranean Integrated Studies aT Regional And Local Scales). ChArMEx-France was principally funded by INSU, ADEME, ANR, CNES, CTC (Corsica region), EU/FEDER, Météo-France, and CEA.

The authors thank the Group of Atmospheric Optics, Valladolid University, for the provision of the CÆLIS tool (www.caelis.uva.es) used in this publication and the technicians, pilots and ground crew of SAFIRE (Service des Avions Français Instrumentés pour la Recherche en Environnement) for facilitating the instrument integration and conducting flying operations. We thank the AERONET, PHOTONS and RIMA staff for their support. The research leading to these results has received funding from the European Union's Horizon 2020 research and innovation programme under grant agreement No 654109.

ECMWF data used in this study have been obtained from the ECMWF Data Server.

The Level 2 MODIS Land Surface Temperature and Emissivity product was retrieved from the online Data Pool, courtesy of the NASA Land Processes Distributed Active Archive Center (LP DAAC), USGS/Earth Resources Observation and Science (EROS) Center, Sioux Falls, South Dakota, https://lpdaac.usgs.gov/data_access/data_pool.

The authors are grateful to Dr. Claudia Di Biagio for providing the dust infrared refractive indices of African soils and for the useful discussion about their use, and to Prof. Carmine Serio and Prof. Guido Masiello for providing the IASI spectra. Finally, thanks to the two reviewers' suggestions and questions the paper has improved and gained in readability.

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

Table 1. Summary of the time intervals, average (with standard deviation) AOD at 500 nm, IWV, and surface temperature during the descent portion of the three flights selected for this study. The altitude ranges of the ATR-42 and Falcon 20 observations are also shown.

| Day/Flight | Time (UT) | Radiosonde launch time (UT) | AOD at 500 nm | IWV (cm) | Surface T (K) | ATR-42 altitude range (m) | Falcon 20 altitude range (m) |
|---|---|---|---|---|---|---|---|
| 22/06 F35 | 10:23-11:18 | n. a. | 0.36±0.01 | 2.65±0.04 | 298.5±0.1 | 5800-15 | 10555-10565 |
| 28/06 F38 | 12:25-13:30 | 11:30 | 0.21±0.01 | 2.19±0.02 | 296.0±0.1 | 5420-15 | 9530-9560 |
| 03/07 F42 | 9:47-10:01 | 11:31 | 0.259±0.006 | 3.02±0.01 | 298.5±0.1 | 4830-1600 | 9280-9310 |

Table 2. Summary of the size distributions and complex refractive indices combinations used to calculate AOPs in the infrared spectral interval. The abbreviations for each combination and for aerosol-free conditions are reported in the first column. The column AOD at 8.6 µm is also shown. The upper table panel refers to the simulations for the flight F35, the middle panel for the flight F38, and the lower panel for the flight F42.

| Abbreviation | Aerosol layer | Size distribution (SD) | Refractive index (RI) | AOD at 8.6 µm |
|---|---|---|---|---|
| NOAER | | No aerosol | | |
| COL1 | 3600-5800 m | AERONET | OPAC mineral (Hess et al., 1998) | 0.058 |
| | 1400-3600 m | AERONET | OPAC mineral (Hess et al., 1998) | |
| | 0-1400 m | AERONET | OPAC water soluble (Hess et al., 1998) | |
| COL2 | 3600-5800 m | AERONET | Saharan dust (Volz, 1973) | 0.055 |
| | 1400-3600 m | AERONET | Saharan dust (Volz, 1973) | |
| | 0-1400 m | AERONET | OPAC water soluble (Hess et al., 1998) | |
| COL3 | 3600-5800 m | AERONET | Algeria dust (Di Biagio et al., 2017) | 0.040 |
| | 1400-3600 m | AERONET | Morocco dust (Di Biagio et al., 2017) | |
| | 0-1400 m | AERONET | OPAC water soluble (Hess et al., 1998) | |
| INSU1 | 3600-5800 m | *in situ* | OPAC mineral (Hess et al., 1998) | 0.146 |
| | 1400-3600 m | *in situ* | OPAC mineral (Hess et al., 1998) | |
| | 0-1400 m | *in situ* | OPAC water soluble (Hess et al., 1998) | |
| INSU2 | 3600-5800 m | *in situ* | Saharan dust (Volz, 1973) | 0.135 |
| | 1400-3600 m | *in situ* | Saharan dust (Volz, 1973) | |
| | 0-1400 m | *in situ* | OPAC water soluble (Hess et al., 1998) | |
| INSU3 | 3600-5800 m | *in situ* | Algeria dust (Di Biagio et al., 2017) | 0.110 |
| | 1400-3600 m | *in situ* | Morocco dust (Di Biagio et al., 2017) | |
| | 0-1400 m | *in situ* | OPAC water soluble (Hess et al., 1998) | |

| Abbreviation | Aerosol layer | Size distribution (SD) | Refractive index (RI) | AOD at 8.6 µm |
|---|---|---|---|---|
| NOAER | | No aerosol | | |
| COL1 | 1000-5400 m | AERONET | OPAC mineral (Hess et al., 1998) | 0.031 |
| | 0-1000 m | AERONET | OPAC water soluble (Hess et al., 1998) | |
| COL3 | 1000-5400 m | AERONET | Tunisia dust (Di Biagio et al., 2017) | 0.023 |
| | 0-1000 m | AERONET | OPAC water soluble (Hess et al., 1998) | |
| INSU1 | 1000-5400 m | *in situ* | OPAC mineral (Hess et al., 1998) | 0.070 |
| | 0-1000 m | *in situ* | OPAC water soluble (Hess et al., 1998) | |
| INSU3 | 1000-5400 m | *in situ* | Tunisia dust (Di Biagio et al., 2017) | 0.061 |
| | 0-1000 m | *in situ* | OPAC water soluble (Hess et al., 1998) | |

| Abbreviation | Aerosol layer | Size distribution (SD) | Refractive index (RI) | AOD at 8.6 µm |
|---|---|---|---|---|
| NOAER | | No aerosol | | |
| COL1 | 3500-4800 m | AERONET | OPAC mineral (Hess et al., 1998) | 0.043 |
| | 1600-3500 m | AERONET | OPAC mineral (Hess et al., 1998) | |
| | 0-1600 m | AERONET | OPAC water soluble (Hess et al., 1998) | |
| COL3 | 3500-4800 m | AERONET | Tunisia dust (Di Biagio et al., 2017) | 0.032 |
| | 1600-3500 m | AERONET | Morocco dust (Di Biagio et al., 2017) | |
| | 0-1600 m | AERONET | OPAC water soluble (Hess et al., 1998) | |
| INSU1 | 3500-4800 m | *in situ* | OPAC mineral (Hess et al., 1998) | 0.078 |
| | 1600-3500 m | *in situ* | OPAC mineral (Hess et al., 1998) | |
| | 0-1600 m | *in situ* | OPAC water soluble (Hess et al., 1998) | |
| INSU3 | 3500-4800 m | *in situ* | Tunisia dust (Di Biagio et al., 2017) | 0.064 |
| | 1600-3500 m | *in situ* | Morocco dust (Di Biagio et al., 2017) | |
| | 0-1600 m | *in situ* | OPAC water soluble (Hess et al., 1998) | |

Table 3. Measured and simulated downward LW and WINDOW irradiance and sky BT at the surface for the three analysed cases; model calculations are performed with different AOPs. The expanded uncertainty associated to the measurements is shown in parentheses together with the measured values. The model-measurement difference is shown in parentheses together with the modelled values. Differences within the measurement uncertainty are in bold.

| AOP | NOAER | COL1 | COL2 | COL3 | INSU1 | INSU2 | INSU3 | Measurement |
|---|---|---|---|---|---|---|---|---|
| *22 June* | | | | | | | | |
| LW (W m$^{-2}$) | 356.3 **(-2.3)** | 361.1 **(+2.5)** | 361.0 **(+2.4)** | 359.6 **(+1.0)** | 367.2 (+8.6) | 366.9 (+8.3) | 364.4 (+5.8) | 358.6 (±5) |
| WINDOW (W m$^{-2}$) | 82.5 (-3.4) | 87.3 **(+1.4)** | 87.2 **(+1.3)** | 85.8 **(-0.1)** | 93.4 (+7.5) | 93.1 (+7.2) | 90.5 (+4.6) | 85.9 (±2) |
| IR BT (K) | 224.2 (-7.7) | 228.4 (-3.5) | 229.1 (-2.8) | 227.5 (-4.4) | 234.2 (+2.3) | 234.6 (+2.7) | 232.5 **(+0.6)** | 231.9 (±1) |
| *28 June* | | | | | | | | |
| LW (W m$^{-2}$) | 335.2 **(-2.7)** | 338.6 **(+0.5)** | | 336.9 **(-1.0)** | 341.5 **(+3.6)** | | 339.3 **(+2.4)** | 337.9 (±5) |
| WINDOW (W m$^{-2}$) | 73.4 **(-0.5)** | 76.7 (+2.8) | | 75.1 **(+1.2)** | 79.5 (+5.6) | | 77.4 (+3.5) | 73.9 (±2) |
| IR BT (K) | 219.9 (-4.9) | 223.0 (-1.8) | | 221.6 (-3.2) | 225.9 (+1.1) | | 224.2 **(-0.6)** | 224.8 (±1) |
| *3 July* | | | | | | | | |
| LW (W m$^{-2}$) | 362.4 **(-1.3)** | 367.4 **(+3.7)** | | 365.6 **(+1.9)** | 370.8 (+7.1) | | 368.0 **(+4.3)** | 363.7 (±5) |
| WINDOW (W m$^{-2}$) | 92.0 **(+1.4)** | 96.9 (+6.3) | | 95.1 (+4.5) | 100.1 (+9.5) | | 97.4 (+6.8) | 90.6 ( ±2) |
| IR BT (K) | 233.6 (-3.4) | 237.4 **(+0.4)** | | 236.1 **(-0.9)** | 240.5 (+3.5) | | 238.2 (+1.2) | 237.0 (±1) |

Table 4. RMSD in W m$^{-2}$ between modelled and measured downward and upward LW irradiances for the ATR-42 profiles in the three analysed cases. Model calculations are made with different AOPs. The measurement uncertainty is ±6 W m$^{-2}$. Values within the measurement uncertainty are in bold.

| AOP | NOAER | COL1 | COL2 | COL3 | INSU1 | INSU2 | INSU3 |
|------|-------|------|------|------|-------|-------|-------|
| | | | *22 June* | | | | |
| LW↓ | **3.4** | **3.2** | **3.0** | **2.2** | 9.3 | 8.8 | **5.6** |
| LW↑ | **5.9** | **4.9** | **4.9** | **5.2** | **3.9** | **4.0** | **4.4** |
| | | | *28 June* | | | | |
| LW↓ | **3.8** | **3.2** | | **3.5** | **3.4** | | **3.2** |
| LW↑ | 7.7 | 7.2 | | 7.4 | 6.8 | | 7.0 |
| | | | *3 July* | | | | |
| LW↓ | **4.0** | **5.9** | | **4.8** | 7.2 | | **5.6** |
| LW↑ | 10.9 | 9.2 | | 9.6 | 8.6 | | 9.1 |

Table 5. Downward and upward LW irradiances measured on-board the Falcon 20 and modelled for on the three analysed cases. Model calculations are made with different AOPs. The measurements uncertainty is ±6 W m$^{-2}$. Differences with respect to measurements are shown in parentheses and are in bold if lower than the measurement uncertainty.

| AOP | NOAER | COL1 | COL2 | COL3 | INSU1 | INSU2 | INSU3 | Measurement |
|---|---|---|---|---|---|---|---|---|
| | | | | *22 June* | | | | |
| LW↓ | 41.0 **(-1.7)** | 41.0 **(-1.7)** | 41.0 **(-1.7)** | 41.0 **(-1.7)** | 41.0 **(-1.7)** | 41.0 **(-1.7)** | 41.0 **(-1.7)** | 42.7 |
| LW↑ | 298.0 **(+4.8)** | 294.1 **(+0.9)** | 294.3 **(+1.1)** | 295.6 **(+2.4)** | 288.9 **(-4.3)** | 289.2 **(-4.0)** | 291.7 **(-1.5)** | 293.2 |
| | | | | *28 June* | | | | |
| LW↓ | 51.2 **(-5.0)** | 51.2 **(-5.0)** | | 51.2 **(-5.0)** | 51.2 **(-5.0)** | | 51.**2** **(-5.0)** | 56.2 |
| LW↑ | 300.0 (+6.9) | 298.6 **(+5.5)** | | 299.2 (+6.1) | 297.2 **(+4.1)** | | 298.0 **(+4.9)** | 293.1 |
| | | | | *3 July* | | | | |
| LW↓ | 57.0 **(+0.2)** | 57.0 **(+0.2)** | | 57.0 **(+0.2)** | 57.0 **(+0.2)** | | 57.0 **(+0.2)** | 56.8 |
| LW↑ | 306.6 **(+4.7)** | 304.9 **(+3.0)** | | 305.7 **(+3.8)** | 303.6 **(+1.7)** | | 304.8 **(+2.9)** | 301.9 |

Table 6. Comparison of the measured and modelled CLIMAT BTs. Model calculations are made with different AOPs. Differences are expressed as RMSD (K) from data at three different altitudes and for the three CLIMAT channels (see text). The RMSD at the maximum altitude is shown in parentheses.

| AOP | NOAER | COL1 | COL2 | COL3 | INSU1 | INSU2 | INSU3 |
|---|---|---|---|---|---|---|---|
| 22 June | 0.36 (0.61) | 0.10 (0.15) | 0.11 (0.17) | 0.19 (0.32) | 0.27 (0.47) | 0.25 (0.43) | 0.10 (0.16) |
| 28 June | 0.27 (0.36) | 0.14 (0.15) | | 0.21 (0.27) | 0.09 (0.09) | | 0.12 (0.11) |
| 3 July | 0.22 (0.23) | 0.17 (0.02) | | 0.19 (0.14) | 0.21 (0.16) | | 0.18 (0.03) |

Table 7. Differences (K) between modelled and measured IASI BT spectra in the 780-980 $cm^{-1}$ and 1070-1200 $cm^{-1}$ spectral intervals of the atmospheric window. Model calculations with different AOPs are shown. Differences are expressed as RMSD and standard deviation. In bold the significant differences with respect to the NOAER simulations.

| AOP | NOAER | COL1 | COL2 | COL3 | INSU1 | INSU2 | INSU3 |
|---|---|---|---|---|---|---|---|
| *22 June* | | | | | | | |
| 780-980 $cm^{-1}$ | 1.0±0.3 | **0.4±0.3** | 0.5±0.5 | 0.7±0.3 | 1.1±0.4 | 1.5±0.8 | 0.8±0.6 |
| 1070-1200 $cm^{-1}$ | 1.3±0.3 | **0.6±0.1** | **0.6±0.2** | **0.8±0.1** | **0.4±0.3** | 0.8±0.7 | **0.2±0.2** |
| *28 June* | | | | | | | |
| 780-980 $cm^{-1}$ | 0.2±0.1 | 0.2±0.1 | | 0.1±0.1 | 0.5±0.1 | | 0.3±0.1 |
| 1070-1200 $cm^{-1}$ | 0.5±0.2 | 0.3±0.2 | | 0.4±0.2 | 0.2±0.2 | | 0.2±0.2 |
| *3 July* | | | | | | | |
| 780-980 $cm^{-1}$ | 0.8±0.1 | **0.4±0.2** | | 0.6±0.2 | **0.2±0.2** | | **0.4±0.2** |
| 1070-1200 $cm^{-1}$ | 0.9±0.1 | **0.6±0.2** | | 0.7±0.2 | 0.5±0.4 | | **0.6±0.2** |

Table 8. LW ARF and ARFE at the surface, TOA, and in the atmosphere (in W m$^{-2}$) on 22 June calculated with the AOPs which produce the best agreement with respect to the irradiance profiles (COL1, COL2, COL3, and INSU3) and with INSU1 AOP in addition.

| AOP | ARF | | | | | ARFE | | | | |
|---|---|---|---|---|---|---|---|---|---|---|
| | COL1 | COL2 | COL3 | INSU1 | INSU3 | COL1 | COL2 | COL3 | INSU1 | INSU3 |
| Surface | 5.0 | 4.9 | 3.5 | 11.3 | 8.5 | 13.9 | 13.6 | 9.6 | 31.5 | 23.7 |
| TOA | 3.5 | 3.3 | 2.1 | 8.3 | 5.7 | 9.7 | 9.2 | 5.8 | 23.1 | 15.8 |
| Atmosphere | -1.5 | -1.6 | -1.4 | -3.0 | -2.8 | -5.8 | -5.6 | -3.8 | -8.4 | -7.9 |

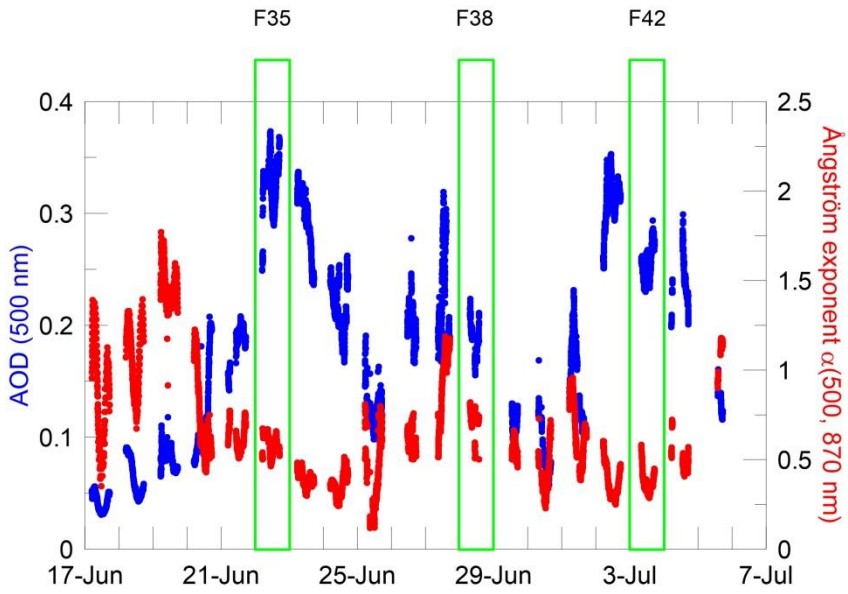

**Figure 1: AOD at 500 nm (blue dots) and Ångström exponent (red dots) measured by the MFRSR during the campaign.**

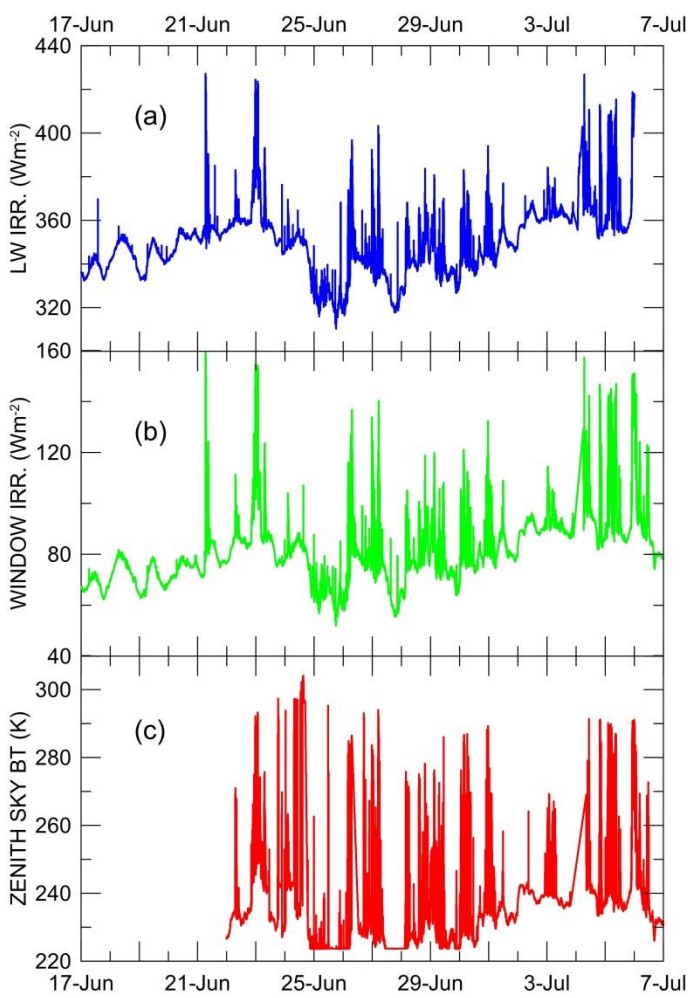

**Figure 2: Time evolution of (a) the downward broadband LW irradiance, (b) the WINDOW irradiance, and (c) the zenith sky BT during the ADRIMED campaign.**

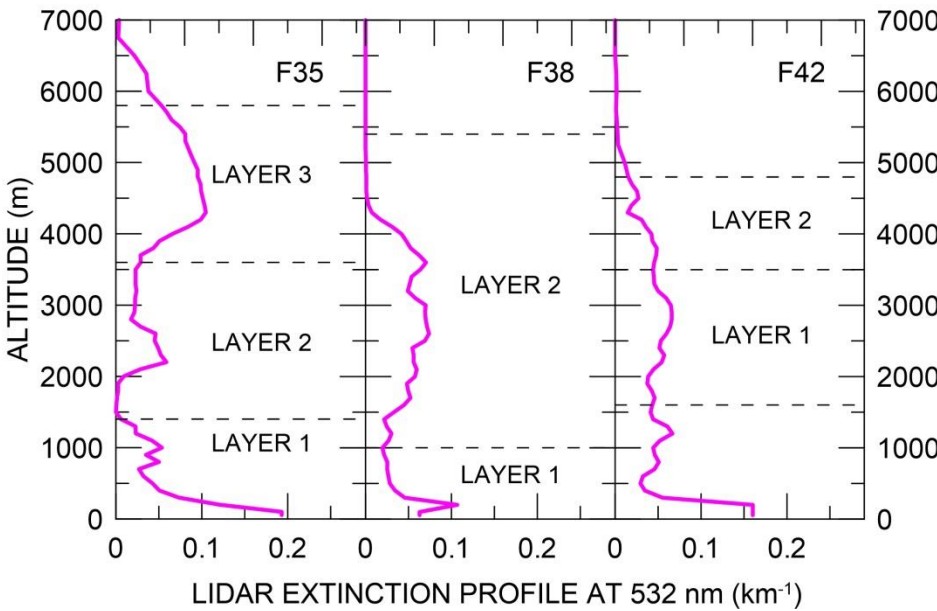

**Figure 3: Lidar extinction profiles at 532 nm averaged during the three ATR-42 descents. From left to right, on 22 and 28 June, and on 3 July. The aerosol layers identified by different optical properties from the *in situ* airborne measurements are also evidenced. Note that on 3 July the *in situ* profiles end at the bottom of layer 1.**

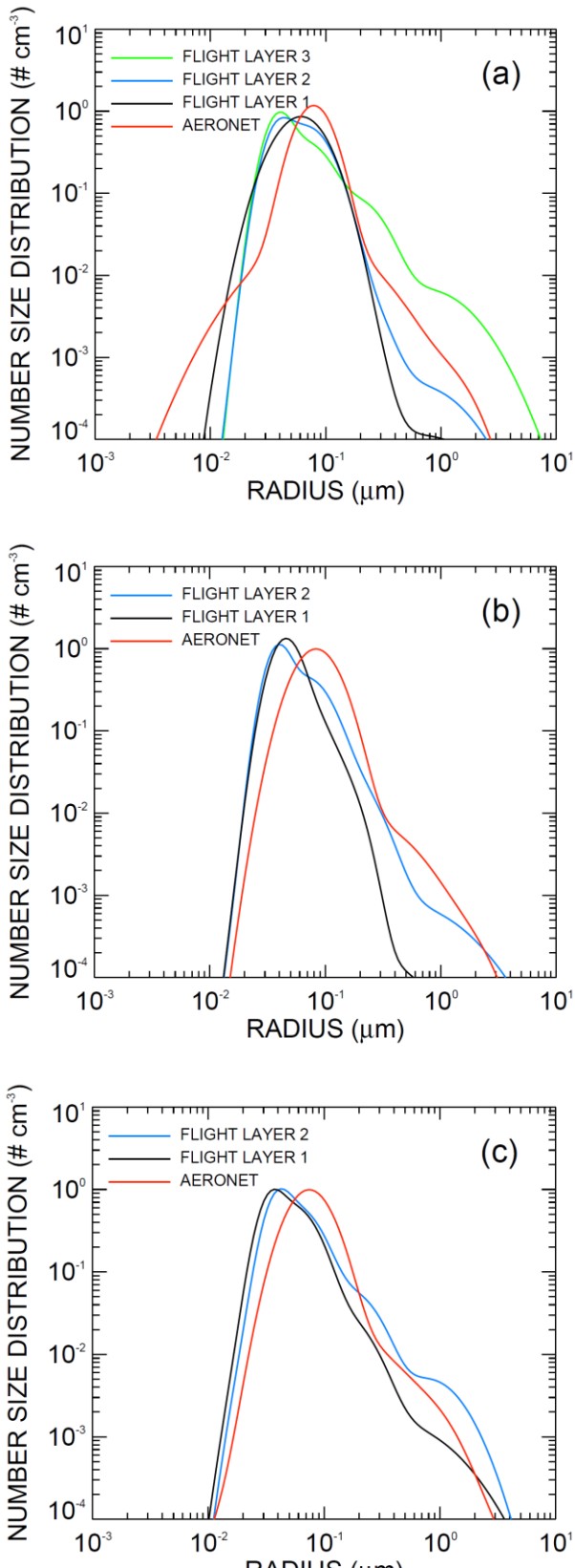

**Figure 4: Normalized aerosol number size distribution derived from AERONET measurements (red curve) and from the airborne in situ measurements for the layers of figure 3 for (a) F35, (b) F38, and (c) F42.**

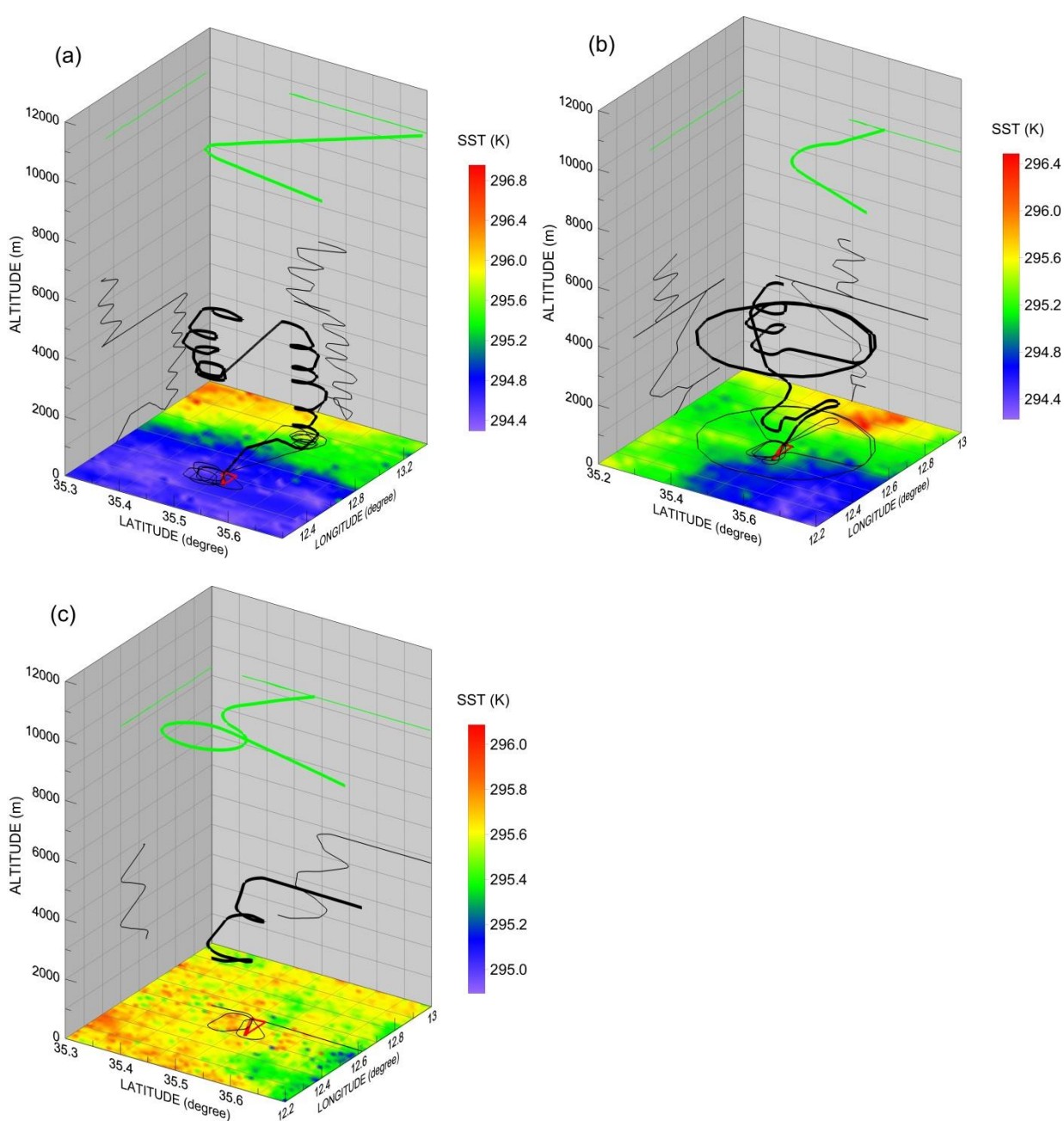

**Figure 5: ATR-42 (black curve) and Falcon 20 (green curve) flight paths on (a) 22 June (F35), (b) 28 June (F38), and (c) 3 July (F42), and projections on horizontal and vertical planes. The surface map of the SST from MODIS is shown with the colour scale. A schematic map of Lampedusa contour is shown in red. Please, note that the latitude-longitude and the SST scales are different in each graph.**

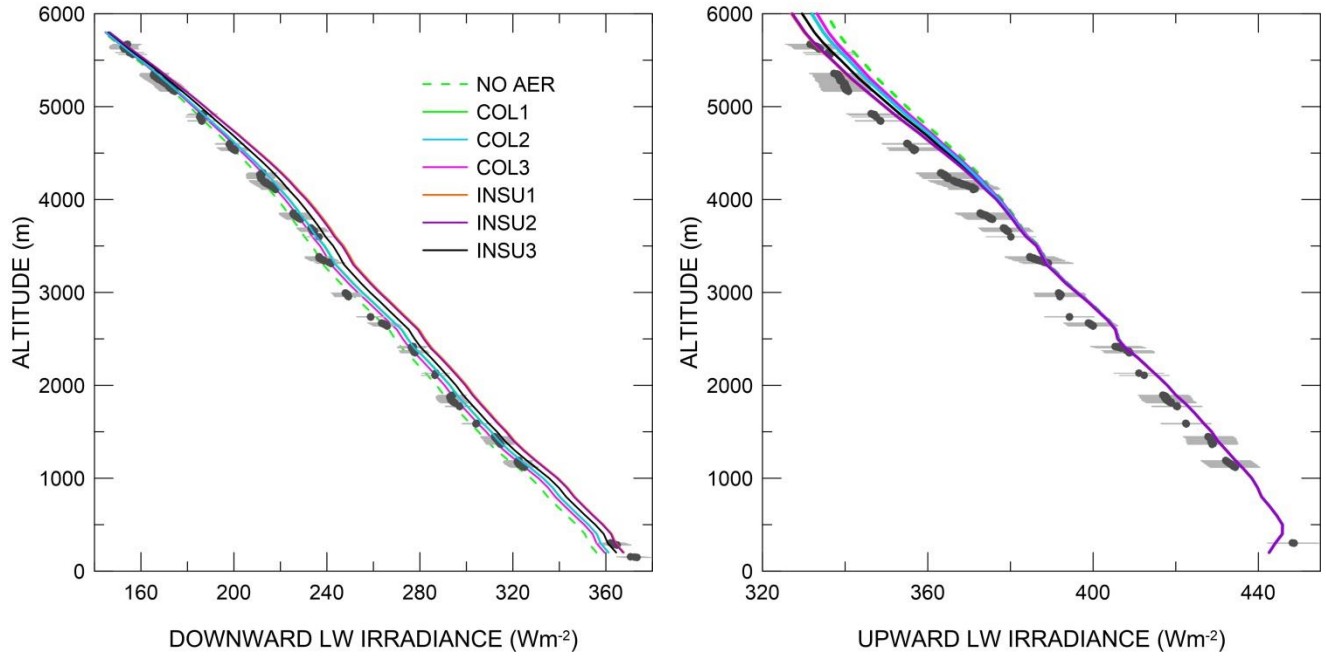

**Figure 6: Downward (left) and upward (right) LW irradiance profiles for F35. Grey dots are the measurements selected for small pitch and roll angles, with the associated uncertainty (one sigma) in light grey. Simulated profiles with different SDs and RIs are shown in different colours. See text for details.**

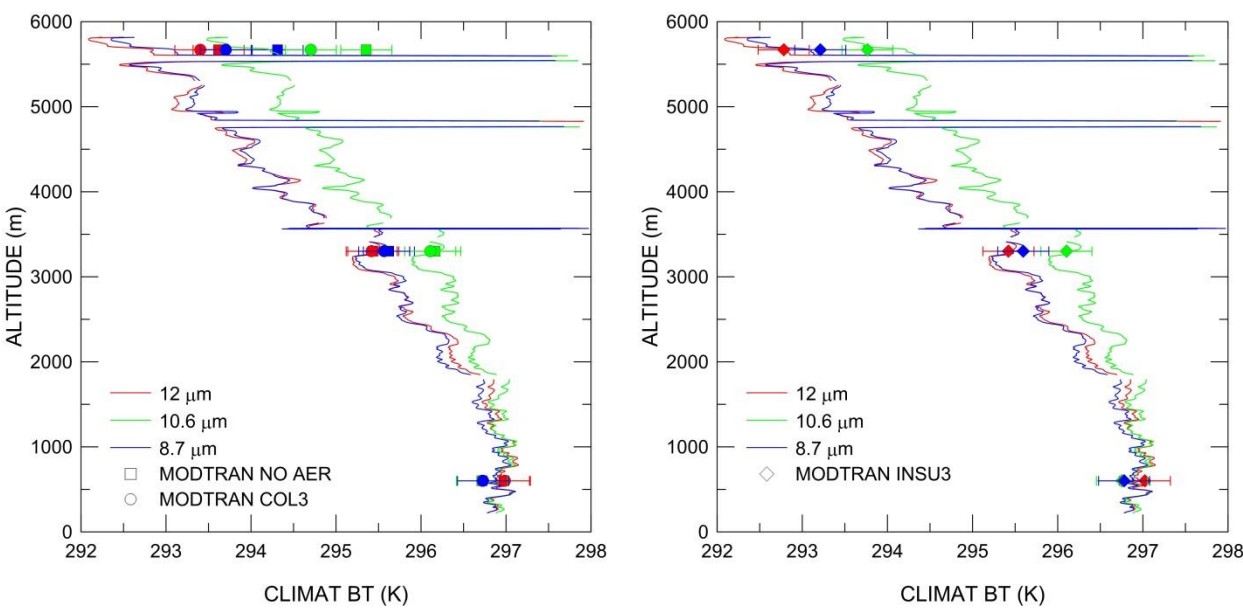

**Figure 7: CLIMAT BTs vertical profiles and simulated BTs at selected altitudes during F35 for aerosol-free conditions and with different AOPs: Algerian and Moroccan dust RIs with AERONET left) and *in situ* (right) SD.**

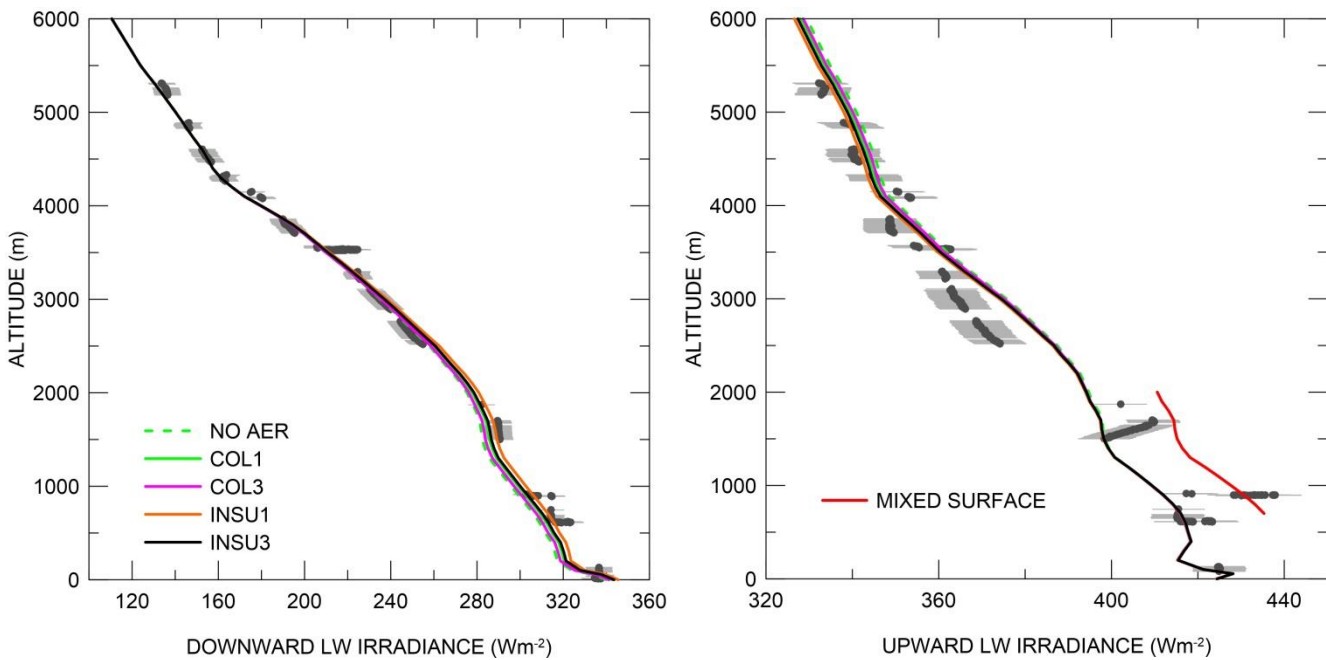

**Figure 8: Same as Figure 6, but for profile F38. The red curve in the upward LW irradiance plot is obtained using Lampedusa surface temperature and emissivity. See text for details.**

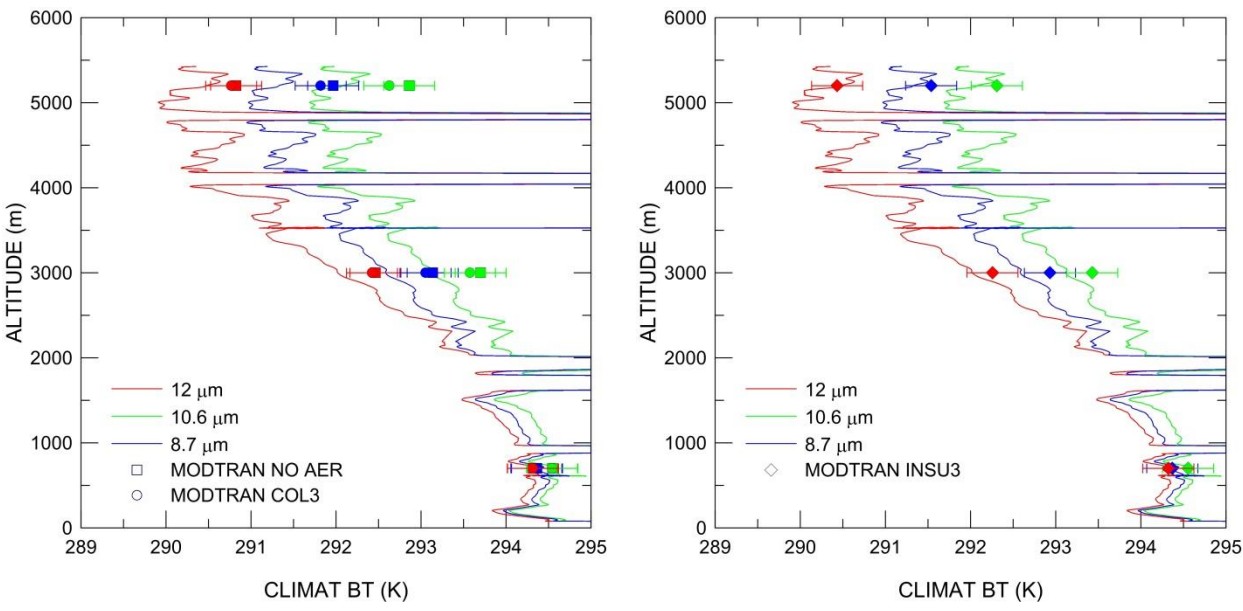

**Figure 9: Same as Figure 7, but for profile F38. The simulations are made for *in situ* SD and Tunisian dust RI in the right plot.**

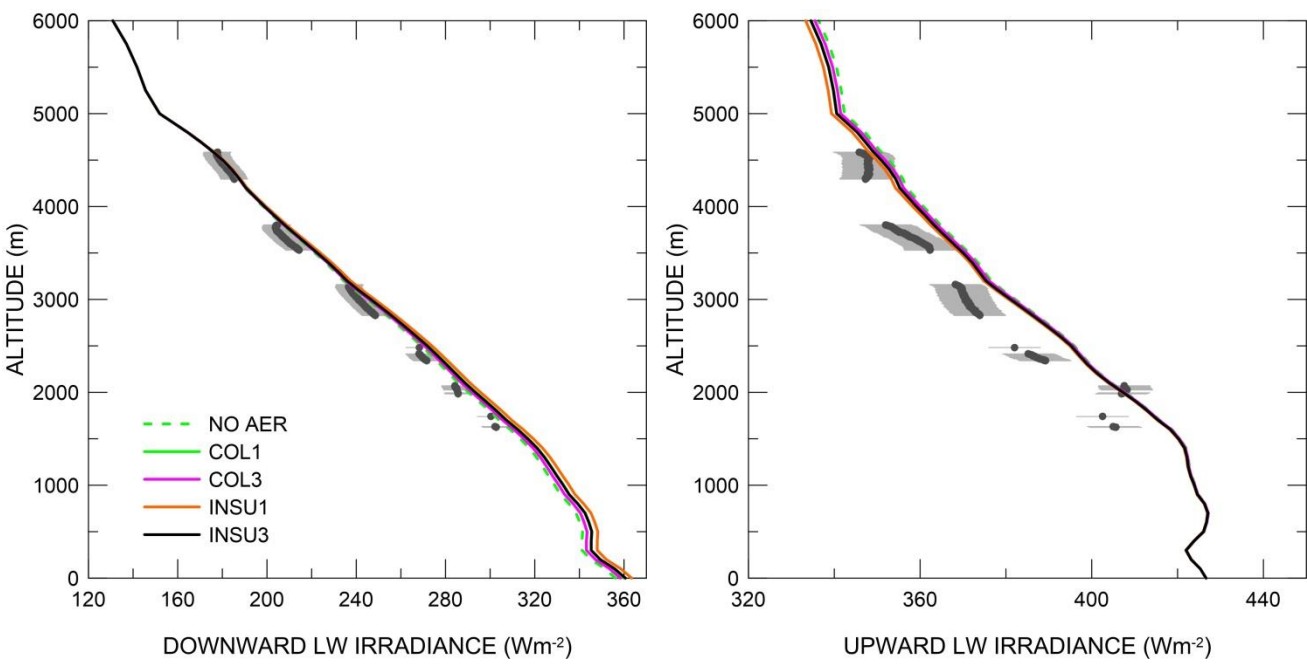

**Figure 10: Same as Figure 6, but for profile F42.**

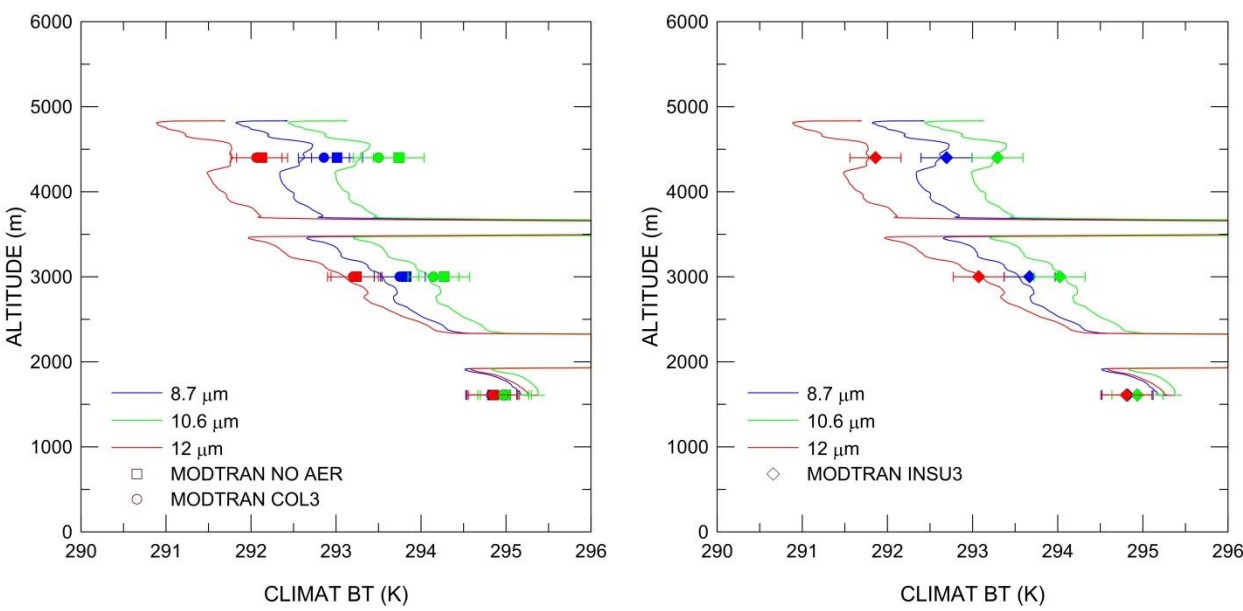

**Figure 11: Same as Figure 9, but for profile F42. Simulated data are for the *in situ* SD and Tunisian and Moroccan dust RIs in the right panel.**

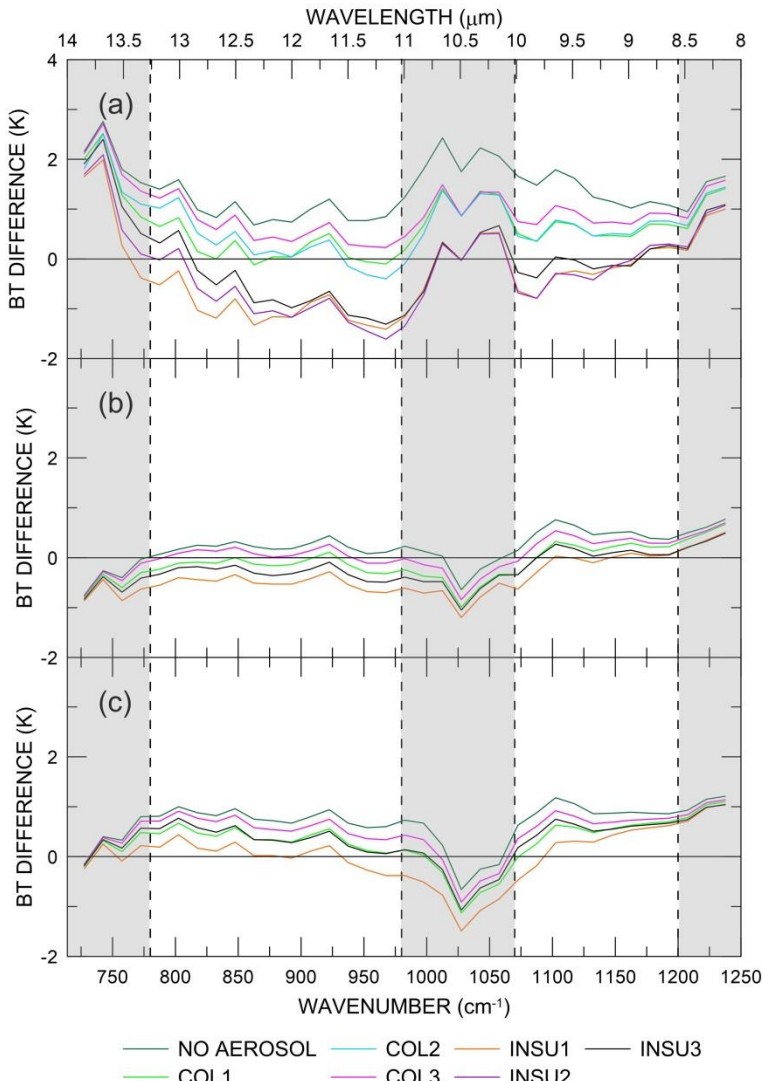

**Figure 12: Differences between modelled and measured IASI BT spectra at TOA for (a) 22 June, (b) 28 June, and (c) 3 July. Simulations have been performed with different AOPs and sampled at 15 cm$^{-1}$ intervals. The vertical black dashed lines delimit the two spectral intervals (780-980 cm$^{-1}$ and 1070-1200 cm$^{-1}$) where the differences are discussed, while the shaded area are not considered in the analysis.**

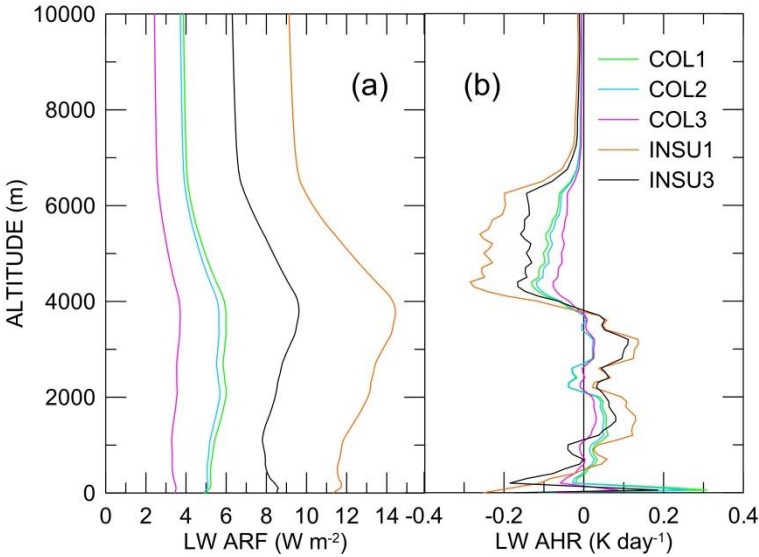

**Figure 13: Dust LW radiative forcing (a) and heating rate (b) on 22 June calculated with different AOPs.**