# Peer review of "Determining the infrared radiative effects of Saharan dust: a radiative transfer modelling study based on vertically resolved measurements at Lampedusa"

_Atmospheric Chemistry and Physics, 2017_

## Referee Comment (RC1) · Anonymous Referee #1 · 21 Sep 2017

General comments: This paper lies in the framework of the ChArMEx/ADRIMED experiment, that took place in the Mediterranean in summer 2013. Three vertical profiles of atmospheric and aerosol properties, made at Lampedusa in conjunction with surface, airborne and satellite IR broadband and narrowband radiation as well as radiosonde are analyzed in order to 1) identify the sensitivity of the different radiative measurements to mineral dust microphysical properties (size distribution and refractive index) and 2) analyze their impact in term of radiative forcing. The main result of this study is that if LW irradiance is poorly sensitive to aerosol microphysical properties compared

to brightness temperature, the IR dust radiative forcing is non-negligible, and strongly depends on size distribution (SD) and refractive index (RI). This study highlights the importance of a precise knowledge of the dust microphysics to infer correctly their radiative effect. The paper is an interesting sensitivity study of the radiative variables to the aerosol microphysics, leading, in particular, to the conclusion that spectrally resolved measurements of brightness temperature is more fitted to infer dust properties than broadband LW irradiances. However, the part that concludes on the most appropriate refractive indices is less convincing. Such a study would require a more detailed analysis of the differences between refractive indices (at minimum a figure displaying their values in the spectral domain concerned), as well as a more exhaustive variability in the choice of the indices. Here among the three indices used, two indices are quite similar and only one coming from recent measurement campaign of DiBiagio et al., 2017 is really different. Moreover, the study, based principally on RT simulations, lacks of discussions on the uncertainties due to the RT model itself, as well as to the different hypothesis used. In particular, which is the impact of an error in surface temperature or surface emissivity? An error on the water vapor profile? No reference error is calculated under clear sky condition for example, to distinguish error directly due to the model from errors due to the impact of aerosols properties. The resulting biases obtained with the different aerosol properties configurations cannot therefore be really discussed. For example, large biases between simulated and calculated irradiances are not explained (and apparently not due to wrong aerosol properties), implying that something is missing in the RT model, but not enough discussed. The section on IASI data is not enough developed. All the spectra within a box of about 100 kmx100 km are averaged before analysis, causing a standard deviation in the averaged spectrum larger than the effect of the aerosols properties analyzed! Here again, since no reference errors are given, biases obtained from the different aerosol configurations are finally equivalent and it's not possible to state on the best configuration. This part doesn't really bring new information compared to the previous sections or previous studies, or required a more precise development. Finally, some details on the inputs

used are missing. A few details are provided on the size parameters used (the reader has to refer to the paper of Denjean et al. 2016 to have the precisions) and the exact refractive index from DiBiagio et al., 2017 use in this study is not given: 3 different indices are coming from ń the source regions (Tunisia, Algeria, Morocco) in DiBiagio et al., 2017 whereas only one is used here without any precision!

Specific comments: - p2line 21 ă: ń ăMost of these studies have been carried out close to the dust source regions and did not take into account the possible modifications in dust optical properties during long range transport. ": You can also find some studies dealing with the variation of the aerosol properties with transport (e.g. Maring 2003; Ryder et al., 2013; Weinzierl et al., 2017 and so on). Moreover, I don't see the link with the subject of this paper since there is no discussion on the possible change of dust properties with transport. Maring, H.: Vertical distributions of dust and sea-salt aerosols over Puerto Rico during PRIDE measured from a light aircraft, J. Geophys. Res., 108(D19), 1–11, doi:10.1029/2002JD002544, 2003. Ryder, C. L., Highwood, E. J., Lai, T. M., Sodemann, H. and Marsham, J. H.: Impact of atmospheric transport on the evolution of microphysical and optical properties of Saharan dust, Geophys. Res. Lett., 40(10), 2433–2438, doi:10.1002/grl.50482, 2013. Weinzierl, B., Pros-Pero, J., Chouza, F., FomBa, W., Freudenthaler, V., GasteiGer, J. and Toledano, C.: THE SAHARAN AEROSOL LONG-RANGE TRANSPORT AND AEROSOL–CLOUD-INTERACTION EXPERIMENT Overview and Selected Highlights, [online] Available from: http://journals.ametsoc.org/doi/pdf/10.1175/BAMS-D-15-00142.1 - P2line 29: the paper Sellitto et al., 2016 deals with aerosols in the ULTS and not with dust. This reference is not really appropriate here. - p4line21: why describing AERONET AOD and associated uncertainties if not used? Even not compared in the following to the MFRSR? - P5: For the surface observations, several instruments measuring irradiance are described, but it's not clear if they are all used in this study. Why describing every instrument available if they are not used? Which ones are really used? This section would gain in clarity if simplified. - P6line13: maybe a summary of the description of the meteorological and dust conditions given in Denjean et al., 2016 would help? It's

easier for a reader to get all the relevant information in one paper. - P7line15: unity switched from $\mu$m to cm-1. Maybe it would be clearer for the reader to stay in $\mu$m? - P8line 18-19: How the AOP (i.e. spectral extinction, single scattering albedo, and phase function at each layer) can be derived from AERONET observation and from Deanjean et al., 2016, in particular for longwave? Observations made by AERONET or Denjean et al. are made in visible wavelength, not in the infrared part of the spectrum. Optical properties cannot be derived in the longwave by these measurements. This sentence is in contradiction with the procedure described later where the size distribution from AERONET and the ATR-42 are used with independent refractive indices to derived these optical properties. - P9line 14-15: which is the RI used from OPAC? MITR? Must be cleared. Similarly, which RI from DB2017 is used? In Table 2 "Algeria-Tunisia-Morocco dust" is mentioned, but it corresponds to 3 different indices, which one is used? - P9line 19: the sentence "the AOPs are calculated using the AERONET and the in situ SD and the OPAC water soluble RI" have to be rewritten by something like "the AOPs are calculated using either the AERONET or the in situ SD and the OPAC water soluble RI. - P9 line 4: A table given the main parametrization of the SD (radius and width of the distribution size) used would avoid to refer systematically to the paper of Deanjean et al., 2016. - P10 Eq 2: What does the symbol Delta stand for? - P10-11 and Table 3: the uncertainty for the WINDOW irradiance have been changed from 2 to 6 W.m2 from the previous version of the paper. In any cases in the text I read an uncertainty of 3W.m2 (p5, line 19). Which is the good one? In addition, the observed values for the downward irradiance and BT correspond to the average over a 10 minutes interval, what is the standard deviation of the measurement compared to the uncertainty? If the standard deviation is of the order of uncertainty, it means that the signal present small variation within the 10 minutes and maybe it would be better to compare simulations with observation, for every observation within the 10 minutes and average after instead of comparing with the average observation? If the standard deviation is larger, it means that using constant aerosol distribution is not valid. - All the section 4.1 need to be slightly reorganized. In particular, the sentence line 23-24 page 10 is very general

for the three days and the three variables, and therefore need to be at the beginning of the paragraph, as well as the sentence line 18-21, which is the associated explanation or put at the end of the section as conclusion. Furthermore, the paragraph need an overall analysis of the results obtained at the end: For LW irradiance and WINDOW irradiance, the impact of the refractive index is below the uncertainty of observation, the impact of the RI for a given SD is close to the uncertainty. For WINDOW, simulations always overestimate the observation, implying that the RT model or the calibration is not correct for this simulation and therefore it is not clear to understand what bring this variable in the study... On the contrary, for IR BT, the impact of the SD as well as the RI is significant compared to the uncertainty, this variable seems to be more appropriate to analyze AOP. - P11 line 27: "The average AOD during the descent is assumed as model input.": Which AOD is used? The average column integrated value measured by the MFRSR? - P11line30-33: there is no reason for which the NOAER simulation agree well for all the profile except close to the surface given the aerosol distribution of Figure 3. Something may be missing in the simulation to reproduce the observed downward irradiance in the lower part of the atmosphere that is not due to aerosols. - P12line 1-3: "This confirms the results found for the surface irradiance, i.e. that the broadband irradiance alone cannot help discriminating which SD and RI provide the best representation of the dust optical properties". This conclusion is not clearly stated in the previous part (see my comment on the section 4.1). - P12line 12-13: "However, while the model-measurement agreement is very good at 600 and 3300 m, where the aerosol impact is small, a systematic overestimation is obtained at 5670 m.": from Figure 7, there is no evidence of a "systematic overestimation [...] at 5670 m". COL1 and INSU3 seems to fit well observations at $12\mu$m, whereas an overestimation is obtained at $8.7\mu$m. At $10.6\mu$m COL1 induces an overestimation, but INSU3 an underestimation of the observation. I don't see therefore "a systematic overestimation". - P12 line 15-16: "These results show that exploring the BT in the thermal infrared is a useful tool to infer dust optical properties if the SD is provided.": this sentence should be slightly attenuated: These results show the better sensitivity of BT to dust optical properties

than broadband irradiance but for the two other days, where aerosols are lower or with a smaller AOD, the differences between simulations with different AOP are of the order of the observed uncertainty. - P13line 26: "while no BT increase is detected at 1700 m": I don't understand this sentence, on Figure 9, there is an increase of BT at 1700m. - As for the section 4.1, this section lacks of a conclusion that summarizes the different simulations. Basically, the LW irradiance is not really sensitive to AOP (impact under the uncertainties). In addition, something appears to be missing in the simulations, because simulated upward irradiances are systematically overestimated by simulation in some part of the profile (for the three cases, even it is less important in the first one). An explanation, or at least some hypothesis, of this overestimation is missing in the paper. - P14 line 14-15 "The resulting standard deviation on the TOA spectral radiance is around 1% for 22 and 28 June and 0.5% for 3 July": this value requires to be in K, in order to be compared with the radiometric noise and more over to be compared with the impact of the different AOP. Given that 1% corresponds to a variation of about 2.9K (~1% of the surface temperature), this standard deviation is larger than the impact of the aerosol properties themselves. It should be better to apply the simulations to each spectrum and then average the differences. - Table 7: In the caption is written "Differences (K) between modelled and measured IASI BT spectra" at the beginning and "Differences are expressed as percent RMSD and standard deviation" at the end. The differences are in K or in %? The sentence "In bold the significant differences with respect to the NOAER simulations." Is not clear. What means "significant differences with respect to the NOAER simulations"? What is the criterion to put in bold the difference? - P14line23 to p15line 6: this paragraph need to be reorganized by day instead of analyzing figure 12, and then Table 7 since the conclusions are the same and it would avoid redundant sentences. - P15 line 4-5: this sentence repeat statement already given in the previous paragraph or has to be rewritten. - P15 line 7-24: the paragraph describing the analysis of Liuzzi et al. (2017) is very long to finally conclude that the impact of INSU3 is of the same order. Either better details of what this study bring compared to the previous one, or why this study use a simplify RT models compared

to the previous one is given, or this paragraph has to be shortened. But for now, it's difficult to see where the author is going. - P15 line 10: correct "or" by "for" in "The real part is also generally lower or Shettle and Fen " - Section 4.4: it would be interesting to see also the results for the two other days and the AOP INSU1 in order to have an idea of the variations of the radiative forcing from very different cases.

---

## Referee Comment (RC2) · Anonymous Referee #2 · 28 Oct 2017

Review of paper: acp-2017-591 "Determining the infrared radiative effects of Saharan dust: a radiative transfer modelling study based on vertically resolved measurements at Lampedusa" by D. Meloni et al.

General comments

In this paper radiation closure experiments are made in order to determine the infrared radiative effects of dust and to assess the role of dust size distribution (SD) and refractive index (RI). To this aim, in situ data from aircraft (ATR-42 and Falcon), surface

(AERONET, radiometer, pyranometers, pyrgeometers and pyrometer) radiosonde and satellite (IASI) measurements are utilized for the closure. The measurements come from the ADRIMED/ChArMEx campaign in 2013. The vertically resolved simulations are performed with the MODTRAN radiative transfer model (RTM) initialized by in-situ vertical and remotely sensed columnar SD and RI along with data for a series of surface and atmospheric parameters relevant to LW radiation transfer, coming from radiosoundings, spectrophotometer measurements, ECMWF reanalysis and MODIS satellite products. The assessment lies in comparing simulated and measured LW irra-diances and brightness temperatures (BTs), while the dust LW radiative forcing (ARF) and atmospheric heating/cooling rates (AHR) is estimated with the RTM. Three cases (summer days) during a period of dust intrusions (late June and early July) are exam-ined, and the study is performed for Lampedusa in central Meditteranean, in proximity to northern Africa and Sahara.

The study is detailed and makes synergistic use of a variety of data. Some interesting findings are reported, from which some are not always new, e.g. that the dust LW radiative effects are non negligible or that the heating rate profile of dust depends on its vertical distribution as well as on SD and RI. Yet, some others provide new information and give insight regarding the role of dust SD and RI for their LW radiative and thermal effects and for BT, e.g. that using dust RI from local dust sources (Algeria and Morrocco, DB2017) produces best agreement with observations or that the use of inaccurate, although optically equivalent SD and RI has a large impact on the dust ARF. The paper is well organized and nicely written although it sometimes lacks clarity in the discussion of its results.

The main issue is that the paper seems to fail to convince about the best performance and appropriateness, and to provide a clear message on what is the optimal combi-nation of dust properties for achieving the radiation closure. The relevant messages drawn from the simulations-measurements comparisons of LW and WINDOW fluxes, and of BTs, are not consistent and appear to be somewhat contradictory, as it is for example the case in Table 3. Even the authors state (page 15, lines 20-21) that "the MOD-TRAN spectral resolution impacts the standard deviation of the model-measurements differences, making the results obtained with different AOPs equivalent". More specifically:

Main Comments

1) In general, quite small differences between the 7 examined configurations, consisting in different model setups (Table 2), are found between results obtained without aerosols and with aerosols, as well as between the 6 configurations with aerosols (3 columnar and 3 in-situ). This does not help to draw a clear conclusion on which one configuration and aerosol properties combination is the best, although this isn expected to be the main finding of such a radiation closure study.

2) The ascertained/computed differences of each one of 7 configurations with respect to measurements (LW, WINDOW, BTs) mostly fall within the range of uncertainty of measurements, making difficult to decide on which one is really the best configuration.

3) A main conclusion drawn from the analysis is that there is a systematic model overestimation of upward LW fluxes within the peak of dust layers, in all 3 days. In other words, there seems to be an inherent problem with the modelling tool, which needs to be assessed.

4) The estimated small differences between the no-aerosol and aerosol configurations, indicate that the RTM LW computations are relatively insensitive to dust.

5) The reported conclusions are sometimes contradictory. For example in page 17, lines 14-15 it is stated that dust RI from DB2017 produces the best agreement with observations, but this is not supported by and it is not in line with the results of Table 3 where NOAER and COL1 also provide good results, even better than INSU3, if all three parameters, i.e. LW, WINDOW, BT, and three days are considered.

6) It is not clear why BTs were computed and are reported only at 3 levels, which

sometimes are not collocated with the peaks of dust layers; why similar BT computations were not made at more levels.

7) The conclusions drawn from the BT analysis are different from those obtained from the analysis of LW fluxes. This is for example the case of the results of profile 42, in Figures 10 and 11. May this point to a possible modelling problem/inconsistency?

8) The role of clouds is not reported. Were all the tree days/cases cloud-free? If so, how is this confirmed/ensured? A relevant discussion should be made since the effect of clouds on LW is significant and interplay or even dominate the effect of dust (e.g. possible implications for Fig. 2).

Specific Comments

1. Page 1, line 28: define IASI acronym.

2. Page 4, Figure 1: the AERONET AOD may also be overplotted.

3. Page 4, line 18: the reported angstrom exponent is high, it is about the maximum one; give a more realistic value (range).

4. Page 5, lines 32-35: why? Please explain.

5. Page 5, lines 35-37, "The pyrometer ... for the IRP BT": this sentence is over-simplified. A quick look at the 3 figures reveals significant differences between BT and irradiances. For example, what happens in June 24 and 25 (when LW-WINDOW curves do not have peaks, opposite to IRP BT)? What about the role of temperature and clouds?

6. Page 7, line 7: define FWHM acronym.

7. Page 7, line 30: up to which altitudes? How much the use of standard profiles can affect the radiative fluxes? Was any sensitivity study performed to assess this? Especially the LW fluxes should be sensitive.

8. Page 7, line 33, regarding the absorbing gases: similarly, it would be worth to discuss/assess the sensitivity of fluxes to these parameters, especially given the scaling applied to their vertically distributed values.

9. Page 8, about ECMWF: The use of reanalysis data is inevitable in this case. However, an assessment of the induced uncertainties associated with their coarser resolution could be made by comparing similar ECMWF data with available measurements for the other two days. This could provide an estimation of induced uncertainties in June 22.

10. Page 8, line 10: a few words about the measured aerosol properties and the identified aerosol layers can be added. For example, apart from the layers and their extension neither information is given nor reference is made to the type of aerosols in each layer, with reference to corresponding measurements that cloud provide this kind of information.

11. Page 8, line 14: so, what values of emissivity were assumed in the study? Do they differ and how much from day to day.

12. Page 8, Figure 4: the quality should be improved, e.g. by thicknenning the curves, so that the coloured curves can be more easily discerned.

13. Page 8, line 29: As mentioned, different factors influence and differentiate the AERONET and in-situ SDs, one important being their different value, .e. columnar versus vertically resolved. The value of detailed measurements is that they provide vertically resolved SDs. Therefore, emphasis should be given to them. Discuss a bit more how the measured SDs differ to AERONET ones, referring to their agreement and disagreement. For example, larger differences appear in June 22 than in July 03. Refer to this difference referring to the nature of vertical profiles of Fig. 3 and the type of aerosols that are present in the different layers of every daily profile.

14. Page 9, line 19: explain why this choice of water soluble RI was made and not any

other.

15. Page 9, lines 20-26: Table 2 is not discussed enough. It should be said more clearly what exactly has been done and how the Mie-based computations of AOD compare to AERONET ones, whenever applicable, i.e. in visible wavelengths.

16. Page 10, lines 16-17: does this refer to July 03? In Table 3 no results for INSU2 are displayed.

17. Page 10, lines 19-20: why the stronger infrared emission? Is it a matter of larger mass? Please explain.

18. Page 10, line 24: clarify that "all cases" refer to LW, WINDOW and IR BT.

19. Page 10, line 34: here it should be clarified what is exactly the spectral interval/coverage of the measurements (IRP). This not clear based on what is said in page 7, line 6, about the IRP centered at 3 wavelengths etc. It is essential to clarify what is exactly the spectral coverage of measurements since they are used as the reference to which the simulations are compared, and given the significant sensitivity of theoretical computations to the spectral interval. Also explain why the reduction in WINDOW irradiances has different magnitude despite the same spectral reduction (0.4 microns) in different spectral parts.

20. Section 4.1: what is missing is a critical approach providing insight into possible physical reasons for better agreement between the 7 examined cases. A quite exhaustive and very detailed description of results is made, referring to various numbers (Table 3). This is not enough while it turns to be confusing to the reader. What is more important is to determine which set of AOPs is more efficient and compared better to the measurements for the 3 cases. The discussion should conclude on this, stating at least if there is a "best" choice or if there is not and why. Moreover, in both cases, the discussion should provide a physical basis for the outcome of the analysis and the closure of Table 3. For example, a summary of the results of Table 3 should point to

Interactive
comment

NOAER being the most efficient simulation, providing better results than the other 6 sets of AOPs in 4 cases (out of totally 9, i.e. 3 days by 3 parameters). NOAER is followed by COL1 (3 cases with best performance) and INSU3 (2 cases). So, questions may arise, like why simulations without aerosols should be more appropriate/realistic, or why INSU3, which may be expected to be the most realistic, is finally not.

21. Page 12, line 3: as to upward LW, authors may want to comment on why the smallest differences are for COL1 in Table 5, while the smallest RMSDs in Table 4 are for INSU1.

22. Figure 7: why only points for NOAER, COL1 and INSU3 are given and not for the other cases? All these appear in Table 6.

23. Page 12, lines 15-16: add "in-situ" before SD. This sentence needs to be re-written, since it is introduced all suddenly without being given evidence and discussed based on the results of Fig. 7.

24. Page 12, lines 17-19: while discussion is made no results are shown/given.

25. Section 4.2.1: A quite exhaustive discussion is made making frequent reference to numbers that differ a while between the 6 examined cases. Also the question arises why NOAER sometimes performs equally or better than dust-including cases. It could point to potential artifacts due to counteracting effects of other parameters than aerosol, which affect the LW radiation transfer and BT.

26. Page 12, lines 35-36, "These differences . . . airborne instrumentations": so is there an inherent problem with the model?

27. Page 14, line 20" add "was" before "evaluated".

28. Page 14, line 21, "resampled": how it was done?

29. Page 14, lines 3237: why there is difference on what provides the best match with reference to best match with the measured spectra and BTs?
30. Page 15, lines 2-3: this is not applicable to 780-980/cm for June 22 and 28.

31. Page 15, lines 20-22, "In our case, . . . AOPs equivalent.": what exactly is it meant by this? By which means. Please explain. Is it implied that this (having very high resolution) is preferable? If so, why? If valid, it would mean that AOPs are not important for accurately computing LW radiation and dust LW radiative effects. Is this the meaning?

32. Page 15, line 28, "The combination . . . downward": this is not clearly evidenced in the discussion of sections 4.1 and 4.2.

33. Page 26, Table 2: the Table needs further/better explanation, it is not very easy for the reader to understand what exactly is the information given in this Table.

34. Page 37, Figure 5: what have been the criteria for the design of flight paths.? Nothing is said about this and deserves to be mentioned in the text.

35. Page 44, Figure 12: wavelengths could be added, e.g. on the top x-axis.

---

## Author Comment (AC1) · 6 Dec 2017

General comments: This paper lies in the framework of the ChArMEx/ADRIMED experiment, that took place in the Mediterranean in summer 2013. Three vertical profiles of atmospheric and aerosol properties, made at Lampedusa in conjunction with surface, airborne and satellite IR broadband and narrowband radiation as well as radiosonde are analyzed in order to 1) identify the sensitivity of the different radiative measurements to mineral dust microphysical properties (size distribution and refractive index) and 2) analyze their impact in term of radiative forcing. The main result of this study is that if LW irradiance is poorly sensitive to aerosol microphysical properties compared to brightness temperature, the IR dust radiative forcing is non-negligible, and strongly depends on size distribution (SD) and refractive index (RI). This study highlights the importance of a precise knowledge of the dust microphysics to infer correctly their radiative effect. The paper is an interesting sensitivity study of the radiative variables to the aerosol microphysics, leading, in particular, to the conclusion that spectrally resolved measurements of brightness temperature is more fitted to infer dust properties than broadband LW irradiances. However, the part that concludes on the most appropriate refractive indices is less convincing. Such a study would require a more detailed analysis of the differences between refractive indices (at minimum a figure displaying their values in the spectral domain concerned), as well as a more exhaustive variability in the choice of the indices. Here among the three indices used, two indices are quite similar and only one coming from recent measurement campaign of DiBiagio et al., 2017 is really different. Moreover, the study, based principally on RT simulations, lacks of discussions on the uncertainties due to the RT model itself, as well as to the different hypothesis used. In particular, which is the impact of an error in surface temperature or surface emissivity? An error on the water vapor profile? No reference error is calculated under clear sky condition for example, to distinguish error directly due to the model from errors due to the impact of aerosols properties. The resulting biases obtained with the different aerosol properties configurations cannot therefore be really discussed. For example, large biases between simulated and calculated irradiances are not explained (and apparently not due to wrong aerosol properties), implying that something is missing in the RT model, but not enough discussed. The section on IASI data is not enough developed. All the spectra within a box of about 100 kmx100 km are averaged before analysis, causing a standard deviation in the averaged spectrum larger than the effect of the aerosols properties analyzed! Here again, since no reference errors are given, biases obtained from the different aerosol configurations are finally equivalent and it's not possible to state on the best configuration. This part doesn't really bring new information compared to the previous sections or previous studies, or required a more precise development. Finally, some details on the inputs used are missing. A few details are provided on the size parameters used (the reader has to refer to the paper of Denjean et al. 2016 to have the precisions) and the exact refractive index from DiBiagio et al., 2017 use in this study is not given: 3 different indices are coming from ´n¢athe source regions (Tunisia, Algeria, Morocco) in DiBiagio et al., 2017 whereas only one is used here without any precision!

We agree with the reviewer that a comprehensive analysis including the impact of the uncertainty on the input parameters on the model simulations (of either irradiances and brightness temperature profiles and of radiances at the top of the atmosphere) is useful to better constrain the results. Following the reviewer's suggestion, a sensitivity study addressing the uncertainties of the modelled radiation quantities due to the uncertainty on the input parameters has been carried out, either in aerosol-free conditions or including the aerosol particles. The main model input parameters

affecting infrared radiation in aerosol-free conditions that have been considered are: integrated water vapour, temperature profile, sea surface temperature, surface emissivity. Each quantity has been perturbed one at a time by the amount of its uncertainty, than all the resulting model uncertainties has been combined to provide the overall uncertainty. Similarly, the sensitivity with respect to AOD and dust complex refractive index has been quantified.

The results have been added to the manuscript under the new Section 3.3.

The model LW irradiance uncertainty decreases with increasing altitude for both the downward and upward components. The estimated model uncertainty on the downward and upward LW irradiances at the surface is 2.2 and 2.0 W m$^{-2}$ for simulations without and with aerosols, respectively. At the Falcon 20 altitude (about 10 km) the uncertainties are 0.6 and 1.5 W m$^{-2}$ for the downward and upward component, respectively, for both simulations with and without aerosol. The upward LW irradiance uncertainty is 1.4 W m$^{-2}$ at TOA, for simulations with and without aerosol.. The estimated uncertainty on ARF is obtained by the combination of the above values, and is 4.2 W m$^{-2}$ at the surface and 2.0 W m$^{-2}$ at the TOA. The uncertainty on AHR is largest at about 4.5 km altitude (0.030 K day$^{-1}$ with aerosol and 0.026 K day$^{-1}$ without aerosol), and close to the surface (0.050 K day$^{-1}$ with and without aerosol).

The uncertainty on the downward WINDOW irradiance is 0.9 and 0.6 W m$^{-2}$, with and without aerosol, respectively. The estimated uncertainty on the modelled zenith BT is 0.7 and 0.3 K, with and without aerosol, respectively.

The aerosol-free CLIMAT BT is much sensitive to SST and surface emission, with slightly larger values at 600 m (0.3 K) than at 5670 m (0.28 K). The overall uncertainty for the case with aerosol is 0.31 K at 600 m and 0.37 K at 5670 m.

The uncertainty on the spectral BT at TOA in the atmospheric window varies between 0.25 and 0.29 K in aerosol-free conditions, and between 0.32 and 0.50 K with aerosol.

The model-measurement differences have been discussed in the text taking into account the uncertainties on the model estimates.

A figure displaying the spectral complex refractive indices used in this study has been added as Supplement Material (Figure S1). With this regard, a summary of the most common complex refractive indices of desert dust is provided in Di Biagio, C., Boucher, H., Caquineau, S., Chevaillier, S., Cuesta, J., and Formenti, P.: Variability of the infrared complex refractive index of African mineral dust: experimental estimation and implications for radiative transfer and satellite remote sensing, Atmos. Chem. Phys., 14, 11093-11116, doi:10.5194/acp-14-11093-2014, 2014. Moreover, the spectral (0-40 µm) normalized extinction coefficients, single scattering albedoes, and asymmetry factors computed using the combination of SDs and RIs described in the text for each layer of the three profiles have been shown in Figure S2, S3, and S4 of the Supplement Material.

The dust refractive indices that we use in the 0-40 µm range have been chosen because they are specific for the source regions found during the ChArMEx campaign (like those from Tunisia, Algeria, Morocco by Di Biagio et al., 2017), or because they are widely used in the retrieval of satellite products or in climate models (like the ones from OPAC by Hess et al., 1998, and by Volz, 1973).

OPAC and Volz (1973) have very similar real and imaginary parts, except for the 9.5-14 µm spectral interval, where we explore the dust impact on the surface irradiance (in the 8-14 µm window) and brightness temperature (9.6-11.5 µm) and in the CLIMAT and IASI brightness temperatures (BTs). The results in Table 3 of the manuscript confirm that the model-measurement differences in the 9.6-11.5 µm BT can be significant (0.7 K) when the OPAC size distribution is used, while are modest (0.4 K) with the *in situ* size distribution.

Other dust refractive indices found in literature were taken into account, although the results are not reported in the manuscript: for example, the Volz (1972) one, which is equivalent to the one

published in Shettle and Fenn (1979). The imaginary part is much lower than that of OPAC and Volz (1973), so lower surface LW and WINDOW irradiances and infrared BT are expected, with consequently modest radiative effect.

The aerosol optical properties obtained with the Longtin et al. (1988) dust refractive index were already examined in a previous paper (Meloni et al., Altitude-resolved shortwave and longwave radiative effects of desert dust in the Mediterranean during the GAMARF campaign: Indications of a net daily cooling in the dust layer, J. Geophys. Res. Atmos., 120, 3386–3407, 2015).

All the three refractive indices by Di Biagio et al., 2017 (Algeria, Tunisia, Morocco) have been used in the study, since the analysis by Denjean et al. (2016) based on back-trajectories and MSG-SEVIRI satellite products shows that dust collected during F35, F38, and F4 flights have different source regions (details are given in Table 1 of Denjean et al., 2016). More specifically, for flight F35 the dust layer above-3.5 km originated from southern Algeria, while the dust layer between 1.5 and 0.5 km was transported from southern Morocco. On 28 June (F38) dust was lifted from Tunisia. Finally, on 3 July (F42) dust originated from Tunisia (above 3 km) and from southern Morocco (below 3 km).

A better description of the choice of the appropriate refractive index on the base of the dust source region was added in Section 3.1.1 (page 9, lines 1-19). We have used different refractive indices from Di Biagio et al. (2017) for each flight and each dust layer based on the source regions found in Denjean et al. (2016). For F35 we used the refractive index for Algerian dust in layer 3 (see Figure 3), that for Moroccan dust in layer 2, and the OPAC water soluble refractive index for layer 1, i.e. below the dust layer. Similarly, the Tunisian dust refractive index is used for F38 flight in layer 2 and the OPAC water soluble one in layer 1. For F42 the Tunisian and the Moroccan dust refractive index are used in layer 2 and 1, respectively.

We clarified the choice of the dust refractive index in Section 3.1.1 and prepared a new Table 2 which includes three tables, one for each day. The tables present the combination of SD and RI used in each layer identified by the lidar and ATR-42 measurements, and the AOD value at 8.6 μm.

Specific comments:
- p2line 21ˇ a: ˊnˇaMost of these studies have been carried out close to the dust source regions and did not take into account the possible modifications in dust optical properties during long range transport. ": You can also find some studies dealing with the variation of the aerosol properties with transport (e.g. Maring 2003; Ryder et al., 2013; Weinzierl et al., 2017 and so on). Moreover, I don't see the link with the subject of this paper since there is no discussion on the possible change of dust properties with transport. Maring, H.: Vertical distributions of dust and sea-salt aerosols over Puerto Rico during PRIDE measured from a light aircraft, J. Geophys. Res., 108(D19), 1–11, doi:10.1029/2002JD002544, 2003. Ryder, C. L., Highwood, E. J., Lai, T. M., Sodemann, H. and Marsham, J. H.: Impact of atmospheric transport on the evolution of microphysical and optical properties of Saharan dust, Geophys. Res. Lett., 40(10), 2433–2438, doi:10.1002/grl.50482, 2013. Weinzierl, B., ProsPero, J., Chouza, F., FomBa, W., Freudenthaler, V., GasteiGer, J. and Toledano, C.: THE SAHARAN AEROSOL LONG-RANGE TRANSPORT AND AEROSOL–CLOUDINTERACTION EXPERIMENT Overview and Selected Highlights, [online] Available from: http://journals.ametsoc.org/doi/pdf/10.1175/BAMS-D-15-00142.1

We thank the reviewer for suggesting the papers that have been integrated in the Introduction, citing them as example of studies carried out by means of aircraft measurements dealing with the temporal evolution of dust properties occurring during long-range transport.

Our study focuses on the optical properties of transported dust and on its infrared radiative effect in the Mediterranean. The present paper does not aim at assessing how dust optical properties change during transport (Denjean et al., 2016 show indeed that the coarse mode of dust did not change after 5 days of transport possibly due to strong vertical turbulence within the dust layer, preventing the deposition of large particles), but highlights the importance of knowing the dust microphysical and

optical properties to reasonably estimate the IR dust radiative forcing and heating rate at the surface, in the atmosphere, and at the top of the atmosphere.

- P2line 29:
the paper Sellitto et al., 2016 deals with aerosols in the ULTS and not with dust. This reference is not really appropriate here.
The reference has been removed.

- p4line21: why describing AERONET AOD and associated uncertainties if not used? Even not compared in the following to the MFRSR?
We agree with the reviewer. The AERONET AOD is not used because of some missing data, so the sentence in lines 22-23 about AOD uncertainty has been removed.

- P5: For the surface observations, several instruments measuring irradiance are described, but it's not clear if they are all used in this study. Why describing every instrument available if they are not used? Which ones are really used? This section would gain in clarity if simplified.
In Section 2.1 all the ground-based instruments are presented. For sake of clarity, we removed all instruments (like the shortwave radiometers and the pyrheliometer) whose measurements are not used in the analysis.

- P6line13: maybe a summary of the description of the meteorological and dust conditions given in Denjean et al., 2016 would help? It's easier for a reader to get all the relevant information in one paper.
A short description of the synoptic conditions causing dust transport from the Sahara desert to Lampedusa during the campaign has been added in the text.

- P7line15: unity switched from µm to cm-1. Maybe it would be clearer for the reader to stay in µm?
IASI spectral characteristics, like spectral interval, sampling and resolution, are provided in units of $cm^{-1}$. We have used both µm and $cm^{-1}$ in the text and in figures and table whenever possible to help the reader.

- P8line 18-19: How the AOP (i.e. spectral extinction, single scattering albedo, and phase function at each layer) can be derived from AERONET observation and from Deanjean et al., 2016, in particular for longwave? Observations made by AERONET or Denjean et al. are made in visible wavelength, not in the infrared part of the spectrum. Optical properties cannot be derived in the longwave by these measurements. This sentence is in contradiction with the procedure described later where the size distribution from AERONET and the ATR-42 are used with independent refractive indices to derived these optical properties.
We agree with the reviewer that the sentence in lines 18-19 is misleading. The IR AOPs are not derived from AERONET or airborne observations alone.
The aerosol optical properties in the infrared spectral range are calculated applying the Mie theory using the AERONET and the *in situ* size distributions and the complex refractive indices in the 3-40 µm (OPAC) and in the 2-16 µm (Di Biagio et al., 2017) intervals.
The sentence has been rephrased.

- P9line 14-15: which is the RI used from OPAC? MITR? Must be cleared. Similarly, which RI from DB2017 is used? In Table 2 "Algeria-Tunisia-Morocco dust" is mentioned, but it corresponds to 3 different indices, which one is used?

We used the mineral dust refractive index from OPAC, which is the same for the four dust types (accumulation, coarse, nucleation, transported) of the model. This has been clarified in the text. As for the answer to the reviewer's general comments, we have used different refractive indices from Di Biagio et al. 2017 for each flight and each dust layer based on the source regions found in Denjean et al. 2016. More details are now given in section 3.1.1 and Table 2.

- P9line 19: the sentence "the AOPs are calculated using the AERONET and the in situ SD and the OPAC water soluble RI" have to be rewritten by something like "the AOPs are calculated using either the AERONET or the in situ SD and the OPAC water soluble RI.
We rephrased the sentence according to the reviewer's suggestions.

- P9 line 4: A table given the main parametrization of the SD (radius and width of the distribution size) used would avoid to refer systematically to the paper of Deanjean et al., 2016.
We have produced Table S1 with the median radius, standard deviation, and normalized number concentration for each mode of either the AERONET and the *in situ* log-normal size distributions for the three cases as Supplement material.

- P10 Eq 2: What does the symbol Delta stand for?
The Delta symbol in the heating rate equation represents the variation of net flux and pressure between two contiguous layers.

- P10-11
and Table 3: the uncertainty for the WINDOW irradiance have been changed from 2 to 6 W.m2 from the previous version of the paper. In any cases in the text I read an uncertainty of 3W.m2 (p5, line 19). Which is the good one? In addition, the observed values for the downward irradiance and BT correspond to the average over a 10 minutes interval, what is the standard deviation of the measurement compared to the uncertainty? If the standard deviation is of the order of uncertainty, it means that the signal present small variation within the 10 minutes and maybe it would be better to compare simulations with observation, for every observation within the 10 minutes and average after instead of comparing with the average observation? If the standard deviation is larger, it means that using constant aerosol distribution is not valid.
The measurement uncertainty reported in Table 3 is the expanded (2-sigma) uncertainty, but this was not specified in the table caption, so it seems to disagree with what explained in the text (page 5, lines 19-20), referring to $\pm 3$ Wm$^{-2}$ as one sigma uncertainty. This has been better specified in the revised paper.
However, the CGR3 participating to the ChArMEx campaign has been recently tested by PMOD/WRC to assess the possible effect due to the leakage of solar radiation on the WINDOW irradiance. The tests have shown that the effect is negligible, and so the WINDOW irradiance data used in the present analysis have to be reconsidered because they were corrected by subtracting a shortwave stray-light correction of about 4 Wm$^{-2}$ per 1000 Wm$^{-2}$ solar irradiance. So in the revised paper the CGR3 expanded uncertainty returns to be $\pm 2$ Wm$^{-2}$ and the WINDOW irradiance data have been corrected, either in Table 3 and in Figure 2.
The standard deviations of the LW and WINDOW irradiance, and of the zenith BT over the 10 minute interval are much lower than the measurement uncertainty. For example on 22 June the standard deviation values are 0.2 Wm$^{-2}$ for the LW irradiance, 0.3 Wm$^{-2}$ for the WINDOW irradiance, and 0.1 K for the BT. We assume that no significant variations occur within the 10 minute interval and that differences can be calculated between the average value and the model simulation. A sentence has been added in the text to state the very low variability within the 10 minute time interval.

- All the section 4.1 need

to be slightly reorganized. In particular, the sentence line 23-24 page 10 is very general for the three days and the three variables, and therefore need to be at the beginning of the paragraph, as well as the sentence line 18-21, which is the associated explanation or put at the end of the section as conclusion. Furthermore, the paragraph need an overall analysis of the results obtained at the end: For LW irradiance and WINDOW irradiance, the impact of the refractive index is below the uncertainty of observation, the impact of the RI for a given SD is close to the uncertainty. For WINDOW, simulations always overestimate the observation, implying that the RT model or the calibration is not correct for this simulation and therefore it is not clear to understand what bring this variable in the study: : : On the contrary, for IR BT, the impact of the SD as well as the RI is significant compared to the uncertainty, this variable seems to be more appropriate to analyze AOP.

We agree with the reviewer. Section 4.1 has been revised, the results commented taking into account the model uncertainties, and a concluding sentence has been added at the end of the section. Although the WINDOW irradiance is overestimated by the model, its simulations with different AOPs have been included in the analysis to show that, even when reducing the spectral interval compared to the broadband, the irradiance is not sensitive to varying AOPs.

- P11 line 27: "The average AOD during the descent is assumed as model input.": Which AOD is used? The average column integrated value measured by the MFRSR?

Yes, the AOD measured by the MFRSR during the flight and reported in the IR as explained in section3.1.1 has been averaged and used as model input.

- P11line30-33: there is no reason for which the NOAER simulation agree well for all the profile except close to the surface given the aerosol distribution of Figure 3. Something may be missing in the simulation to reproduce the observed downward irradiance in the lower part of the atmosphere that is not due to aerosols.

Figure 6 shows that model without dust (NOAER) underestimates measurements by an amount that is negligible at higher altitudes and increases close to the surface. This effect is due to emission of infrared radiation by the dust above each altitude layer which induces an increase of the downward LW irradiance, and which depends on the dust optical depth and optical properties.

- P12line 1-3: "This confirms the results found for the surface irradiance, i.e. that the broadband irradiance alone cannot help discriminating which SD and RI provide the best representation of the dust optical properties". This conclusion is not clearly stated in the previous part (see my comment on the section 4.1).

A concluding sentence has been added in Section 4.1 to summarize the results.

- P12line 12-13: "However, while the model-measurement agreement is very good at 600 and 3300 m, where the aerosol impact is small, a systematic overestimation is obtained at 5670 m.": from Figure 7, there is no evidence of a "systematic overestimation […] at 5670 m". COL1 and INSU3 seems to fit well observations at 12_m, whereas an overestimation is obtained at 8.7µm. At 10.6µm COL1 induces an overestimation, but INSU3 an underestimation of the observation. I don't see therefore "a systematic overestimation".

The sentence was present in a previous version of the manuscript and was erroneously maintained in the submitted version. It has been removed. As the reviewer points out, the overestimate of the model depends on spectral band and on aerosol optical properties.

- P12 line 15-16: "These results show that exploring the BT in the thermal infrared is a useful tool to infer dust optical properties if the SD is provided.": this sentence should be slightly attenuated: These results show the better sensitivity of BT to dust optical properties than broadband irradiance but for the two other days, where aerosols are lower or with a smaller AOD, the differences between simulations with different AOP are of the order of the observed uncertainty.

The sentence has been modified according to the reviewer's suggestion in "These results show the better sensitivity of BT to dust optical properties than broadband irradiance", since the paragraph refers to flight F35 only.

- P13line 26: "while no BT increase is detected at 1700 m": I don't understand this sentence, on Figure 9, there is an increase of BT at 1700m.
The reviewer's comment is right. At 1700 m, as well as at 900 m, either the pyrgeometer and the CLIMAT capture the infrared increase due to the island emission. The sentence has been changed in "The spikes at 900 m and around 1700 m indicate that also CLIMAT captures the island emission.".

- As for the section 4.1, this section lacks of a conclusion that summarizes the different simulations. Basically, the LW irradiance is not really sensitive to AOP (impact under the uncertainties). In addition, something appears to be missing in the simulations, because simulated upward irradiances are systematically overestimated by simulation in some part of the profile (for the three cases, even it is less important in the first one). An explanation, or at least some hypothesis, of this overestimation is missing in the paper.
A concluding sentence has been added at the end of the section, explaining that irradiance is not sensitive to the AOPs, while the aerosol perturbation to the upward infrared BT can be appreciated only at altitudes above the bulk aerosol emission, like for the flight F35.
Differences between the measured and the model upward LW irradiance profiles for 28 June and 3 July cannot be explained by a wrong representation of the temperature and humidity profiles in the model, that would have affected the CLIMAT profiles also, as described in the text. What is observed is that the CLIMAT BT profiles are well reproduced at different altitudes, also close to the surface for flight F38, suggesting a proper choice of atmospheric profiles and of sea surface temperature and emission.
We formulated an hypothesis of some negative bias affecting the pyrgeometers' measurements due to the fact that the instrument needs some time (the typical response time is 6 s but for airborne measurements the required time may be significantly longer, as shown for example by Albrecht et al., Pyrgeometer measurements from aircraft, Rev. Sci. Instrum., 45, 33–38, 1974) to establish equilibrium with the air temperature. So we expect that when the aircraft is descending rapidly pyrgeometer measurements may be affected. We verified that during flights F38 (from 3500 to 2000 m) and F42 (from 4800 to 1600 m), where model values are larger than measurements, the vertical velocities were -5.5 and -5.3 m/s, respectively. On the contrary, from 5400 to 4000 m during flight F38, where non-significant biases between model and measurements are observed, the vertical velocity is sensibly lower, about 2.6 m/s. Similarly, during flight F35, when model-measurement differences are small, the vertical velocity is 2.8 m/s.
Opposite to CGR4, the CLIMAT measurements are not affected by the ATR-42 descent speed because the instrumental response time is much shorter (160 ms).

- P14 line 14-15 "The resulting standard deviation on the TOA spectral radiance is around 1% for 22 and 28 June and 0.5% for 3 July": this value requires to be in K, in order to be compared with the radiometric noise and more over to be compared with the impact of the different AOP. Given that 1% corresponds to a variation of about 2.9K (_1% of the surface temperature), this standard deviation is larger than the impact of the aerosol properties themselves. It should be better to apply the simulations to each spectrum and then average the differences.
The IASI spectra have been expressed as BTs, and averaged within the chosen area: the standard deviation (in the 8-14 µm interval) is about 0.6 K for 22 and 28 June, and about 0.3 K on 3 July. These values, although larger than the IASI radiometric noise, express the variability of the TOA BT in the domain. The aim of the simulation of the IASI spectra is to show that including the aerosol the TOA-leaving radiance decreases by an amount which is larger than the model uncertainty and the IASI radiometric noise. Moreover, we want to investigate the sensitivity of the

modelled TOA-leaving radiance to different AOPs. The results in Table 7 show that an appreciable aerosol effect is detected on 22 June and on 3 July, leading to an improvement in the model simulations compared to the aerosol-free case.

- Table 7: In the caption is written "Differences (K) between modelled and measured IASI BT spectra" at the beginning and "Differences are expressed as percent RMSD and standard deviation" at the end. The differences are in K or in %? The sentence "In bold the significant differences with respect to the NOAER simulations." Is not clear. What means "significant differences with respect to the NOAER simulations"? What is the criterion to put in bold the difference?
The differences between model and measurements are calculated in the two spectral intervals 780-980 cm$^{-1}$ and 1070-1200 cm$^{-1}$. Then for each interval the RMSD and the standard deviation are calculated, with the various AOP and also in the aerosol-free case, and expressed in K. The values that have been highlighted in bold are those for which the TOA BT with aerosol and that without aerosol are different taking into account the respective mean and standard deviation (significant difference). The caption of the table has been corrected.

- P14line23 to p15line 6: this paragraph need to be reorganized by day instead of analysing figure 12, and then Table 7 since the conclusions are the same and it would avoid redundant sentences.
The paragraph has been reorganized in order to present the results more concisely and clearly.

- P15 line 4-5: this sentence repeat statement already given in the previous paragraph or has to be rewritten.
The sentence has been removed.

- P15 line 7-24: the paragraph
describing the analysis of Liuzzi et al. (2017) is very long to finally conclude that the impact of INSU3 is of the same order. Either better details of what this study bring compared to the previous one, or why this study use a simplify RT models compared to the previous one is given, or this paragraph has to be shortened. But for now, it's difficult to see where the author is going.
Liuzzi et al. (2017) use an ad-hoc model to reproduce the IASI TOA spectra with the native spectrometer resolution, with the *in situ* size distribution and Di Biagio et al. (2017) refractive indices, tuning some parameters like SST to achieve a good agreement with the measured spectra. We use a model with lower spectral resolution and all the atmospheric and surface parameters derived from observations.
Achieving similar results of those of Liuzzi et al. (2017) at TOA and reproducing also surface and atmospheric irradiances and BT is not obvious and represents an important result of the closure study.
The description of the analysis carried out by Liuzzi et al. has been shortened and the aim of the simulations of the IASI spectra has been better explained.

- P15 line 10: correct "or" by "for" in "The real part is also generally lower or Shettle and Fen "
Done.

- Section 4.4: it would be interesting to see also the results for the two other days and the AOP INSU1 in order to have an idea of the variations of the radiative forcing from very different cases.
The aerosol radiative forcing and heating rate obtained with the INSU1 AOPs have been added and discussed. The ARF and AHR for the other cases have not been presented because the paper is already long. Below we report the tables with the ARF and ARFE for 28 June and 3 July. The values of the ARF is low on both days, either at the surface and at TOA. When taking into account the uncertainties on the modelled ARF, the ARF is different from zero using INSU1 on 28 June and

with INSU1 and INSU3 (only at the surface) on 3 July. It is worth noticing that the ARFE values are similar in the two days.

The ARF values with INSU3 AOPs are much lower on 28 June and 3 July than on 22 June and we expect an analogous behaviour for the aerosol heating rates.

Table 1. LW ARF and ARFE at the surface, TOA, and in the atmosphere (in W m$^{-2}$) on 28 June calculated with all the AOPs.

| | ARF | | | | ARFE | | | |
|---|---|---|---|---|---|---|---|---|
| AOP | COL1 | COL3 | INSU1 | INSU3 | COL1 | COL3 | INSU1 | INSU3 |
| Surface | 3.0 | 1.4 | 5.7 | 3.6 | 14.3 | 6.7 | 27.1 | 17.1 |
| TOA | 1.3 | 0.6 | 2.5 | 1.8 | 6.2 | 2.9 | 11.9 | 8.6 |
| Atmosphere | -1.7 | -0.8 | -3.2 | -1.8 | -8.1 | -3.8 | -15.2 | -8.5 |

Table 2. LW ARF and ARFE at the surface, TOA, and in the atmosphere (in W m$^{-2}$) on 3 July calculated with all the AOPs.

| | ARF | | | | ARFE | | | |
|---|---|---|---|---|---|---|---|---|
| AOP | COL1 | COL3 | INSU1 | INSU3 | COL1 | COL3 | INSU1 | INSU3 |
| Surface | 3.8 | 2.1 | 7.1 | 4.4 | 14.7 | 8.1 | 27.4 | 17.0 |
| TOA | 1.5 | 0.8 | 2.6 | 1.6 | 5.8 | 3.1 | 10.0 | 6.2 |
| Atmosphere | -2.3 | -1.3 | -4.5 | -2.8 | -8.9 | -5.0 | -17.4 | -10.8 |

---

## Author Comment (AC2) · 6 Dec 2017

Review of paper: acp-2017-591 "Determining the infrared radiative effects of Saharan dust: a radiative transfer modelling study based on vertically resolved measurements at Lampedusa" by D. Meloni et al.

General comments
In this paper radiation closure experiments are made in order to determine the infrared radiative effects of dust and to assess the role of dust size distribution (SD) and refractive index (RI). To this aim, in situ data from aircraft (ATR-42 and Falcon), surface (AERONET, radiometer, pyranometers, pyrgeometers and pyrometer) radiosonde and satellite (IASI) measurements are utilized for the closure. The measurements come from the ADRIMED/ChArMEx campaign in 2013. The vertically resolved simulations are performed with the MODTRAN radiative transfer model (RTM) initialized by insitu vertical and remotely sensed columnar SD and RI along with data for a series of surface and atmospheric parameters relevant to LW radiation transfer, coming from radiosoundings, spectrophotometer measurements, ECMWF reanalysis and MODIS satellite products. The assessment lies in comparing simulated and measured LW irradiances and brightness temperatures (BTs), while the dust LW radiative forcing (ARF) and atmospheric heating/cooling rates (AHR) is estimated with the RTM. Three cases (summer days) during a period of dust intrusions (late June and early July) are examined, and the study is performed for Lampedusa in central Meditteranean, in proximity to northern Africa and Sahara.
The study is detailed and makes synergistic use of a variety of data. Some interesting findings are reported, from which some are not always new, e.g. that the dust LW radiative effects are non negligible or that the heating rate profile of dust depends on its vertical distribution as well as on SD and RI. Yet, some others provide new information and give insight regarding the role of dust SD and RI for their LW radiative and thermal effects and for BT, e.g. that using dust RI from local dust sources (Algeria and Morrocco, DB2017) produces best agreement with observations or that the use of inaccurate, although optically equivalent SD and RI has a large impact on the dust ARF. The paper is well organized and nicely written although it sometimes lacks clarity in the discussion of its results.
The main issue is that the paper seems to fail to convince about the best performance and appropriateness, and to provide a clear message on what is the optimal combination of dust properties for achieving the radiation closure. The relevant messages drawn from the simulations-measurements comparisons of LW and WINDOW fluxes, and of BTs, are not consistent and appear to be somewhat contradictory, as it is for ex- ample the case in Table 3. Even the authors state (page 15, lines 20-21) that "the MODTRAN spectral resolution impacts the standard deviation of the model-measurements differences, making the results obtained with different AOPs equivalent". More specifically:

Main Comments
1) In general, quite small differences between the 7 examined configurations, consisting in different model setups (Table 2), are found between results obtained without aerosols and with aerosols, as well as between the 6 configurations with aerosols (3 columnar and 3 in-situ). This does not help to draw a clear conclusion on which one configuration and aerosol properties combination is the best, although this is expected to be the main finding of such a radiation closure study.

The study is aimed at investigating how dust particles affect various radiation quantities ( irradiance and BT at the surface and in the atmosphere, BT at the top of the atmosphere) compared to the aerosol-free case and how the magnitude of the dust radiative effect depends on different aerosol optical properties.

This study has been carried out with information on the atmospheric vertical structure and surface characteristics, as well as on the aerosol burden and physical properties, derived from observations. Specific values of the dust complex refractive index, including some recently determined region-dependent values, have been used. The model outputs are then compared with measurements from various instruments installed on different platforms (from the surface radiometers and pyrometer to the airborne radiometers, to the satellite IASI interferometer).

The aim is not the determination of the best combination of aerosol size distribution and refractive index. In fact, we have used the vertically resolved in situ measured size distribution as a reference, since it is directly measured and, in our opinion, best represents the occurring aerosol properties. Under this assumption, we show that the more recent determination of regionally dependent refractive indices perform better than the frequently used literature values.

The use of vertically averaged SDs derived from AERONET with the most commonly used values of RIs (as it is quite frequently done in similar studies) is finalized at assessing the influence they have on the radiation field.

It must be pointed out that the closure is done on a quite large number of radiation measurements, made at the surface, airborne (and at different altitudes), and on satellite. In our opinion this is a quite unique analysis, and the comparison with different types of radiation measurements gives strong constraints and robustness on the results.

It must be also said that unfortunately the atmospheric conditions occurring during the ChArMEx/ADRIMED campaign did not bring large AOD and this aspect, combined with the model and measurements uncertainties, causes the modelled LW irradiances to be equivalent. Nonetheless, we show that this is not true for the WINDOW and the IR zenith BT, for which significant differences are obtained for the *in situ* SD when using OPAC and DB2017 RIs.

For the day with the largest AOD, i.e. 22 June, we also assessed which combination of SD and RI gives the best model-measurement match for the overall set of LW irradiances (downward at the surface, upward and downward components on the ATR-42 and Falcon 20) by calculating the RMSD of all model-measurement absolute differences, and selecting only those AOPs for which the RMSD is below the $\pm 5$ W m$^{-2}$ threshold value. For the AERONET SD all the three RIs meet the requirement (RMSD between 3.2 and 3.3 W m$^{-2}$), while for the *in situ* SD only the DB2017 (RMSD 4.7 W m$^{-2}$). This conclusion has been added at the end of Section 4.2. If we assume that the vertically-resolved *in situ* SD is the best representation of the effective SD, the DB2017 RIs provide the best AOPs.

2) The ascertained/computed differences of each one of 7 configurations with respect to measurements (LW, WINDOW, BTs) mostly fall within the range of uncertainty of measurements, making difficult to decide on which one is really the best configuration.

This is true for the LW and WINDOW irradiances, but not for the zenith BT. We added a sentence at the end of Section 4.1 summarizing the best combination of SD and RI that provide the best model-measurement match: "The final results of the analysis of the surface measurements show that irradiances, either broadband and in the 8-14 μm spectral interval, are not useful to reduce the uncertainty on the dust RI, since the impact of different RIs is below the measurement and model uncertainty. On the contrary, narrowband zenith BT seems to be suitable to constrain the dust RI which better represents the dust optical properties either in moderate and in low dust loading conditions. Under the assumption that the in situ SD is the most representative of the real aerosol dimensions, the DB2017 RI provides the best agreement between model and measurements, either LW and WINDOW irradiances, and sky BT in the two cases where the atmospheric meteorological profiles and the in situ SD are measured down to surface level (22 and 28 June)."

3) A main conclusion drawn from the analysis is that there is a systematic model overestimation of upward LW fluxes within the peak of dust layers, in all 3 days. In other words, there seems to be an inherent problem with the modelling tool, which needs to be assessed.

We believe that a modeling problem is difficult to expect, either because the CLIMAT BT, obtained with the same input parameters as the irradiance components, are very well reproduced, and because for the 22 June case the model succeeds in resolving both irradiances and BTs. We have further investigated this aspect and have tentatively attributed the observed bias to the CGR4 slow time response; in fact, significant model-measurements differences are found when the aircraft velocities during the descents are too high. See details of the answer to point 26 of the specific comments.

4) The estimated small differences between the no-aerosol and aerosol configurations, indicate that the RTM LW computations are relatively insensitive to dust.

That is true but depends mainly on the AOD value. For the 22 June case, (AOD at 500 nm of 0.36) the increase in LW irradiance at the surface due to dust compared to aerosol-free conditions (NOAER) is +4.8, 4.7, 3.3, 10.9, 10.6, and 8.1 Wm$^{-2}$ with COL1, COL2, COL3, INSU1, INSU2, and INSU3, respectively. With the *in situ* SD the dust effect is larger than the uncertainty of LW irradiance measurement (5 Wm$^{-2}$) and of the model (4.2 Wm$^{-2}$).

The values decrease with AOD: indeed, the values for the 28 June case (AOD at 500 nm of 0.21) are +3.2, 1.7, 6.3, 5.1 Wm$^{-2}$ with COL1, COL3, INSU1, and INSU3. This explains why the simulations in aerosol-free conditions agree with measurements within their uncertainties.

For larger AOD, like the case presented in the paper by Meloni et al., 2015, Altitude-resolved shortwave and longwave radiative effects of desert dust in the Mediterranean during the GAMARF campaign: Indications of a net daily cooling in the dust layer, J. Geophys. Res. Atmos., 120, 3386–3407, the perturbation induced by the dust with AOD at 500 nm of 0.59 was 16.2 and 16.1 Wm$^{-2}$ with AOPs analogous to COL1 and INSU1, respectively. In that case the modelled LW irradiance in aerosol-free conditions is outside the expanded uncertainty of the measurements.

5) The reported conclusions are sometimes contradictory. For example in page 17, lines 14-15 it is stated that dust RI from DB2017 produces the best agreement with observations, but this is not supported by and it is not in line with the results of Table 3 where NOAER and COL1 also provide good results, even better than INSU3, if all three parameters, i.e. LW, WINDOW, BT, and three days are considered.

The sentence refers to the results obtained with the *in situ* size distribution. One point that has been clarified in Sections 4.1 and 4.2 is that we can assume that the *in situ* SD is more realistic than the AERONET one, since the first is derived from vertically resolved optical counter measurements covering the diameter range 0.032-32 μm, while the second is derived from surface visible and near-infrared radiance measurements and is representative of the whole atmospheric column. Under this assumption the best model-measurement agreement is obtained with INSU3 AOPs, as stated in page 15, line 32.

6) It is not clear why BTs were computed and are reported only at 3 levels, which sometimes are not collocated with the peaks of dust layers; why similar BT computations were not made at more levels.

The selected altitudes correspond to those witha horizontal flying attitude of the ATR-42. Moreover, while all the other simulated profiles (upward and downward LW irradiances) where obtained with a single model run, the CLIMAT BTs require a run for each altitude. The vertical profiles of the modelled irradiances show a smooth change with altitude, so we believe that BT at few altitudes are sufficient to describe the vertical variations and adding simulations at other altitudes would not provide additional information.

7) The conclusions drawn from the BT analysis are different from those obtained from the analysis of LW fluxes. This is for example the case of the results of profile 42, in Figures 10 and 11. May this point to a possible modelling problem/inconsistency?

The model has proven to perform well either for the LW fluxes and for the CLIMAT BTs on 22 June, so we cannot justify the results for flight F42 with model inconsistency. We also exclude as a cause an incorrect choice of the model input parameters, like the sea surface temperature, emissivity, and/or the atmospheric temperature/humidity profiles, because the CLIMAT BTs are well reproduced at all altitudes. Following a further check on the data, We found that the bias in the LW irradiances occurs where the ATR-42 descent velocity is high. We believe that the associated fast change of ambient air temperature, associated with the relatively long response time of the CGR4 pyrgeometer, may produce the bias. See details of the answer to point 26 of the specific comments.

8) The role of clouds is not reported. Were all the tree days/cases cloud-free? If so, how is this confirmed/ensured? A relevant discussion should be made since the effect of clouds on LW is significant and interplay or even dominate the effect of dust (e.g. possible implications for Fig. 2).

The three flights were carried out under cloud-free conditions.

The sky conditions at Lampedusa were monitored by the TSI-440 sky imager, collecting hemispheric pictures every minute. The absence of clouds during the flights is also confirmed by the time series of downward SW, LW and WINDOW irradiances, and by the information provided by the zenith-looking pyrometer and lidar at the surface. The profiles of downward LW irradiance from airborne radiometers also show that clouds were not detected: indeed the signal due to cloud emission would have been evident in the measurements with positive spikes, like those in Figure 2. A sentence has been added to the description of Figure 2, to highlight the large increase in irradiance/BT due to cloud presence.

Specific Comments

1. Page 1, line 28: define IASI acronym.

The acronym has been defined.

2. Page 4, Figure 1: the AERONET AOD may also be overplotted.

We received a comment from Reviewer #1 saying that we presented many instruments but some of them were not used in the analysis, and he/she suggested to describe only instruments and measurements that were actually used. We used MFRSR AOD measurements because of their higher temporal resolution (about 1 minute) compared to that of the Cimel sunphotometer (about 15 minutes). Moreover, some Cimel data are missing because of some malfunctioning of the instrument.

However, di Sarra et al., Empirical correction of MFRSR aerosol optical depths for the aerosol forward scattering and development of a long-term integrated MFRSR-Cimel dataset at Lampedusa, Appl. Optics, 54, 2725-2737, 2015, compare MFRSR and AERONET AOD measurements, deriving a mean bias in AOD not larger than 0.004 and a root mean square difference ≤0.031 at all wavelengths. Plotting the available AERONET AOD with the MFRSR AOD would require some discussion. Since the paper is already long we prefer not to add the AERONET AOD measurements.

3. Page 4, line 18: the reported angstrom exponent is high, it is about the maximum one; give a more realistic value (range).

During the first days of the campaign (17-20 June) the Ångström exponent is between 0.3 and 1.75. The text has been modified accordingly.

4. Page 5, lines 32-35: why? Please explain.

Figure 1 of the paper by Gröbner, J., Wacker, S., Vuilleumier, L., and Kämpfer, N.: Effective atmospheric boundary layer temperature from longwave radiation measurements, J. Geophys. Res., 114, D19116, doi:10.1029/2009JD012274, 2009 is very explicative. 99% of the downward LW irradiance comes from the lowest atmospheric layers, those where most of the water vapour is concentrated. The atmosphere is nearly transparent in the 8-14 μm spectral range, and the radiation in the interval is emitted from the upper layers.

5. Page 5, lines 35-37, "The pyrometer . . . for the IRP BT": this sentence is oversimplified. A quick look at the 3 figures reveals significant differences between BT and irradiances. For example, what happens in June 24 and 25 (when LW-WINDOW curves do not have peaks, opposite to IRP BT)? What about the role of temperature and clouds?

The pyrometer has a narrow field of view (2.6°), the broadband CGR4 has 180° and the CGR3 has 150° FOV. So the pyrometer is able to detect each single cloud entering its FOV, while the same single cloud can have a minor impact on the LW-WINDOW irradiance measured by radiometers with broad FOVs if the rest of the sky is cloud-free. That is the reason why the IRP BT time series has more peaks than irradiances time series.

6. Page 7, line 7: define FWHM acronym.

The acronym has been defined in the revised text.

7. Page 7, line 30: up to which altitudes? How much the use of standard profiles can affect the radiative fluxes? Was any sensitivity study performed to assess this? Especially the LW fluxes should be sensitive.

The standard mid-latitude profiles have been used above the maximum altitude of the radiosounding on 28 June and 3 July, i.e. above 32 km and 26 km, respectively. The surface, as well as the profiles, irradiances and BT are not sensitive to variations of upper level atmospheric profiles.

In the revised manuscript a sensitivity study of the modelled quantities has been performed with respect to the main parameters affecting infrared radiation, either in aerosol-free conditions (i.e. IWV, temperature profile, SST, and surface emission) and with aerosol (we have tested the sensitivity to AOD and to the imaginary part of the dust refractive index).

An increase of 0.3 K in the temperature profile causes a 2.2 $Wm^{-2}$ increase in downward LW irradiance at the surface, decreasing for increasing altitudes (becoming 1.3 $Wm^{-2}$ at the ATR-42 top altitude of 5.7 km, and 0.6 $Wm^{-2}$ at the Falcon 20 altitude of 10 km). The upward LW irradiance increases by 1.8 $Wm^{-2}$ at 5.7 km, by 1.5 $Wm^{-2}$ at 10 km, and by 1.4 $Wm^{-2}$ at the TOA.

8. Page 7, line 33, regarding the absorbing gases: similarly, it would be worth to discuss/assess the sensitivity of fluxes to these parameters, especially given the scaling applied to their vertically distributed values.

The sensitivity of the irradiance to changes in the integrated water vapour has been tested by increasing the average measured value by its uncertainty (+0.2 mm), which is of the same order of magnitude of the IWV standard deviation within the considered time interval.

The downward LW irradiance increases by 0.4 $Wm^{-2}$ at the surface, by 0.1 $Wm^{-2}$ at 5.7 km, and by 0.06 $Wm^{-2}$ at 10 km, while the upward LW irradiance at the TOA decreases by 0.1 $Wm^{-2}$.

The sensitivity to other absorbing aerosols has not been tested.

9. Page 8, about ECMWF: The use of reanalysis data is inevitable in this case. However, an assessment of the induced uncertainties associated with their coarser resolution could be made by comparing similar ECMWF data with available measurements for the other two days. This could provide an estimation of induced uncertainties in June 22.

As suggested by the reviewer, the ECMWF profiles at 12:00 have been compared with the temperature and relative humidity profiles measured by the radiosonde and by the airborne meteorological instruments. The T profile is well reproduced on both days, although the ECMWF profiles does not capture the fine vertical structures due its vertical resolution. An exception is the atmosphere below 4 km on 28 June, where the profile sounded by the ATR-42 presents lower temperatures compared to that of the ECMWF profile. The RH profile of the ECMWF operational model generally follows the measurements, but differences can be large at certain levels, like around 3 km on 3 July.

[Figure]

10. Page 8, line 10: a few words about the measured aerosol properties and the identified aerosol layers can be added. For example, apart from the layers and their extension neither information is given nor reference is made to the type of aerosols in each layer, with reference to corresponding measurements that cloud provide this kind of information.

A short discussion on the aerosol stratification identified by the airborne measurements and by the airmass back trajectory analyses has been included in Sections 3.1 and 3.1.1. A detailed analysis is presented in the paper by Denjean et al. (2016).

11. Page 8, line 14: so, what values of emissivity were assumed in the study? Do they differ and how much from day to day.
The values are the same for all the three cases because surface wind speeds are similar. A sentence has been added in the text to better explain, and some spectral values are included.

12. Page 8, Figure 4: the quality should be improved, e.g. by thicknenning the curves, so that the coloured curves can be more easily discerned.
The quality of the figure has been improved.

13. Page 8, line 29: As mentioned, different factors influence and differentiate the AERONET and in-situ SDs, one important being their different value, .e. columnar versus vertically resolved. The value of detailed measurements is that they provide vertically resolved SDs. Therefore, emphasis should be given to them. Discuss a bit more how the measured SDs differ to AERONET ones, referring to their agreement and disagreement. For example, larger differences appear in June 22 than in July 03. Refer to this difference referring to the nature of vertical profiles of Fig. 3 and the type of aerosols that are present in the different layers of every daily profile.
Table S1 has been produced as Supplement Material with the median radius, standard deviation, and normalized number concentration for each mode of either the AERONET and the *in situ* log-normal size distributions for the layers identified in the three cases. The differences in SD among the various layers have been discussed further, as suggested by the reviewer, and related to the transport pathways and to the mixing of dust with pollution particles.

14. Page 9, line 19: explain why this choice of water soluble RI was made and not any other.
Polluted maritime aerosol is the most probable aerosol type characterizing the lowest atmospheric layers over Lampedusa. This is supported by the analysis of the airmass back trajectories in the boundary layer, showing airmasses originating from Europe or recirculating within the Mediterranean basis, and from the chemical analyses carried out on the aerosol samples (Denjean et al., 2016). Water soluble is one of the components of the maritime aerosol, either clean and polluted, according to the OPAC definitions (see Hess et al., 1998). The other components are sea-salt and soot (on for the polluted maritime). Among the components, water soluble and sea salt have similar values of the imaginary part of the RIs below 8 µm and above 11 µm, while the water soluble is more absorbing than sea salt in the 8-11 µm interval. The RI of soot is too absorbing to be representative of the average aerosol. For these reasons the water soluble RI has been chosen.

15. Page 9, lines 20-26: Table 2 is not discussed enough. It should be said more clearly what exactly has been done and how the Mie-based computations of AOD compare to AERONET ones, whenever applicable, i.e. in visible wavelengths.
The combination of the SDs and RIs for the three flights has been better clarified in a new version of Table 2, which includes three tables (one for each flight), describing the SD and RI used in each aerosol layer identified by the ATR-42 and lidar vertical profiles. Moreover, the choice of the different RIs has been better explained in Section 3.1.1.
The spectral AOPs (extinction coefficient, absorption coefficient) accepted by MODTRAN are all referred to the extinction coefficient at 550 nm, for which the vertical distribution, $ext_{550}(z)$, is derived from the lidar backscatter profile and AOD measurements.
So the AOD at 550 nm is fixed, and corresponds to the value obtained from the MFRSR measurements at 500 and 868 nm using the Ångström law. AOPs are allowed to differ in no more than four aerosol layers in the troposphere.
We compute the spectral AOPs from Mie theory for a single particle in a wavelength grid from 2 to 100 µm and including 550 nm, and then divide them by the calculated extinction coefficient at 550 nm. So we have:

$$ext_\lambda(z) = \frac{ext_\lambda(z)}{ext_{550}(z)}$$

$$abs_\lambda(z) = \frac{abs_\lambda(z)}{ext_{550}(z)}$$

This ensures that the values of the AOD at 550 nm remain constant, whatever the AOPs.

16. Page 10, lines 16-17: does this refer to July 03? In Table 3 no results for INSU2 are displayed.
The sentence originally referred to 22 June. Section 4.1 has been modified in order to be clearer for the reader.

17. Page 10, lines 19-20: why the stronger infrared emission? Is it a matter of larger mass? Please explain.
For a particle with diameter D the absorption (and emission according to Kirchhoff law) and scattering properties are calculated with the Mie theory if D is of the same order of magnitude of the wavelength, as is the case for dust particles in the infrared spectral region. According to Mie theory, the absorption and scattering efficiency ($Q_{abs,ext}$) depend on the complex refractive index and on D and the absorption and scattering coefficient ($\sigma_{abs,ext}$) are proportional to Q and to the particle's cross section $\pi D^2$. Thus increasing the particles' dimension increases the absorption and emission coefficient.

18. Page 10, line 24: clarify that "all cases" refer to LW, WINDOW and IR BT.
"all cases" refers to the three days.

19. Page 10, line 34: here it should be clarified what is exactly the spectral interval/coverage of the measurements (IRP). This not clear based on what is said in page 7, line 6, about the IRP centered at 3 wavelengths etc. It is essential to clarify what is exactly the spectral coverage of measurements since they are used as the reference to which the simulations are compared, and given the significant sensitivity of theoretical computations to the spectral interval. Also explain why the reduction in WINDOW irradiances has different magnitude despite the same spectral reduction (0.4 microns) in different spectral parts.
The infrared pyrometer (IRP) measures the zenith BT in the 9.6-11.5 µm interval, while the CLIMAT measures the nadir BT in three infrared channels centred at 8.7, 10.6, and 12 µm with about 1 µm full width at half maximum. The sentence on page 10, line 34, refers to the WINDOW irradiance, measured by the PMOD/WRC CGR3 modified pyrgeometer, which is sensitive to the radiation in the 8-14 µm spectral interval.
The sensitivity analysis carried out on the WINDOW irradiance shows that reducing the spectral interval by a small amount (0.4 µm) significantly reduces the downward irradiance, and the reduction depends on where in the spectrum the reduction is operated: this is caused by the asymmetry of the irradiance spectrum with respect to the centre of the interval.

20. Section 4.1: what is missing is a critical approach providing insight into possible physical reasons for better agreement between the 7 examined cases. A quite exhaustive and very detailed description of results is made, referring to various numbers (Table 3). This is not enough while it turns to be confusing to the reader. What is more important is to determine which set of AOPs is more efficient and compared better to the measurements for the 3 cases. The discussion should conclude on this, stating at least if there is a "best" choice or if there is not and why. Moreover, in both cases, the discussion should provide a physical basis for the outcome of the analysis and the closure of Table 3. For example, a summary of the results of Table 3 should point to NOAER being the most efficient simulation, providing better results than the other 6 sets of AOPs in 4 cases (out of totally 9, i.e. 3 days by 3 parameters). NOAER is followed by COL1 (3 cases with best performance) and INSU3 (2 cases). So, questions may arise, like why simulations without aerosols

should be more appropriate/realistic, or why INSU3, which may be expected to be the most realistic, is finally not.

Section 4.1 has been modified to better present the results and the conclusions. We state that for the conditions met during the campaign, with moderate and not really large AOD, the surface irradiances are not useful to reduce the uncertainty on the dust RI, since the impact of different RIs is below the measurement and model uncertainty. This also explains why NOAER simulations agree with measurements within their respective uncertainties. On the contrary, narrowband zenith BT seems to be suitable to constrain the dust RI which better represents the dust optical properties either in moderate and in low dust loading conditions, assuming that the *in situ* airborne measurements better describes the local aerosol distribution. Indeed, in the two cases where the atmospheric meteorological profiles and the *in situ* SD are measured down to surface level (22 and 28 June) the zenith BT is well reproduced with the DB2017 RIs.

21. Page 12, line 3: as to upward LW, authors may want to comment on why the smallest differences are for COL1 in Table 5, while the smallest RMSDs in Table 4 are for INSU1.

Tables 4 and 5 show that the upward LW irradiance at Falcon 20 and ATR-42 altitudes is reproduced within model and measurement uncertainties with all AOPs, including the NOAER case. The fact that COL1 AOPs give the best match with Falcon 20 measurements while INSU1 AOPs provide the best agreement with the ATR-42 ones may be attributed to different reasons: among them, the Falcon 20 passage is not exactly simultaneous with the ATR-42; the Falcon 20 simulations may be affected by the ECMWF temperature/humidity profile above 6 km, while the ATR-42 upward irradiance rely on the *in situ* measurements of the meteorological vertical profiles.

22. Figure 7: why only points for NOAER, COL1 and INSU3 are given and not for the other cases? All these appear in Table 6.

Overlapping the points corresponding to all AOPs would have made the Figure 7 very difficult to read. After the reviewer's suggestion, we have considered that presenting the results of COL3, instead of COL1, may be better to show the dust perturbation compared to NOAER and the effect of changing the SD, but not the RI, compared to INSU3. The estimated uncertainties on the calculated BTs are also shown.

23. Page 12, lines 15-16: add "in-situ" before SD. This sentence needs to be re-written, since it is introduced all suddenly without being given evidence and discussed based on the results of Fig. 7.

The sentence has been modified as follows: "These results show the better sensitivity of BT to dust optical properties than broadband irradiance. When considering that *in situ* SD better represents the local aerosol distribution, the DB2017 RI from Algeria and Morocco provide the best model-measurement agreement".

24. Page 12, lines 17-19: while discussion is made no results are shown/given.

This comment is not clear. Figure 6 for the upward LW irradiance and Figure 7 for the CLIMAT BTs show that model simulations with and without aerosols are overlapped below 4 km, and differ at higher altitudes, where the aerosol effect is discernible. Moreover, the differences in BT due to dust compared to the aerosol-free simulations are provided on page 12, lines 19-20.

25. Section 4.2.1: A quite exhaustive discussion is made making frequent reference to numbers that differ a while between the 6 examined cases. Also the question arises why NOAER sometimes performs equally or better than dust-including cases. It could point to potential artifacts due to counteracting effects of other parameters than aerosol, which affect the LW radiation transfer and BT.

The limited aerosol effect is due to the moderate AOD measured during the flights and not to modelling problems, and this is the reason why in some cases (more often for the irradiance but not for the BT) the NOAER simulations agree with measurements. This aspect has been remarked in the revised manuscript (Sections 4.1, 4.2.1, and in the conclusions).

26. Page 12, lines 35-36, "These differences . . . airborne instrumentations": so is there an inherent problem with the model?
We cannot state that there is problem with the model, because the same input parameters allow to fairly reproduce the CLIMAT BTs. Moreover, the LW irradiance is well reproduced below 2000 m and above 4000 m and at the Falcon20 altitude.  As discussed, a possible bias in the CGR4 measurements may be found when the air temperature is fastly changing, due to inhomogeneities in the instrument temperature. For example the ATR-42 path (Figure 4b) shows a steep aircraft fall from 3.5 km to 1.2 km. A rapid decrease in altitude, with a consequent increase in temperature, may be not registered by the pyrgeometer, which has 6 seconds response time (1/e), but needs a longer time to establish equilibrium with the ambient air temperature (as also found by Albrecht et al., Pyrgeometer measurements from aircraft, Rev. Sci. Instrum., 45, 33–38, 1974). The same problem may have affected the irradiance measurements on 3 July. This may also explain the slight overestimation of the downward LW irradiance by the model in the same altitude ranges where the upward component is overestimated. The CLIMAT BT measurements is not expected to suffer from the same problem because the response time of the instrument is much faster (about 160 ms).

27. Page 14, line 20" add "was" before "evaluated".
Done.

28. Page 14, line 21, "resampled": how it was done?
IASI spectra have been averaged in 15 cm$^{-1}$ intervals. The first interval is centred at 652.5 cm$^{-1}$ and includes the BT values between 645.0 and 660.0, the second interval is centred at 667.5 cm$^{-1}$, and so on.

29. Page 14, lines 32-37: why there is difference on what provides the best match with reference to best match with the measured spectra and BTs?
The IASI spectra used in the comparison with the model are averaged over a region, instead of instantaneous measurements like those from CLIMAT.  Moreover, IASI and CLIMAT measurements are not simultaneous.  Finally, measurements are made over different spectral intervals, and the AOPs producing the best agreement with the mode result may somewhat differ. This has been stated in the text.

30. Page 15, lines 2-3: this is not applicable to 780-980/cm for June 22 and 28.
The sentence states that when the AOD is sufficiently large the aerosol perturbation to the BT simulated in aerosol-free conditions is significant, so this does not apply for the 28 June case. About 22 June, this applies clearly to the 1070-1200 cm$^{-1}$ interval for different AOPs, and only for the COL1 combination in the 780-980 cm$^{-1}$ interval.

31. Page 15, lines 20-22, "In our case, . . . AOPs equivalent.": what exactly is it meant by this? By which means. Please explain. Is it implied that this (having very high resolution)  is preferable? If so, why? If valid, it would mean that AOPs are not important for accurately computing LW radiation and dust LW radiative effects. Is this the meaning?
The sentence refers to the fact that MODTRAN resolution (0.1 cm$^{-1}$) is lower than that of the IASI measurements and of the σ-IASI-as radiative transfer model (0.01 cm$^{-1}$), so we do not expect model-measurements differences as low as those found by Liuzzi et al. (2017). The sentence has been eliminated and text has been modified, also to answer to Reviewer #1, as follows: "The

limitation in the MODTRAN5 resolution does not allow to reproduce the high resolution IASI spectra: however, the scope of simulating the IASI measurements is to show that TOA BTs are sensitive to the dust presence and to the various AOPs, and that they can be reproduced with the same input parameters that allow to simulate irradiance and BT at the surface and in the atmosphere.".

32. Page 15, line 28, "The combination . . . downward": this is not clearly evidenced in the discussion of sections 4.1 and 4.2.
The sentence has been moved at the end of Section 4.2.

33. Page 26, Table 2: the Table needs further/better explanation, it is not very easy for the reader to understand what exactly is the information given in this Table.
We agree with the reviewer. A new Table 2 has been prepared, which includes three tables, one for each day. The tables present the combination of SD and RI used in each layer identified by the lidar and ATR-42 measurements, and the AOD value at 8.6 µm.

34. Page 37, Figure 5: what have been the criteria for the design of flight paths.? Nothing is said about this and deserves to be mentioned in the text.
This aspect has been treated in Section 2.1(Aircraft strategy) of the paper by Denjean et al. (2016). In particular the following text explains the flight strategy: "The general flight strategy consisted of two main parts. First, profiles from 300m up to 6 km above sea level (a.s.l.) were conducted by performing a spiral trajectory 10–20 km wide to sound the vertical structure of the atmosphere and identify interesting dust layers. Afterwards, the identified dust layers were probed by straight levelled runs (SLRs), where the aircraft flew at fixed altitudes, to provide information on dust spatial variability and properties. Horizontal flight legs in the dust layers lasted 20–40 min to allow aerosol collection on filters for chemical analyses in the laboratory".
The sentence "The ATR-42 sounded the atmosphere during profile descents and ascents to infer the vertical structure and composition and identify layers with different properties" in Section 2.2 clarifies the criteria beyond the flight paths.

35. Page 44, Figure 12: wavelengths could be added, e.g. on the top x-axis.
The top x-axis has been expressed in wavelength units, as suggest by the reviewer.

---

## Referee Report (RR1)

Review :
The paper is now better organized and much more clear. They are still some minor comments I have listed below before the manuscript should be accepted for final publication.

- P10l18-19: *"The AOD at 8.6 μm is calculated from the MFRSR AOD at 500 nm and the ratio between the extinction coefficient at 8.6 μm and 500 nm obtained from the Mie calculations* " : how is obtained the extinction coefficient at 500 nm since the refractive indices of DB2017 are given only in the 2-16 μm interval?
- P1425: *"The agreement is good for both downward and upward LW irradiances with the AERONET SD, and is best with COL1 and COL2 (RMSD of both components 4.1 W m-2).* ": I don't understand why COL3 is not the best? The RMSD of both components is lower than 4 W m-2
- P14 21-25: for my opinion, this paragraph should be placed either in the previous paragraph where irradiance results are analysed or in the section 4.4 where the configurations used for the radiative forcing estimation are defined.
- P20l12: *"the scope of simulating the IASI measurements in this work is to show that TOA BTs are sensitive to the dust occurrence and to its AOPs"*: since results are important but not new, see for example 2 references given below : Capelle et al., 2014 and Vandenbussche et al., 2013.
- Fig. 4: The AERONET size distribution for 22 June appears to present 3 modes and not 2 and therefore doesn't correspond to values given in Table S1
- Table S1 : Where come from the size distribution parameters for AERONET since they didn't apparently come from the AERONET website (where only two modes are retrieved).

Capelle, V., Chédin, A., Siméon, M., Tsamalis, C., Pierangelo, C., Pondrom, M., Crevoisier, C., Crepeau, L. and Scott, N. A.: Evaluation of IASI-derived dust aerosol characteristics over the tropical belt, Atmos. Chem. Phys., 14(17), 9343–9362, doi:10.5194/acp-14-9343-2014, 2014.

Vandenbussche, S., Kochenova, S., Vandaele, A. C., Kumps, N. and De Mazière, M.: Retrieval of desert dust aerosol vertical profiles from IASI measurements in the TIR atmospheric window, Atmos. Meas. Tech, 6, 2577–2591, doi:10.5194/amt-6-2577-2013, 2013.

---

## Referee Report (RR2)

General comments: This paper lies in the framework of the ChArMEx/ADRIMED experiment, that took place in the Mediterranean in summer 2013. Three vertical profiles of atmospheric and aerosol properties, made at Lampedusa in conjunction with surface, airborne and satellite IR broadband and narrowband radiation as well as radiosonde are analyzed in order to 1) identify the sensitivity of the different radiative measurements to mineral dust microphysical properties (size distribution and refractive index) and 2) analyze their impact in term of radiative forcing. The main result of this study is that if LW irradiance is poorly sensitive to aerosol microphysical properties compared to brightness temperature, the IR dust radiative forcing is non-negligible, and strongly depends on size distribution (SD) and refractive index (RI). This study highlights the importance of a precise knowledge of the dust microphysics to infer correctly their radiative effect. The paper is an interesting sensitivity study of the radiative variables to the aerosol microphysics, leading, in particular, to the conclusion that spectrally resolved measurements of brightness temperature is more fitted to infer dust properties than broadband LW irradiances. However, the part that concludes on the most appropriate refractive indices is less convincing. Such a study would require a more detailed analysis of the differences between refractive indices (at minimum a figure displaying their values in the spectral domain concerned), as well as a more exhaustive variability in the choice of the indices. Here among the three indices used, two indices are quite similar and only one coming from recent measurement campaign of DiBiagio et al., 2017 is really different. Moreover, the study, based principally on RT simulations, lacks of discussions on the uncertainties due to the RT model itself, as well as to the different hypothesis used. In particular, which is the impact of an error in surface temperature or surface emissivity? An error on the water vapor profile? No reference error is calculated under clear sky condition for example, to distinguish error directly due to the model from errors due to the impact of aerosols properties. The resulting biases obtained with the different aerosol properties configurations cannot therefore be really discussed. For example, large biases between simulated and calculated irradiances are not explained (and apparently not due to wrong aerosol properties), implying that something is missing in the RT model, but not enough discussed. The section on IASI data is not enough developed. All the spectra within a box of about 100 kmx100 km are averaged before analysis, causing a standard deviation in the averaged spectrum larger than the effect of the aerosols properties analyzed! Here again, since no reference errors are given, biases obtained from the different aerosol configurations are finally equivalent and it's not possible to state on the best configuration. This part doesn't really bring new information compared to the previous sections or previous studies, or required a more precise development. Finally, some details on the inputs used are missing. A few details are provided on the size parameters used (the reader has to refer to the paper of Denjean et al. 2016 to have the precisions) and the exact refractive index from DiBiagio et al., 2017 use in this study is not given: 3 different indices are coming from ´n¢athe source regions (Tunisia, Algeria, Morocco) in DiBiagio et al., 2017 whereas only one is used here without any precision!

We agree with the reviewer that a comprehensive analysis including the impact of the uncertainty on the input parameters on the model simulations (of either irradiances and brightness temperature profiles and of radiances at the top of the atmosphere) is useful to better constrain the results. Following the reviewer's suggestion, a sensitivity study addressing the uncertainties of the modelled radiation quantities due to the uncertainty on the input parameters has been carried out, either in aerosol-free conditions or including the aerosol particles. The main model input parameters

affecting infrared radiation in aerosol-free conditions that have been considered are: integrated water vapour, temperature profile, sea surface temperature, surface emissivity. Each quantity has been perturbed one at a time by the amount of its uncertainty, than all the resulting model uncertainties has been combined to provide the overall uncertainty. Similarly, the sensitivity with respect to AOD and dust complex refractive index has been quantified.

The results have been added to the manuscript under the new Section 3.3.

The model LW irradiance uncertainty decreases with increasing altitude for both the downward and upward components. The estimated model uncertainty on the downward and upward LW irradiances at the surface is 2.2 and 2.0 W m$^{-2}$ for simulations without and with aerosols, respectively. At the Falcon 20 altitude (about 10 km) the uncertainties are 0.6 and 1.5 W m$^{-2}$ for the downward and upward component, respectively, for both simulations with and without aerosol. The upward LW irradiance uncertainty is 1.4 W m$^{-2}$ at TOA, for simulations with and without aerosol.. The estimated uncertainty on ARF is obtained by the combination of the above values, and is 4.2 W m$^{-2}$ at the surface and 2.0 W m$^{-2}$ at the TOA. The uncertainty on AHR is largest at about 4.5 km altitude (0.030 K day$^{-1}$ with aerosol and 0.026 K day$^{-1}$ without aerosol), and close to the surface (0.050 K day$^{-1}$ with and without aerosol).

The uncertainty on the downward WINDOW irradiance is 0.9 and 0.6 W m$^{-2}$, with and without aerosol, respectively. The estimated uncertainty on the modelled zenith BT is 0.7 and 0.3 K, with and without aerosol, respectively.

The aerosol-free CLIMAT BT is much sensitive to SST and surface emission, with slightly larger values at 600 m (0.3 K) than at 5670 m (0.28 K). The overall uncertainty for the case with aerosol is 0.31 K at 600 m and 0.37 K at 5670 m.

The uncertainty on the spectral BT at TOA in the atmospheric window varies between 0.25 and 0.29 K in aerosol-free conditions, and between 0.32 and 0.50 K with aerosol.

The model-measurement differences have been discussed in the text taking into account the uncertainties on the model estimates.

A figure displaying the spectral complex refractive indices used in this study has been added as Supplement Material (Figure S1). With this regard, a summary of the most common complex refractive indices of desert dust is provided in Di Biagio, C., Boucher, H., Caquineau, S., Chevaillier, S., Cuesta, J., and Formenti, P.: Variability of the infrared complex refractive index of African mineral dust: experimental estimation and implications for radiative transfer and satellite remote sensing, Atmos. Chem. Phys., 14, 11093-11116, doi:10.5194/acp-14-11093-2014, 2014. Moreover, the spectral (0-40 µm) normalized extinction coefficients, single scattering albedoes, and asymmetry factors computed using the combination of SDs and RIs described in the text for each layer of the three profiles have been shown in Figure S2, S3, and S4 of the Supplement Material.

The dust refractive indices that we use in the 0-40 µm range have been chosen because they are specific for the source regions found during the ChArMEx campaign (like those from Tunisia, Algeria, Morocco by Di Biagio et al., 2017), or because they are widely used in the retrieval of satellite products or in climate models (like the ones from OPAC by Hess et al., 1998, and by Volz, 1973).

OPAC and Volz (1973) have very similar real and imaginary parts, except for the 9.5-14 µm spectral interval, where we explore the dust impact on the surface irradiance (in the 8-14 µm window) and brightness temperature (9.6-11.5 µm) and in the CLIMAT and IASI brightness temperatures (BTs). The results in Table 3 of the manuscript confirm that the model-measurement differences in the 9.6-11.5 µm BT can be significant (0.7 K) when the OPAC size distribution is used, while are modest (0.4 K) with the *in situ* size distribution.

Other dust refractive indices found in literature were taken into account, although the results are not reported in the manuscript: for example, the Volz (1972) one, which is equivalent to the one

published in Shettle and Fenn (1979). The imaginary part is much lower than that of OPAC and Volz (1973), so lower surface LW and WINDOW irradiances and infrared BT are expected, with consequently modest radiative effect.

The aerosol optical properties obtained with the Longtin et al. (1988) dust refractive index were already examined in a previous paper (Meloni et al., Altitude-resolved shortwave and longwave radiative effects of desert dust in the Mediterranean during the GAMARF campaign: Indications of a net daily cooling in the dust layer, J. Geophys. Res. Atmos., 120, 3386–3407, 2015).

All the three refractive indices by Di Biagio et al., 2017 (Algeria, Tunisia, Morocco) have been used in the study, since the analysis by Denjean et al. (2016) based on back-trajectories and MSG-SEVIRI satellite products shows that dust collected during F35, F38, and F4 flights have different source regions (details are given in Table 1 of Denjean et al., 2016). More specifically, for flight F35 the dust layer above-3.5 km originated from southern Algeria, while the dust layer between 1.5 and 0.5 km was transported from southern Morocco. On 28 June (F38) dust was lifted from Tunisia. Finally, on 3 July (F42) dust originated from Tunisia (above 3 km) and from southern Morocco (below 3 km).

A better description of the choice of the appropriate refractive index on the base of the dust source region was added in Section 3.1.1 (page 9, lines 1-19). We have used different refractive indices from Di Biagio et al. (2017) for each flight and each dust layer based on the source regions found in Denjean et al. (2016). For F35 we used the refractive index for Algerian dust in layer 3 (see Figure 3), that for Moroccan dust in layer 2, and the OPAC water soluble refractive index for layer 1, i.e. below the dust layer. Similarly, the Tunisian dust refractive index is used for F38 flight in layer 2 and the OPAC water soluble one in layer 1. For F42 the Tunisian and the Moroccan dust refractive index are used in layer 2 and 1, respectively.

We clarified the choice of the dust refractive index in Section 3.1.1 and prepared a new Table 2 which includes three tables, one for each day. The tables present the combination of SD and RI used in each layer identified by the lidar and ATR-42 measurements, and the AOD value at 8.6 μm.

Specific comments:
- p2line 21¢ a: ´n¢aMost of these studies have been carried out close to the dust source regions and did not take into account the possible modifications in dust optical properties during long range transport. ": You can also find some studies dealing with the variation of the aerosol properties with transport (e.g. Maring 2003; Ryder et al., 2013; Weinzierl et al., 2017 and so on). Moreover, I don't see the link with the subject of this paper since there is no discussion on the possible change of dust properties with transport. Maring, H.: Vertical distributions of dust and sea-salt aerosols over Puerto Rico during PRIDE measured from a light aircraft, J. Geophys. Res., 108(D19), 1–11, doi:10.1029/2002JD002544, 2003. Ryder, C. L., Highwood, E. J., Lai, T. M., Sodemann, H. and Marsham, J. H.: Impact of atmospheric transport on the evolution of microphysical and optical properties of Saharan dust, Geophys. Res. Lett., 40(10), 2433–2438, doi:10.1002/grl.50482, 2013. Weinzierl, B., ProsPero, J., Chouza, F., FomBa, W., Freudenthaler, V., GasteiGer, J. and Toledano, C.: THE SAHARAN AEROSOL LONG-RANGE TRANSPORT AND AEROSOL–CLOUDINTERACTION EXPERIMENT Overview and Selected Highlights, [online] Available from: http://journals.ametsoc.org/doi/pdf/10.1175/BAMS-D-15-00142.1

We thank the reviewer for suggesting the papers that have been integrated in the Introduction, citing them as example of studies carried out by means of aircraft measurements dealing with the temporal evolution of dust properties occurring during long-range transport.

Our study focuses on the optical properties of transported dust and on its infrared radiative effect in the Mediterranean. The present paper does not aim at assessing how dust optical properties change during transport (Denjean et al., 2016 show indeed that the coarse mode of dust did not change after 5 days of transport possibly due to strong vertical turbulence within the dust layer, preventing the deposition of large particles), but highlights the importance of knowing the dust microphysical and

optical properties to reasonably estimate the IR dust radiative forcing and heating rate at the surface, in the atmosphere, and at the top of the atmosphere.

- P2line 29:
the paper Sellitto et al., 2016 deals with aerosols in the ULTS and not with dust. This reference is not really appropriate here.
The reference has been removed.

- p4line21: why describing AERONET AOD and associated uncertainties if not used? Even not compared in the following to the MFRSR?
We agree with the reviewer. The AERONET AOD is not used because of some missing data, so the sentence in lines 22-23 about AOD uncertainty has been removed.

- P5: For the surface observations, several instruments measuring irradiance are described, but it's not clear if they are all used in this study. Why describing every instrument available if they are not used? Which ones are really used? This section would gain in clarity if simplified.
In Section 2.1 all the ground-based instruments are presented. For sake of clarity, we removed all instruments (like the shortwave radiometers and the pyrheliometer) whose measurements are not used in the analysis.

- P6line13: maybe a summary of the description of the meteorological and dust conditions given in Denjean et al., 2016 would help? It's easier for a reader to get all the relevant information in one paper.
A short description of the synoptic conditions causing dust transport from the Sahara desert to Lampedusa during the campaign has been added in the text.

- P7line15: unity switched from µm to cm-1. Maybe it would be clearer for the reader to stay in µm?
IASI spectral characteristics, like spectral interval, sampling and resolution, are provided in units of $cm^{-1}$. We have used both µm and $cm^{-1}$ in the text and in figures and table whenever possible to help the reader.

- P8line 18-19: How the AOP (i.e. spectral extinction, single scattering albedo, and phase function at each layer) can be derived from AERONET observation and from Deanjean et al., 2016, in particular for longwave? Observations made by AERONET or Denjean et al. are made in visible wavelength, not in the infrared part of the spectrum. Optical properties cannot be derived in the longwave by these measurements. This sentence is in contradiction with the procedure described later where the size distribution from AERONET and the ATR-42 are used with independent refractive indices to derived these optical properties.
We agree with the reviewer that the sentence in lines 18-19 is misleading. The IR AOPs are not derived from AERONET or airborne observations alone.
The aerosol optical properties in the infrared spectral range are calculated applying the Mie theory using the AERONET and the *in situ* size distributions and the complex refractive indices in the 3-40 µm (OPAC) and in the 2-16 µm (Di Biagio et al., 2017) intervals.
The sentence has been rephrased.

- P9line 14-15: which is the RI used from OPAC? MITR? Must be cleared. Similarly, which RI from DB2017 is used? In Table 2 "Algeria-Tunisia-Morocco dust" is mentioned, but it corresponds to 3 different indices, which one is used?

We used the mineral dust refractive index from OPAC, which is the same for the four dust types (accumulation, coarse, nucleation, transported) of the model. This has been clarified in the text. As for the answer to the reviewer's general comments, we have used different refractive indices from Di Biagio et al. 2017 for each flight and each dust layer based on the source regions found in Denjean et al. 2016. More details are now given in section 3.1.1 and Table 2.

- P9line 19: the sentence "the AOPs are calculated using the AERONET and the in situ SD and the OPAC water soluble RI" have to be rewritten by something like "the AOPs are calculated using either the AERONET or the in situ SD and the OPAC water soluble RI.
We rephrased the sentence according to the reviewer's suggestions.

- P9 line 4: A table given the main parametrization of the SD (radius and width of the distribution size) used would avoid to refer systematically to the paper of Deanjean et al., 2016.
We have produced Table S1 with the median radius, standard deviation, and normalized number concentration for each mode of either the AERONET and the *in situ* log-normal size distributions for the three cases as Supplement material.

- P10 Eq 2: What does the symbol Delta stand for?
The Delta symbol in the heating rate equation represents the variation of net flux and pressure between two contiguous layers.

- P10-11
and Table 3: the uncertainty for the WINDOW irradiance have been changed from 2 to 6 W.m2 from the previous version of the paper. In any cases in the text I read an uncertainty of 3W.m2 (p5, line 19). Which is the good one? In addition, the observed values for the downward irradiance and BT correspond to the average over a 10 minutes interval, what is the standard deviation of the measurement compared to the uncertainty? If the standard deviation is of the order of uncertainty, it means that the signal present small variation within the 10 minutes and maybe it would be better to compare simulations with observation, for every observation within the 10 minutes and average after instead of comparing with the average observation? If the standard deviation is larger, it means that using constant aerosol distribution is not valid.
The measurement uncertainty reported in Table 3 is the expanded (2-sigma) uncertainty, but this was not specified in the table caption, so it seems to disagree with what explained in the text (page 5, lines 19-20), referring to $\pm 3$ Wm$^{-2}$ as one sigma uncertainty. This has been better specified in the revised paper.
However, the CGR3 participating to the ChArMEx campaign has been recently tested by PMOD/WRC to assess the possible effect due to the leakage of solar radiation on the WINDOW irradiance. The tests have shown that the effect is negligible, and so the WINDOW irradiance data used in the present analysis have to be reconsidered because they were corrected by subtracting a shortwave stray-light correction of about 4 Wm$^{-2}$ per 1000 Wm$^{-2}$ solar irradiance. So in the revised paper the CGR3 expanded uncertainty returns to be $\pm 2$ Wm$^{-2}$ and the WINDOW irradiance data have been corrected, either in Table 3 and in Figure 2.
The standard deviations of the LW and WINDOW irradiance, and of the zenith BT over the 10 minute interval are much lower than the measurement uncertainty. For example on 22 June the standard deviation values are 0.2 Wm$^{-2}$ for the LW irradiance, 0.3 Wm$^{-2}$ for the WINDOW irradiance, and 0.1 K for the BT. We assume that no significant variations occur within the 10 minute interval and that differences can be calculated between the average value and the model simulation. A sentence has been added in the text to state the very low variability within the 10 minute time interval.

- All the section 4.1 need

to be slightly reorganized. In particular, the sentence line 23-24 page 10 is very general for the three days and the three variables, and therefore need to be at the beginning of the paragraph, as well as the sentence line 18-21, which is the associated explanation or put at the end of the section as conclusion. Furthermore, the paragraph need an overall analysis of the results obtained at the end: For LW irradiance and WINDOW irradiance, the impact of the refractive index is below the uncertainty of observation, the impact of the RI for a given SD is close to the uncertainty. For WINDOW, simulations always overestimate the observation, implying that the RT model or the calibration is not correct for this simulation and therefore it is not clear to understand what bring this variable in the study: : : On the contrary, for IR BT, the impact of the SD as well as the RI is significant compared to the uncertainty, this variable seems to be more appropriate to analyze AOP.

We agree with the reviewer. Section 4.1 has been revised, the results commented taking into account the model uncertainties, and a concluding sentence has been added at the end of the section. Although the WINDOW irradiance is overestimated by the model, its simulations with different AOPs have been included in the analysis to show that, even when reducing the spectral interval compared to the broadband, the irradiance is not sensitive to varying AOPs.

- P11 line 27: "The average AOD during the descent is assumed as model input.": Which AOD is used? The average column integrated value measured by the MFRSR?
Yes, the AOD measured by the MFRSR during the flight and reported in the IR as explained in section3.1.1 has been averaged and used as model input.

- P11line30-33: there is no reason for which the NOAER simulation agree well for all the profile except close to the surface given the aerosol distribution of Figure 3. Something may be missing in the simulation to reproduce the observed downward irradiance in the lower part of the atmosphere that is not due to aerosols.
Figure 6 shows that model without dust (NOAER) underestimates measurements by an amount that is negligible at higher altitudes and increases close to the surface. This effect is due to emission of infrared radiation by the dust above each altitude layer which induces an increase of the downward LW irradiance, and which depends on the dust optical depth and optical properties.

- P12line 1-3: "This confirms the results found for the surface irradiance, i.e. that the broadband irradiance alone cannot help discriminating which SD and RI provide the best representation of the dust optical properties". This conclusion is not clearly stated in the previous part (see my comment on the section 4.1).
A concluding sentence has been added in Section 4.1 to summarize the results.

- P12line 12-13: "However, while the model-measurement agreement is very good at 600 and 3300 m, where the aerosol impact is small, a systematic overestimation is obtained at 5670 m.": from Figure 7, there is no evidence of a "systematic overestimation […] at 5670 m". COL1 and INSU3 seems to fit well observations at 12_m, whereas an overestimation is obtained at 8.7µm. At 10.6µm COL1 induces an overestimation, but INSU3 an underestimation of the observation. I don't see therefore "a systematic overestimation".
The sentence was present in a previous version of the manuscript and was erroneously maintained in the submitted version. It has been removed. As the reviewer points out, the overestimate of the model depends on spectral band and on aerosol optical properties.

- P12 line 15-16: "These results show that exploring the BT in the thermal infrared is a useful tool to infer dust optical properties if the SD is provided.": this sentence should be slightly attenuated: These results show the better sensitivity of BT to dust optical properties than broadband irradiance but for the two other days, where aerosols are lower or with a smaller AOD, the differences between simulations with different AOP are of the order of the observed uncertainty.

The sentence has been modified according to the reviewer's suggestion in "These results show the better sensitivity of BT to dust optical properties than broadband irradiance", since the paragraph refers to flight F35 only.

- P13line 26: "while no BT increase is detected at 1700 m": I don't understand this sentence, on Figure 9, there is an increase of BT at 1700m.
The reviewer's comment is right. At 1700 m, as well as at 900 m, either the pyrgeometer and the CLIMAT capture the infrared increase due to the island emission. The sentence has been changed in "The spikes at 900 m and around 1700 m indicate that also CLIMAT captures the island emission.".

- As for the section 4.1, this section lacks of a conclusion that summarizes the different simulations. Basically, the LW irradiance is not really sensitive to AOP (impact under the uncertainties). In addition, something appears to be missing in the simulations, because simulated upward irradiances are systematically overestimated by simulation in some part of the profile (for the three cases, even it is less important in the first one). An explanation, or at least some hypothesis, of this overestimation is missing in the paper.
A concluding sentence has been added at the end of the section, explaining that irradiance is not sensitive to the AOPs, while the aerosol perturbation to the upward infrared BT can be appreciated only at altitudes above the bulk aerosol emission, like for the flight F35.
Differences between the measured and the model upward LW irradiance profiles for 28 June and 3 July cannot be explained by a wrong representation of the temperature and humidity profiles in the model, that would have affected the CLIMAT profiles also, as described in the text. What is observed is that the CLIMAT BT profiles are well reproduced at different altitudes, also close to the surface for flight F38, suggesting a proper choice of atmospheric profiles and of sea surface temperature and emission.
We formulated an hypothesis of some negative bias affecting the pyrgeometers' measurements due to the fact that the instrument needs some time (the typical response time is 6 s but for airborne measurements the required time may be significantly longer, as shown for example by Albrecht et al., Pyrgeometer measurements from aircraft, Rev. Sci. Instrum., 45, 33–38, 1974) to establish equilibrium with the air temperature. So we expect that when the aircraft is descending rapidly pyrgeometer measurements may be affected. We verified that during flights F38 (from 3500 to 2000 m) and F42 (from 4800 to 1600 m), where model values are larger than measurements, the vertical velocities were -5.5 and -5.3 m/s, respectively. On the contrary, from 5400 to 4000 m during flight F38, where non-significant biases between model and measurements are observed, the vertical velocity is sensibly lower, about 2.6 m/s. Similarly, during flight F35, when model-measurement differences are small, the vertical velocity is 2.8 m/s.
Opposite to CGR4, the CLIMAT measurements are not affected by the ATR-42 descent speed because the instrumental response time is much shorter (160 ms).

- P14 line 14-15 "The resulting standard deviation on the TOA spectral radiance is around 1% for 22 and 28 June and 0.5% for 3 July": this value requires to be in K, in order to be compared with the radiometric noise and more over to be compared with the impact of the different AOP. Given that 1% corresponds to a variation of about 2.9K (_1% of the surface temperature), this standard deviation is larger than the impact of the aerosol properties themselves. It should be better to apply the simulations to each spectrum and then average the differences.
The IASI spectra have been expressed as BTs, and averaged within the chosen area: the standard deviation (in the 8-14 µm interval) is about 0.6 K for 22 and 28 June, and about 0.3 K on 3 July. These values, although larger than the IASI radiometric noise, express the variability of the TOA BT in the domain. The aim of the simulation of the IASI spectra is to show that including the aerosol the TOA-leaving radiance decreases by an amount which is larger than the model uncertainty and the IASI radiometric noise. Moreover, we want to investigate the sensitivity of the

modelled TOA-leaving radiance to different AOPs. The results in Table 7 show that an appreciable aerosol effect is detected on 22 June and on 3 July, leading to an improvement in the model simulations compared to the aerosol-free case.

- Table 7: In the caption is written "Differences (K) between modelled and measured IASI BT spectra" at the beginning and "Differences are expressed as percent RMSD and standard deviation" at the end. The differences are in K or in %? The sentence "In bold the significant differences with respect to the NOAER simulations." Is not clear. What means "significant differences with respect to the NOAER simulations"? What is the criterion to put in bold the difference?
The differences between model and measurements are calculated in the two spectral intervals 780-980 cm$^{-1}$ and 1070-1200 cm$^{-1}$. Then for each interval the RMSD and the standard deviation are calculated, with the various AOP and also in the aerosol-free case, and expressed in K. The values that have been highlighted in bold are those for which the TOA BT with aerosol and that without aerosol are different taking into account the respective mean and standard deviation (significant difference). The caption of the table has been corrected.

- P14line23 to p15line 6: this paragraph need to be reorganized by day instead of analysing figure 12, and then Table 7 since the conclusions are the same and it would avoid redundant sentences.
The paragraph has been reorganized in order to present the results more concisely and clearly.

- P15 line 4-5: this sentence repeat statement already given in the previous paragraph or has to be rewritten.
The sentence has been removed.

- P15 line 7-24: the paragraph
describing the analysis of Liuzzi et al. (2017) is very long to finally conclude that the impact of INSU3 is of the same order. Either better details of what this study bring compared to the previous one, or why this study use a simplify RT models compared to the previous one is given, or this paragraph has to be shortened. But for now, it's difficult to see where the author is going.
Liuzzi et al. (2017) use an ad-hoc model to reproduce the IASI TOA spectra with the native spectrometer resolution, with the *in situ* size distribution and Di Biagio et al. (2017) refractive indices, tuning some parameters like SST to achieve a good agreement with the measured spectra. We use a model with lower spectral resolution and all the atmospheric and surface parameters derived from observations.
Achieving similar results of those of Liuzzi et al. (2017) at TOA and reproducing also surface and atmospheric irradiances and BT is not obvious and represents an important result of the closure study.
The description of the analysis carried out by Liuzzi et al. has been shortened and the aim of the simulations of the IASI spectra has been better explained.

- P15 line 10: correct "or" by "for" in "The real part is also generally lower or Shettle and Fen "
Done.

- Section 4.4: it would be interesting to see also the results for the two other days and the AOP INSU1 in order to have an idea of the variations of the radiative forcing from very different cases.
The aerosol radiative forcing and heating rate obtained with the INSU1 AOPs have been added and discussed. The ARF and AHR for the other cases have not been presented because the paper is already long. Below we report the tables with the ARF and ARFE for 28 June and 3 July. The values of the ARF is low on both days, either at the surface and at TOA. When taking into account the uncertainties on the modelled ARF, the ARF is different from zero using INSU1 on 28 June and

with INSU1 and INSU3 (only at the surface) on 3 July. It is worth noticing that the ARFE values are similar in the two days.

The ARF values with INSU3 AOPs are much lower on 28 June and 3 July than on 22 June and we expect an analogous behaviour for the aerosol heating rates.

Table 1. LW ARF and ARFE at the surface, TOA, and in the atmosphere (in W m$^{-2}$) on 28 June calculated with all the AOPs.

| AOP | ARF | | | | ARFE | | | |
|---|---|---|---|---|---|---|---|---|
| | COL1 | COL3 | INSU1 | INSU3 | COL1 | COL3 | INSU1 | INSU3 |
| Surface | 3.0 | 1.4 | 5.7 | 3.6 | 14.3 | 6.7 | 27.1 | 17.1 |
| TOA | 1.3 | 0.6 | 2.5 | 1.8 | 6.2 | 2.9 | 11.9 | 8.6 |
| Atmosphere | -1.7 | -0.8 | -3.2 | -1.8 | -8.1 | -3.8 | -15.2 | -8.5 |

Table 2. LW ARF and ARFE at the surface, TOA, and in the atmosphere (in W m$^{-2}$) on 3 July calculated with all the AOPs.

| AOP | ARF | | | | ARFE | | | |
|---|---|---|---|---|---|---|---|---|
| | COL1 | COL3 | INSU1 | INSU3 | COL1 | COL3 | INSU1 | INSU3 |
| Surface | 3.8 | 2.1 | 7.1 | 4.4 | 14.7 | 8.1 | 27.4 | 17.0 |
| TOA | 1.5 | 0.8 | 2.6 | 1.6 | 5.8 | 3.1 | 10.0 | 6.2 |
| Atmosphere | -2.3 | -1.3 | -4.5 | -2.8 | -8.9 | -5.0 | -17.4 | -10.8 |
Review of paper: acp-2017-591 "Determining the infrared radiative effects of Saharan dust: a radiative transfer modelling study based on vertically resolved measurements at Lampedusa" by D. Meloni et al.

General comments
In this paper radiation closure experiments are made in order to determine the infrared radiative effects of dust and to assess the role of dust size distribution (SD) and refractive index (RI). To this aim, in situ data from aircraft (ATR-42 and Falcon), surface (AERONET, radiometer, pyranometers, pyrgeometers and pyrometer) radiosonde and satellite (IASI) measurements are utilized for the closure. The measurements come from the ADRIMED/ChArMEx campaign in 2013. The vertically resolved simulations are performed with the MODTRAN radiative transfer model (RTM) initialized by insitu vertical and remotely sensed columnar SD and RI along with data for a series of surface and atmospheric parameters relevant to LW radiation transfer, coming from radiosoundings, spectrophotometer measurements, ECMWF reanalysis and MODIS satellite products. The assessment lies in comparing simulated and measured LW irradiances and brightness temperatures (BTs), while the dust LW radiative forcing (ARF) and atmospheric heating/cooling rates (AHR) is estimated with the RTM. Three cases (summer days) during a period of dust intrusions (late June and early July) are examined, and the study is performed for Lampedusa in central Meditteranean, in proximity to northern Africa and Sahara.
The study is detailed and makes synergistic use of a variety of data. Some interesting findings are reported, from which some are not always new, e.g. that the dust LW radiative effects are non negligible or that the heating rate profile of dust depends on its vertical distribution as well as on SD and RI. Yet, some others provide new information and give insight regarding the role of dust SD and RI for their LW radiative and thermal effects and for BT, e.g. that using dust RI from local dust sources (Algeria and Morrocco, DB2017) produces best agreement with observations or that the use of inaccurate, although optically equivalent SD and RI has a large impact on the dust ARF. The paper is well organized and nicely written although it sometimes lacks clarity in the discussion of its results.
The main issue is that the paper seems to fail to convince about the best performance and appropriateness, and to provide a clear message on what is the optimal combination of dust properties for achieving the radiation closure. The relevant messages drawn from the simulations-measurements comparisons of LW and WINDOW fluxes, and of BTs, are not consistent and appear to be somewhat contradictory, as it is for ex- ample the case in Table 3. Even the authors state (page 15, lines 20-21) that "the MODTRAN spectral resolution impacts the standard deviation of the model-measurements differences, making the results obtained with different AOPs equivalent". More specifically:

Main Comments
1) In general, quite small differences between the 7 examined configurations, consisting in different model setups (Table 2), are found between results obtained without aerosols and with aerosols, as well as between the 6 configurations with aerosols (3 columnar and 3 in-situ). This does not help to draw a clear conclusion on which one configuration and aerosol properties combination is the best, although this is expected to be the main finding of such a radiation closure study.

The study is aimed at investigating how dust particles affect various radiation quantities ( irradiance and BT at the surface and in the atmosphere, BT at the top of the atmosphere) compared to the aerosol-free case and how the magnitude of the dust radiative effect depends on different aerosol optical properties.

This study has been carried out with information on the atmospheric vertical structure and surface characteristics, as well as on the aerosol burden and physical properties, derived from observations. Specific values of the dust complex refractive index, including some recently determined region-dependent values, have been used. The model outputs are then compared with measurements from various instruments installed on different platforms (from the surface radiometers and pyrometer to the airborne radiometers, to the satellite IASI interferometer).

The aim is not the determination of the best combination of aerosol size distribution and refractive index. In fact, we have used the vertically resolved in situ measured size distribution as a reference, since it is directly measured and, in our opinion, best represents the occurring aerosol properties. Under this assumption, we show that the more recent determination of regionally dependent refractive indices perform better than the frequently used literature values.

The use of vertically averaged SDs derived from AERONET with the most commonly used values of RIs (as it is quite frequently done in similar studies) is finalized at assessing the influence they have on the radiation field.

It must be pointed out that the closure is done on a quite large number of radiation measurements, made at the surface, airborne (and at different altitudes), and on satellite. In our opinion this is a quite unique analysis, and the comparison with different types of radiation measurements gives strong constraints and robustness on the results.

It must be also said that unfortunately the atmospheric conditions occurring during the ChArMEx/ADRIMED campaign did not bring large AOD and this aspect, combined with the model and measurements uncertainties, causes the modelled LW irradiances to be equivalent. Nonetheless, we show that this is not true for the WINDOW and the IR zenith BT, for which significant differences are obtained for the *in situ* SD when using OPAC and DB2017 RIs.

For the day with the largest AOD, i.e. 22 June, we also assessed which combination of SD and RI gives the best model-measurement match for the overall set of LW irradiances (downward at the surface, upward and downward components on the ATR-42 and Falcon 20) by calculating the RMSD of all model-measurement absolute differences, and selecting only those AOPs for which the RMSD is below the $\pm 5$ W m$^{-2}$ threshold value. For the AERONET SD all the three RIs meet the requirement (RMSD between 3.2 and 3.3 W m$^{-2}$), while for the *in situ* SD only the DB2017 (RMSD 4.7 W m$^{-2}$). This conclusion has been added at the end of Section 4.2. If we assume that the vertically-resolved *in situ* SD is the best representation of the effective SD, the DB2017 RIs provide the best AOPs.

2) The ascertained/computed differences of each one of 7 configurations with respect to measurements (LW, WINDOW, BTs) mostly fall within the range of uncertainty of measurements, making difficult to decide on which one is really the best configuration.

This is true for the LW and WINDOW irradiances, but not for the zenith BT. We added a sentence at the end of Section 4.1 summarizing the best combination of SD and RI that provide the best model-measurement match: "The final results of the analysis of the surface measurements show that irradiances, either broadband and in the 8-14 μm spectral interval, are not useful to reduce the uncertainty on the dust RI, since the impact of different RIs is below the measurement and model uncertainty. On the contrary, narrowband zenith BT seems to be suitable to constrain the dust RI which better represents the dust optical properties either in moderate and in low dust loading conditions. Under the assumption that the in situ SD is the most representative of the real aerosol dimensions, the DB2017 RI provides the best agreement between model and measurements, either LW and WINDOW irradiances, and sky BT in the two cases where the atmospheric meteorological profiles and the in situ SD are measured down to surface level (22 and 28 June)."

3) A main conclusion drawn from the analysis is that there is a systematic model overestimation of upward LW fluxes within the peak of dust layers, in all 3 days. In other words, there seems to be an inherent problem with the modelling tool, which needs to be assessed.

We believe that a modeling problem is difficult to expect, either because the CLIMAT BT, obtained with the same input parameters as the irradiance components, are very well reproduced, and because for the 22 June case the model succeeds in resolving both irradiances and BTs. We have further investigated this aspect and have tentatively attributed the observed bias to the CGR4 slow time response; in fact, significant model-measurements differences are found when the aircraft velocities during the descents are too high. See details of the answer to point 26 of the specific comments.

4) The estimated small differences between the no-aerosol and aerosol configurations, indicate that the RTM LW computations are relatively insensitive to dust.

That is true but depends mainly on the AOD value. For the 22 June case, (AOD at 500 nm of 0.36) the increase in LW irradiance at the surface due to dust compared to aerosol-free conditions (NOAER) is +4.8, 4.7, 3.3, 10.9, 10.6, and 8.1 $Wm^{-2}$ with COL1, COL2, COL3, INSU1, INSU2, and INSU3, respectively. With the *in situ* SD the dust effect is larger than the uncertainty of LW irradiance measurement (5 $Wm^{-2}$) and of the model (4.2 $Wm^{-2}$).

The values decrease with AOD: indeed, the values for the 28 June case (AOD at 500 nm of 0.21) are +3.2, 1.7, 6.3, 5.1 $Wm^{-2}$ with COL1, COL3, INSU1, and INSU3. This explains why the simulations in aerosol-free conditions agree with measurements within their uncertainties.

For larger AOD, like the case presented in the paper by Meloni et al., 2015, Altitude-resolved shortwave and longwave radiative effects of desert dust in the Mediterranean during the GAMARF campaign: Indications of a net daily cooling in the dust layer, J. Geophys. Res. Atmos., 120, 3386–3407, the perturbation induced by the dust with AOD at 500 nm of 0.59 was 16.2 and 16.1 $Wm^{-2}$ with AOPs analogous to COL1 and INSU1, respectively. In that case the modelled LW irradiance in aerosol-free conditions is outside the expanded uncertainty of the measurements.

5) The reported conclusions are sometimes contradictory. For example in page 17, lines 14-15 it is stated that dust RI from DB2017 produces the best agreement with observations, but this is not supported by and it is not in line with the results of Table 3 where NOAER and COL1 also provide good results, even better than INSU3, if all three parameters, i.e. LW, WINDOW, BT, and three days are considered.

The sentence refers to the results obtained with the *in situ* size distribution. One point that has been clarified in Sections 4.1 and 4.2 is that we can assume that the *in situ* SD is more realistic than the AERONET one, since the first is derived from vertically resolved optical counter measurements covering the diameter range 0.032-32 μm, while the second is derived from surface visible and near-infrared radiance measurements and is representative of the whole atmospheric column. Under this assumption the best model-measurement agreement is obtained with INSU3 AOPs, as stated in page 15, line 32.

6) It is not clear why BTs were computed and are reported only at 3 levels, which sometimes are not collocated with the peaks of dust layers; why similar BT computations were not made at more levels.

The selected altitudes correspond to those witha horizontal flying attitude of the ATR-42. Moreover, while all the other simulated profiles (upward and downward LW irradiances) where obtained with a single model run, the CLIMAT BTs require a run for each altitude. The vertical profiles of the modelled irradiances show a smooth change with altitude, so we believe that BT at few altitudes are sufficient to describe the vertical variations and adding simulations at other altitudes would not provide additional information.

7) The conclusions drawn from the BT analysis are different from those obtained from the analysis of LW fluxes. This is for example the case of the results of profile 42, in Figures 10 and 11. May this point to a possible modelling problem/inconsistency?

The model has proven to perform well either for the LW fluxes and for the CLIMAT BTs on 22 June, so we cannot justify the results for flight F42 with model inconsistency. We also exclude as a cause an incorrect choice of the model input parameters, like the sea surface temperature, emissivity, and/or the atmospheric temperature/humidity profiles, because the CLIMAT BTs are well reproduced at all altitudes. Following a further check on the data, We found that the bias in the LW irradiances occurs where the ATR-42 descent velocity is high. We believe that the associated fast change of ambient air temperature, associated with the relatively long response time of the CGR4 pyrgeometer, may produce the bias. See details of the answer to point 26 of the specific comments.

8) The role of clouds is not reported. Were all the tree days/cases cloud-free? If so, how is this confirmed/ensured? A relevant discussion should be made since the effect of clouds on LW is significant and interplay or even dominate the effect of dust (e.g. possible implications for Fig. 2).

The three flights were carried out under cloud-free conditions.

The sky conditions at Lampedusa were monitored by the TSI-440 sky imager, collecting hemispheric pictures every minute. The absence of clouds during the flights is also confirmed by the time series of downward SW, LW and WINDOW irradiances, and by the information provided by the zenith-looking pyrometer and lidar at the surface. The profiles of downward LW irradiance from airborne radiometers also show that clouds were not detected: indeed the signal due to cloud emission would have been evident in the measurements with positive spikes, like those in Figure 2. A sentence has been added to the description of Figure 2, to highlight the large increase in irradiance/BT due to cloud presence.

Specific Comments

1. Page 1, line 28: define IASI acronym.

The acronym has been defined.

2. Page 4, Figure 1: the AERONET AOD may also be overplotted.

We received a comment from Reviewer #1 saying that we presented many instruments but some of them were not used in the analysis, and he/she suggested to describe only instruments and measurements that were actually used. We used MFRSR AOD measurements because of their higher temporal resolution (about 1 minute) compared to that of the Cimel sunphotometer (about 15 minutes). Moreover, some Cimel data are missing because of some malfunctioning of the instrument.

However, di Sarra et al., Empirical correction of MFRSR aerosol optical depths for the aerosol forward scattering and development of a long-term integrated MFRSR-Cimel dataset at Lampedusa, Appl. Optics, 54, 2725-2737, 2015, compare MFRSR and AERONET AOD measurements, deriving a mean bias in AOD not larger than 0.004 and a root mean square difference ≤0.031 at all wavelengths. Plotting the available AERONET AOD with the MFRSR AOD would require some discussion. Since the paper is already long we prefer not to add the AERONET AOD measurements.

3. Page 4, line 18: the reported angstrom exponent is high, it is about the maximum one; give a more realistic value (range).

During the first days of the campaign (17-20 June) the Ångström exponent is between 0.3 and 1.75. The text has been modified accordingly.

4. Page 5, lines 32-35: why? Please explain.

Figure 1 of the paper by Gröbner, J., Wacker, S., Vuilleumier, L., and Kämpfer, N.: Effective atmospheric boundary layer temperature from longwave radiation measurements, J. Geophys. Res., 114, D19116, doi:10.1029/2009JD012274, 2009 is very explicative. 99% of the downward LW irradiance comes from the lowest atmospheric layers, those where most of the water vapour is concentrated. The atmosphere is nearly transparent in the 8-14 μm spectral range, and the radiation in the interval is emitted from the upper layers.

5. Page 5, lines 35-37, "The pyrometer . . . for the IRP BT": this sentence is oversimplified. A quick look at the 3 figures reveals significant differences between BT and irradiances. For example, what happens in June 24 and 25 (when LW-WINDOW curves do not have peaks, opposite to IRP BT)? What about the role of temperature and clouds?

The pyrometer has a narrow field of view (2.6°), the broadband CGR4 has 180° and the CGR3 has 150° FOV. So the pyrometer is able to detect each single cloud entering its FOV, while the same single cloud can have a minor impact on the LW-WINDOW irradiance measured by radiometers with broad FOVs if the rest of the sky is cloud-free. That is the reason why the IRP BT time series has more peaks than irradiances time series.

6. Page 7, line 7: define FWHM acronym.

The acronym has been defined in the revised text.

7. Page 7, line 30: up to which altitudes? How much the use of standard profiles can affect the radiative fluxes? Was any sensitivity study performed to assess this? Especially the LW fluxes should be sensitive.

The standard mid-latitude profiles have been used above the maximum altitude of the radiosounding on 28 June and 3 July, i.e. above 32 km and 26 km, respectively. The surface, as well as the profiles, irradiances and BT are not sensitive to variations of upper level atmospheric profiles.

In the revised manuscript a sensitivity study of the modelled quantities has been performed with respect to the main parameters affecting infrared radiation, either in aerosol-free conditions (i.e. IWV, temperature profile, SST, and surface emission) and with aerosol (we have tested the sensitivity to AOD and to the imaginary part of the dust refractive index).

An increase of 0.3 K in the temperature profile causes a 2.2 $Wm^{-2}$ increase in downward LW irradiance at the surface, decreasing for increasing altitudes (becoming 1.3 $Wm^{-2}$ at the ATR-42 top altitude of 5.7 km, and 0.6 $Wm^{-2}$ at the Falcon 20 altitude of 10 km). The upward LW irradiance increases by 1.8 $Wm^{-2}$ at 5.7 km, by 1.5 $Wm^{-2}$ at 10 km, and by 1.4 $Wm^{-2}$ at the TOA.

8. Page 7, line 33, regarding the absorbing gases: similarly, it would be worth to discuss/assess the sensitivity of fluxes to these parameters, especially given the scaling applied to their vertically distributed values.

The sensitivity of the irradiance to changes in the integrated water vapour has been tested by increasing the average measured value by its uncertainty (+0.2 mm), which is of the same order of magnitude of the IWV standard deviation within the considered time interval.

The downward LW irradiance increases by 0.4 $Wm^{-2}$ at the surface, by 0.1 $Wm^{-2}$ at 5.7 km, and by 0.06 $Wm^{-2}$ at 10 km, while the upward LW irradiance at the TOA decreases by 0.1 $Wm^{-2}$.

The sensitivity to other absorbing aerosols has not been tested.

9. Page 8, about ECMWF: The use of reanalysis data is inevitable in this case. However, an assessment of the induced uncertainties associated with their coarser resolution could be made by comparing similar ECMWF data with available measurements for the other two days. This could provide an estimation of induced uncertainties in June 22.

As suggested by the reviewer, the ECMWF profiles at 12:00 have been compared with the temperature and relative humidity profiles measured by the radiosonde and by the airborne meteorological instruments. The T profile is well reproduced on both days, although the ECMWF profiles does not capture the fine vertical structures due its vertical resolution. An exception is the atmosphere below 4 km on 28 June, where the profile sounded by the ATR-42 presents lower temperatures compared to that of the ECMWF profile. The RH profile of the ECMWF operational model generally follows the measurements, but differences can be large at certain levels, like around 3 km on 3 July.

[Figure]

10. Page 8, line 10: a few words about the measured aerosol properties and the identified aerosol layers can be added. For example, apart from the layers and their extension neither information is given nor reference is made to the type of aerosols in each layer, with reference to corresponding measurements that cloud provide this kind of information.
A short discussion on the aerosol stratification identified by the airborne measurements and by the airmass back trajectory analyses has been included in Sections 3.1 and 3.1.1. A detailed analysis is presented in the paper by Denjean et al. (2016).

11. Page 8, line 14: so, what values of emissivity were assumed in the study? Do they differ and how much from day to day.
The values are the same for all the three cases because surface wind speeds are similar. A sentence has been added in the text to better explain, and some spectral values are included.

12. Page 8, Figure 4: the quality should be improved, e.g. by thicknenning the curves, so that the coloured curves can be more easily discerned.
The quality of the figure has been improved.

13. Page 8, line 29: As mentioned, different factors influence and differentiate the AERONET and in-situ SDs, one important being their different value, .e. columnar versus vertically resolved. The value of detailed measurements is that they provide vertically resolved SDs. Therefore, emphasis should be given to them. Discuss a bit more how the measured SDs differ to AERONET ones, referring to their agreement and disagreement. For example, larger differences appear in June 22 than in July 03. Refer to this difference referring to the nature of vertical profiles of Fig. 3 and the type of aerosols that are present in the different layers of every daily profile.
Table S1 has been produced as Supplement Material with the median radius, standard deviation, and normalized number concentration for each mode of either the AERONET and the *in situ* log-normal size distributions for the layers identified in the three cases. The differences in SD among the various layers have been discussed further, as suggested by the reviewer, and related to the transport pathways and to the mixing of dust with pollution particles.

14. Page 9, line 19: explain why this choice of water soluble RI was made and not any other.
Polluted maritime aerosol is the most probable aerosol type characterizing the lowest atmospheric layers over Lampedusa. This is supported by the analysis of the airmass back trajectories in the boundary layer, showing airmasses originating from Europe or recirculating within the Mediterranean basis, and from the chemical analyses carried out on the aerosol samples (Denjean et al., 2016). Water soluble is one of the components of the maritime aerosol, either clean and polluted, according to the OPAC definitions (see Hess et al., 1998). The other components are sea-salt and soot (on for the polluted maritime). Among the components, water soluble and sea salt have similar values of the imaginary part of the RIs below 8 µm and above 11 µm, while the water soluble is more absorbing than sea salt in the 8-11 µm interval. The RI of soot is too absorbing to be representative of the average aerosol. For these reasons the water soluble RI has been chosen.

15. Page 9, lines 20-26: Table 2 is not discussed enough. It should be said more clearly what exactly has been done and how the Mie-based computations of AOD compare to AERONET ones, whenever applicable, i.e. in visible wavelengths.
The combination of the SDs and RIs for the three flights has been better clarified in a new version of Table 2, which includes three tables (one for each flight), describing the SD and RI used in each aerosol layer identified by the ATR-42 and lidar vertical profiles. Moreover, the choice of the different RIs has been better explained in Section 3.1.1.
The spectral AOPs (extinction coefficient, absorption coefficient) accepted by MODTRAN are all referred to the extinction coefficient at 550 nm, for which the vertical distribution, $ext_{550}(z)$, is derived from the lidar backscatter profile and AOD measurements.
So the AOD at 550 nm is fixed, and corresponds to the value obtained from the MFRSR measurements at 500 and 868 nm using the Ångström law. AOPs are allowed to differ in no more than four aerosol layers in the troposphere.
We compute the spectral AOPs from Mie theory for a single particle in a wavelength grid from 2 to 100 µm and including 550 nm, and then divide them by the calculated extinction coefficient at 550 nm. So we have:

$$ext_\lambda(z) = \frac{ext_\lambda(z)}{ext_{550}(z)}$$
$$abs_\lambda(z) = \frac{abs_\lambda(z)}{ext_{550}(z)}$$

This ensures that the values of the AOD at 550 nm remain constant, whatever the AOPs.

16. Page 10, lines 16-17: does this refer to July 03? In Table 3 no results for INSU2 are displayed.
The sentence originally referred to 22 June. Section 4.1 has been modified in order to be clearer for the reader.

17. Page 10, lines 19-20: why the stronger infrared emission? Is it a matter of larger mass? Please explain.
For a particle with diameter D the absorption (and emission according to Kirchhoff law) and scattering properties are calculated with the Mie theory if D is of the same order of magnitude of the wavelength, as is the case for dust particles in the infrared spectral region. According to Mie theory, the absorption and scattering efficiency ($Q_{abs,ext}$) depend on the complex refractive index and on D and the absorption and scattering coefficient ($\sigma_{abs,ext}$) are proportional to Q and to the particle's cross section $\pi D^2$. Thus increasing the particles' dimension increases the absorption and emission coefficient.

18. Page 10, line 24: clarify that "all cases" refer to LW, WINDOW and IR BT.
"all cases" refers to the three days.

19. Page 10, line 34: here it should be clarified what is exactly the spectral interval/coverage of the measurements (IRP). This not clear based on what is said in page 7, line 6, about the IRP centered at 3 wavelengths etc. It is essential to clarify what is exactly the spectral coverage of measurements since they are used as the reference to which the simulations are compared, and given the significant sensitivity of theoretical computations to the spectral interval. Also explain why the reduction in WINDOW irradiances has different magnitude despite the same spectral reduction (0.4 microns) in different spectral parts.
The infrared pyrometer (IRP) measures the zenith BT in the 9.6-11.5 μm interval, while the CLIMAT measures the nadir BT in three infrared channels centred at 8.7, 10.6, and 12 μm with about 1 μm full width at half maximum. The sentence on page 10, line 34, refers to the WINDOW irradiance, measured by the PMOD/WRC CGR3 modified pyrgeometer, which is sensitive to the radiation in the 8-14 μm spectral interval.
The sensitivity analysis carried out on the WINDOW irradiance shows that reducing the spectral interval by a small amount (0.4 μm) significantly reduces the downward irradiance, and the reduction depends on where in the spectrum the reduction is operated: this is caused by the asymmetry of the irradiance spectrum with respect to the centre of the interval.

20. Section 4.1: what is missing is a critical approach providing insight into possible physical reasons for better agreement between the 7 examined cases. A quite exhaustive and very detailed description of results is made, referring to various numbers (Table 3). This is not enough while it turns to be confusing to the reader. What is more important is to determine which set of AOPs is more efficient and compared better to the measurements for the 3 cases. The discussion should conclude on this, stating at least if there is a "best" choice or if there is not and why. Moreover, in both cases, the discussion should provide a physical basis for the outcome of the analysis and the closure of Table 3. For example, a summary of the results of Table 3 should point to NOAER being the most efficient simulation, providing better results than the other 6 sets of AOPs in 4 cases (out of totally 9, i.e. 3 days by 3 parameters). NOAER is followed by COL1 (3 cases with best performance) and INSU3 (2 cases). So, questions may arise, like why simulations without aerosols

should be more appropriate/realistic, or why INSU3, which may be expected to be the most realistic, is finally not.

Section 4.1 has been modified to better present the results and the conclusions. We state that for the conditions met during the campaign, with moderate and not really large AOD, the surface irradiances are not useful to reduce the uncertainty on the dust RI, since the impact of different RIs is below the measurement and model uncertainty. This also explains why NOAER simulations agree with measurements within their respective uncertainties. On the contrary, narrowband zenith BT seems to be suitable to constrain the dust RI which better represents the dust optical properties either in moderate and in low dust loading conditions, assuming that the *in situ* airborne measurements better describes the local aerosol distribution. Indeed, in the two cases where the atmospheric meteorological profiles and the *in situ* SD are measured down to surface level (22 and 28 June) the zenith BT is well reproduced with the DB2017 RIs.

21. Page 12, line 3: as to upward LW, authors may want to comment on why the smallest differences are for COL1 in Table 5, while the smallest RMSDs in Table 4 are for INSU1.

Tables 4 and 5 show that the upward LW irradiance at Falcon 20 and ATR-42 altitudes is reproduced within model and measurement uncertainties with all AOPs, including the NOAER case. The fact that COL1 AOPs give the best match with Falcon 20 measurements while INSU1 AOPs provide the best agreement with the ATR-42 ones may be attributed to different reasons: among them, the Falcon 20 passage is not exactly simultaneous with the ATR-42; the Falcon 20 simulations may be affected by the ECMWF temperature/humidity profile above 6 km, while the ATR-42 upward irradiance rely on the *in situ* measurements of the meteorological vertical profiles.

22. Figure 7: why only points for NOAER, COL1 and INSU3 are given and not for the other cases? All these appear in Table 6.

Overlapping the points corresponding to all AOPs would have made the Figure 7 very difficult to read. After the reviewer's suggestion, we have considered that presenting the results of COL3, instead of COL1, may be better to show the dust perturbation compared to NOAER and the effect of changing the SD, but not the RI, compared to INSU3. The estimated uncertainties on the calculated BTs are also shown.

23. Page 12, lines 15-16: add "in-situ" before SD. This sentence needs to be re-written, since it is introduced all suddenly without being given evidence and discussed based on the results of Fig. 7.

The sentence has been modified as follows: "These results show the better sensitivity of BT to dust optical properties than broadband irradiance. When considering that *in situ* SD better represents the local aerosol distribution, the DB2017 RI from Algeria and Morocco provide the best model-measurement agreement".

24. Page 12, lines 17-19: while discussion is made no results are shown/given.

This comment is not clear. Figure 6 for the upward LW irradiance and Figure 7 for the CLIMAT BTs show that model simulations with and without aerosols are overlapped below 4 km, and differ at higher altitudes, where the aerosol effect is discernible. Moreover, the differences in BT due to dust compared to the aerosol-free simulations are provided on page 12, lines 19-20.

25. Section 4.2.1: A quite exhaustive discussion is made making frequent reference to numbers that differ a while between the 6 examined cases. Also the question arises why NOAER sometimes performs equally or better than dust-including cases. It could point to potential artifacts due to counteracting effects of other parameters than aerosol, which affect the LW radiation transfer and BT.

The limited aerosol effect is due to the moderate AOD measured during the flights and not to modelling problems, and this is the reason why in some cases (more often for the irradiance but not for the BT) the NOAER simulations agree with measurements. This aspect has been remarked in the revised manuscript (Sections 4.1, 4.2.1, and in the conclusions).

26. Page 12, lines 35-36, "These differences . . . airborne instrumentations": so is there an inherent problem with the model?
We cannot state that there is problem with the model, because the same input parameters allow to fairly reproduce the CLIMAT BTs. Moreover, the LW irradiance is well reproduced below 2000 m and above 4000 m and at the Falcon20 altitude. As discussed, a possible bias in the CGR4 measurements may be found when the air temperature is fastly changing, due to inhomogeneities in the instrument temperature. For example the ATR-42 path (Figure 4b) shows a steep aircraft fall from 3.5 km to 1.2 km. A rapid decrease in altitude, with a consequent increase in temperature, may be not registered by the pyrgeometer, which has 6 seconds response time (1/e), but needs a longer time to establish equilibrium with the ambient air temperature (as also found by Albrecht et al., Pyrgeometer measurements from aircraft, Rev. Sci. Instrum., 45, 33–38, 1974). The same problem may have affected the irradiance measurements on 3 July. This may also explain the slight overestimation of the downward LW irradiance by the model in the same altitude ranges where the upward component is overestimated. The CLIMAT BT measurements is not expected to suffer from the same problem because the response time of the instrument is much faster (about 160 ms).

27. Page 14, line 20" add "was" before "evaluated".
Done.

28. Page 14, line 21, "resampled": how it was done?
IASI spectra have been averaged in 15 $cm^{-1}$ intervals. The first interval is centred at 652.5 $cm^{-1}$ and includes the BT values between 645.0 and 660.0, the second interval is centred at 667.5 $cm^{-1}$, and so on.

29. Page 14, lines 32-37: why there is difference on what provides the best match with reference to best match with the measured spectra and BTs?
The IASI spectra used in the comparison with the model are averaged over a region, instead of instantaneous measurements like those from CLIMAT. Moreover, IASI and CLIMAT measurements are not simultaneous. Finally, measurements are made over different spectral intervals, and the AOPs producing the best agreement with the mode result may somewhat differ. This has been stated in the text.

30. Page 15, lines 2-3: this is not applicable to 780-980/cm for June 22 and 28.
The sentence states that when the AOD is sufficiently large the aerosol perturbation to the BT simulated in aerosol-free conditions is significant, so this does not apply for the 28 June case. About 22 June, this applies clearly to the 1070-1200 $cm^{-1}$ interval for different AOPs, and only for the COL1 combination in the 780-980 $cm^{-1}$ interval.

31. Page 15, lines 20-22, "In our case, . . . AOPs equivalent.": what exactly is it meant by this? By which means. Please explain. Is it implied that this (having very high resolution) is preferable? If so, why? If valid, it would mean that AOPs are not important for accurately computing LW radiation and dust LW radiative effects. Is this the meaning?
The sentence refers to the fact that MODTRAN resolution (0.1 $cm^{-1}$) is lower than that of the IASI measurements and of the σ-IASI-as radiative transfer model (0.01 $cm^{-1}$), so we do not expect model-measurements differences as low as those found by Liuzzi et al. (2017). The sentence has been eliminated and text has been modified, also to answer to Reviewer #1, as follows: "The

limitation in the MODTRAN5 resolution does not allow to reproduce the high resolution IASI spectra: however, the scope of simulating the IASI measurements is to show that TOA BTs are sensitive to the dust presence and to the various AOPs, and that they can be reproduced with the same input parameters that allow to simulate irradiance and BT at the surface and in the atmosphere.".

32. Page 15, line 28, "The combination . . . downward": this is not clearly evidenced in the discussion of sections 4.1 and 4.2.
The sentence has been moved at the end of Section 4.2.

33. Page 26, Table 2: the Table needs further/better explanation, it is not very easy for the reader to understand what exactly is the information given in this Table.
We agree with the reviewer. A new Table 2 has been prepared, which includes three tables, one for each day. The tables present the combination of SD and RI used in each layer identified by the lidar and ATR-42 measurements, and the AOD value at 8.6 μm.

34. Page 37, Figure 5: what have been the criteria for the design of flight paths.? Nothing is said about this and deserves to be mentioned in the text.
This aspect has been treated in Section 2.1(Aircraft strategy) of the paper by Denjean et al. (2016). In particular the following text explains the flight strategy: "The general flight strategy consisted of two main parts. First, profiles from 300m up to 6 km above sea level (a.s.l.) were conducted by performing a spiral trajectory 10–20 km wide to sound the vertical structure of the atmosphere and identify interesting dust layers. Afterwards, the identified dust layers were probed by straight levelled runs (SLRs), where the aircraft flew at fixed altitudes, to provide information on dust spatial variability and properties. Horizontal flight legs in the dust layers lasted 20–40 min to allow aerosol collection on filters for chemical analyses in the laboratory".
The sentence "The ATR-42 sounded the atmosphere during profile descents and ascents to infer the vertical structure and composition and identify layers with different properties" in Section 2.2 clarifies the criteria beyond the flight paths.

35. Page 44, Figure 12: wavelengths could be added, e.g. on the top x-axis.
The top x-axis has been expressed in wavelength units, as suggest by the reviewer.

[revised manuscript text omitted]

---

## Author Response (AR2)

In the following the answers to the Referee's comments are provided in red.

Review :
The paper is now better organized and much more clear. They are still some minor comments I have listed below before the manuscript should be accepted for final publication.
- P10l18-19: "*The AOD at 8.6 µm is calculated from the MFRSR AOD at 500 nm and the ratio between the extinction coefficient at 8.6 µm and 500 nm obtained from the Mie calculations* " : how is obtained the extinction coefficient at 500 nm since the refractive indices of DB2017 are given only in the 2-16 µm interval?
In cases when the RI are provided only in the infrared, like for DB2017 and Volz1973, the extinction at 500 nm is obtained from the Mie calculations assuming the OPAC RIs: mineral for the dust layers and water soluble below the dust layers. This choice has been clarified in the text: "When the RIs from DB2017 and Volz1973 are used, which provide values only in the infrared spectral intervals, the aerosol RIs at 500 nm are obtained from OPAC: in particular the mineral dust RI is assumed for the dust layers and the water soluble one below".

- P1425: "*The agreement is good for both downward and upward LW irradiances with the AERONET SD, and is best with COL1 and COL2 (RMSD of both components 4.1 W m-2).* ": I don't understand why COL3 is not the best? The RMSD of both components is lower than 4 W m-2.
The reviewer is right, the agreement with COL3 is as good as with COL1 and COL2. The sentence has been modified in "The agreement is good for both downward and upward LW irradiances with the AERONET SD, resulting in RMSDs of both components of 4.1 W m$^{-2}$ with COL1 and COL2 and of 4.0 W m$^{-2}$ with COL3".

- P14 21-25: for my opinion, this paragraph should be placed either in the previous paragraph where irradiance results are analysed or in the section 4.4 where the configurations used for the radiative forcing estimation are defined.
We preferred to discuss separately the surface and the airborne measurements. For the latter ones, it is useful to compare irradiance and BT profiles to highlight the role of surface temperature/emission.

- P20l12: "*the scope of simulating the IASI measurements in this work is to show that TOA BTs are sensitive to the dust occurrence and to its AOPs*": since results are important but not new, see for example 2 references given below : Capelle et al., 2014 and Vandenbussche et al., 2013.
We thank the reviewer for suggesting the two papers. Both of them show how the IASI radiances are sensitive to the dust optical properties and to the particles' vertical distribution. What we believe is the added value of our study, is that the IASI BTs can be reproduced with the same input parameters that allow to simulate irradiance and BT at the surface and in the atmosphere, without tuning any parameter, and by including in the simulations all the available and relevant observed quantities.
We have included the two suggested papers in the References and cited them in the Introduction, where we highlight the role of dust SD and RI in the retrieval of dust properties from TOA radiance spectra.

- Fig. 4: The AERONET size distribution for 22 June appears to present 3 modes and not 2 and therefore doesn't correspond to values given in Table S1
The reviewer is right.

The AERONET SD has 3 modes and the parameters provided in Table S1 were erroneously reported. All values of Table S1 have been carefully revised and the significant digits among the different cases have been homogenized.

- Table S1 : Where come from the size distribution parameters for AERONET since they didn't apparently come from the AERONET website (where only two modes are retrieved).
The AERONET SD used in this study is the one retrieved by the inversion method of Dubovik and King (2000) in 22 size bins between 0.05 and 15 μm from the Cimel sunphotometer radiance measurements. The SD is downloadable with other inversion products from the AERONET web page (https://aeronet.gsfc.nasa.gov/cgi-bin/webtool_opera_v2_inv?stage=3®ion=Europe&state=Italy&site=Lampedusa&place_code=10).
Another product provided by the AERONET inversion (not used in our analyses) is the bi-modal volume SD, for which effective radius, volume concentration and median radius, and standard deviation are provided for the fine and the coarse aerosol modes, assuming a separation point within the size interval from 0.194 to 0.576 μm.
The SD has been fitted with a combination of log-normal functions with three or four modes, depending on the case. The correlation coefficient ($R^2$) was better than 0.996 in all cases. A sentence describing the fit has been added in Section 3.1.1.

Capelle, V., Chédin, A., Siméon, M., Tsamalis, C., Pierangelo, C., Pondrom, M., Crevoisier, C., Crepeau, L. and Scott, N. A.: Evaluation of IASI-derived dust aerosol characteristics over the tropical belt, Atmos. Chem. Phys., 14(17), 9343–9362, doi:10.5194/acp-14-9343-2014, 2014.

Vandenbussche, S., Kochenova, S., Vandaele, A. C., Kumps, N. and De Mazière, M.: Retrieval of desert dust aerosol vertical profiles from IASI measurements in the TIR atmospheric window, Atmos. Meas. Tech, 6, 2577–2591, doi:10.5194/amt-6-2577-2013, 2013.

**Anonymous Referee #2**

In the following the answers to the Referee's comments are provided in red.

General comments
5) therefore, add at the end of sentence (page 19, line 36): "among cases with RI derived from in-situ measurements", in order to avoid any confusion.
The sentence has been completed as suggested by the Reviewer: "For the ADRIMED campaign, with dust particles originating from northern Algeria, Tunisia and Morocco, the dust RI from DB2017, derived for soil samples from the same regions, produces the best agreement with observations among the cases with the SD derived from *in situ* measurements".

6) both reasons can be acceptable, but make a relevant explanatory note in the text, because otherwise readers may have a similar question about not providing further vertically resolved calculations.
The explanation of the reason for the computation of the CLIMAT BTs at three altitudes has been added in Section 3.2: "The CLIMAT BTs of each flight case have been computed at three altitudes, corresponding to a horizontal flying attitude of the ATR-42 and in layers characterized by different aerosol properties. The modelled BT profile shows a smooth change with height, therefore even only simulations at three altitudes are sufficient to describe the vertical variations".

8) o.k., but also add a note in the text clarifying that all three studied cases were cloud free.
A sentence has been added in Section 2.2 to assess that the three flights were carried out under cloud-free conditions: "The flights were carried out under cloud-free conditions, as confirmed by the time series of surface solar and infrared radiation measurements and by the hemispheric pictures collected by the TSI-440 sky imager installed at the Station".

Specific comments:
5. if you point to clouds as the reason for the differences, then state this in the text.
The different impact of clouds on the irradiance and on the BT measurements has been clarified with the following sentence included in Section 2.1: "Indeed, a cloud entering the narrow IRP causes a relevant increase in BT (depending on the cloud properties), while its impact on the irradiance is minor if the rest of the hemisphere is free from clouds".

22. fine, there is only a technical problem with the new Fig. 7, where the symbols for NOAER, COL3 and INSU3 appear as not filled with colors.
The symbols used in Figure 7 are associated with the aerosol optical properties, while the color is associated to the wavelength, so we prefer not to fill the symbol with color.

24. what I meant actually, was to refer to Fig. 7 in the text.
A reference to Figure 7 has been place at the end of the sentence in Section 4.2.1: "
[revised manuscript text omitted]

Table S1. Parameters (median radius, standard deviation and normalized number concentration) of the log-normal modes used to represent the observed *in situ* size distributions on F35 (top), F38 (middle) and F42 (bottom) for each layer as defined in the text. Parameters for the AERONET size distribution are also shown.

| F35 | | Mode 1 | Mode 2 | Mode 3 | Mode 4 |
|---|---|---|---|---|---|
| LAYER 3 | N | 0.513428 | 0.394945 | 0.078989 | 0.012638 |
| | R (µm) | 0.039 | 0.070 | 0.200 | 0.775 |
| | Sigma | 1.28 | 1.50 | 1.55 | 2.18 |
| LAYER 2 | N | 0.354811 | 0.638660 | 0.005677 | 0.000852 |
| | R (µm) | 0.039 | 0.070 | 0.185 | 0.650 |
| | Sigma | 1.28 | 1.50 | 1.55 | 2.18 |
| LAYER 1 | N | 0.355771 | 0.640387 | 0.003558 | 0.000285 |
| | R (µm) | 0.045 | 0.070 | 0.140 | 0.675 |
| | Sigma | 1.50 | 1.50 | 1.55 | 2.80 |
| AERONET | N | 0.936191 | 0.063372 | 0.000437 | n.a. |
| | R (µm) | 0.079 | 0.080 | 1.166 | n.a. |
| | Sigma | 1.38 | 2.57 | 1.59 | n.a. |

| F38 | | Mode 1 | Mode 2 | Mode 3 | Mode 4 |
|---|---|---|---|---|---|
| LAYER 2 | N | 0.542476 | 0.433981 | 0.021699 | 0.001844 |
| | R (µm) | 0.039 | 0.070 | 0.175 | 0.500 |
| | Sigma | 1.25 | 1.50 | 1.55 | 2.70 |
| LAYER 1 | N | 0.791805 | 0.197951 | 0.009898 | 0.000346 |
| | R (µm) | 0.045 | 0.065 | 0.150 | 0.250 |
| | Sigma | 1.30 | 1.50 | 1.35 | 2.40 |
| AERONET | N | 0.986422 | 0.013314 | 0.000265 | n.a. |
| | R (µm) | 0.083 | 0.24979 | 1.487 | n.a. |
| | Sigma | 1.488 | 2.12 | 1.65 | n.a. |

| F42 | | Mode 1 | Mode 2 | Mode 3 | Mode 4 |
|---|---|---|---|---|---|
| LAYER 2 | N | 0.361011 | 0.577617 | 0.054152 | 0.007220 |
| | R (µm) | 0.040 | 0.060 | 0.185 | 0.800 |
| | Sigma | 1.25 | 1.50 | 1.55 | 1.80 |
| LAYER 1 | N | 0.364742 | 0.607903 | 0.024316 | 0.003040 |
| | R (µm) | 0.035 | 0.055 | 0.165 | 0.450 |
| | Sigma | 1.25 | 1.50 | 1.55 | 2.50 |
| AERONET | N | 0.965820 | 0.032717 | 0.00139658 | 0.000067 |
| | R (µm) | 0.074 | 0.151 | 0.82549 | 2.21986 |
| | Sigma | 1.48 | 2.24 | 1.60 | 1.57 |

[Figure]

Figure S1. Dust spectral refractive indices used in the analysis, either from the most common literature (like OPAC, Volz 1972 and 1973), and from the most recent paper by Di Biagio et al. (2017) for the Algeria, Morocco and Tunisia source regions.

[Figure]

[Figure]

Figure S2. Extinction coefficients normalized to the value at 550 nm (upper plots), single scattering albedos (middle plots), and asymmetry factors (lower plots) in the 0-40 µm spectral interval computed with the Mie theory using the combination of SDs and RIs described in the text for each layer identified for the flight F35. Please note that COL1 and INSU1 AOPs in LAYER1 are calculated with the OPAC water soluble RI.

[Figure]

[Figure]

Figure S3. Same as Fig. S2, but for flight F38.

[Figure]

[Figure]

Figure S4. Same as Fig. S2, but for flight F42.

---

## Author Response (AR3)

In the following the responses to Dr Dubovik are provided in red.

Dear Dr. Meloni,

I am pleased to accept your paper for publication in ACP.

Best regards,

Oleg Dubovik

Non-public comments to the Author:
I have looked your last version of the paper and have my own very minor comments that you could consider (no obligations).

First, when you talk about spheroid model used in AERONET retrieval, you cite Dubovik et al. (2002b) paper. However, the final model that is used in AERONET is described in Dubovik et al. (2006). It is better to use this reference.

For your future studies, I wanted to indicate that in principle there is effect of desert dust non-sphericty in IR, as discussed by Legrand et al. (2017).

Also, even AERONET level 2 doesn't display SD retrieval for in V2 for AOD > 0.4. The retrieved SD should be of sufficient accuracy as discussed in Dubovik et al. 2000.

Dubovik, O., A. Sinyuk, T. Lapyonok, B. N. Holben, M. Mishchenko, P. Yang, T. F. Eck, H. Volten, O. Munoz, B. Veihelmann, W. J. van der Zander, M. Sorokin, and I. Slutsker, Applica-tion of light scattering by spheroids for accounting for particle non-sphericity in remote sensing of desert dust, J. Geophys. Res., 111, D11208, doi:10.1029/2005JD006619d, 2006.

Legrand, M., O. Dubovik, T. Lapyonok, Y. Derimian, Accounting for particle non-sphericity in modeling of mineral dust radiative properties in the thermal infrared, J. Quant. Spectrosc. Radiait. Transfer, 149, 219-240, 2014.

Dubovik, O., A. Smirnov, B. N. Holben, M. D. King, Y. J. Kaufman, T. F. Eck, and I. Slutsker, "Accuracy assessments of aerosol optical properties retrieved from AERONET Sun and sky-radiance measurements", J. Geophys. Res.,105, 9791-9806, 2000.

Dear Dr Dubovik,

I thank you for your comments and suggestions.

The reference for the spheroid model used in the AERONET retrieval has been changed in Dubovik et al. (2006) as suggested.

The effect of non-sphericity of dust particles on the infrared radiation is known to the authors. However, quantitative information about the particles' shape is not available for the campaign and the spherical assumption has been made in the calculation of the dust optical properties.

Concerning the derived size distribution from the AERONET inversion, the statement contained in the manuscript "Level 2 quality assured inversion products are obtained only for AOD at 440 nm larger than 0.4: this is the case only for the Cimel SD retrieval on 3 July, while on 22 and 28 June Level 1.5 SDs are used" is not fully correct, since the volume size distribution is reasonably accurate in all AOD conditions. The restriction to the cases of AOD at 440 nm larger than 0.4 is applied to the retrieval of the single scattering albedo and the complex refractive index. The sentence has been modified in "The Level 2 quality assured AERONET SD is available only on 3 July, while on 22 and 28 June Level 1.5 SDs are used".

Please also note that some small changes have been made to the Figures to label the multiple panels with letters in brackets and to the captions accordingly.

All the changes to the manuscript have been marked-up in the .docx file.

Best regards,

Daniela Meloni